# NEURAL COLLABORATIVE FILTERING BANDITS VIA META LEARNING

## ABSTRACT

Contextual multi-armed bandits provide powerful tools to solve the exploitation-exploration dilemma in decision making, with direct applications in the personalized recommendation. In fact, collaborative effects among users carry the significant potential to improve the recommendation. In this paper, we introduce and study the problem by exploring 'Neural Collaborative Filtering Bandits', where the rewards can be non-linear functions and groups are formed dynamically given different specific contents. To solve this problem, we propose a meta-learning based bandit algorithm, Meta-Ban (**meta-ban**dits), where a meta-learner is designed to represent and rapidly adapt to dynamic groups, along with an informative UCB-based exploration strategy. Furthermore, we analyze that Meta-Ban can achieve the regret bound of $\mathcal{O}(\sqrt{nT \log T})$, which is sharper over state-of-the-art related works. In the end, we conduct extensive experiments showing that Meta-Ban outperforms six strong baselines.

## 1 INTRODUCTION

The contextual multi-armed bandit has been extensively studied in machine learning to resolve the exploitation-exploration dilemma in sequential decision making, with wide applications in personalized recommendation (Li et al., 2010), online advertising (Wu et al., 2016), etc.

Recommender systems play an indispensable role in many online businesses, such as e-commerce platforms and online streaming services. It is well-known that user collaborative effects are strongly associated with the user preference. Thus, discovering and leveraging collaborative information in recommender systems has been studied for decades. In the relatively static environment, e.g., in a movie recommendation platform where catalogs are known and accumulated ratings for items are provided, the classic collaborative filtering methods can be easily deployed (e.g., matrix/tensor factorization (Su and Khoshgoftaar, 2009)). However, such methods can hardly adapt to more dynamic settings, such as news or short-video recommendation, due to: (1) the lack of cumulative interactions for new users or items; (2) the difficulty of balancing the exploitation of current user-item preference knowledge and the exploration of the new potential matches (e.g., presenting new items to the users).

To address this problem, a line of works, clustering of bandits (collaborative filtering bandits) (Gentile et al., 2014; Li et al., 2016; Gentile et al., 2017; Li et al., 2019; Ban and He, 2021), have been proposed to incorporate collaborative effects among users which are largely neglected by conventional bandit algorithms (Dani et al., 2008; Abbasi-Yadkori et al., 2011; Valko et al., 2013; Ban and He, 2020). These works use the graph-based method to adaptively cluster users and explicitly or implicitly utilize the collaborative effects on user sides while selecting an arm. However, this line of works have a significant limitation that they all build on the linear bandit framework (Abbasi-Yadkori et al., 2011) and the user groups are represented by the simple linear combinations of individual user parameters. The linear reward assumptions and linear representation of groups may not be true in real-world applications (Valko et al., 2013).

To learn non-linear reward functions, neural bandits (Collier and Llorens, 2018; Zhou et al., 2020; Zhang et al., 2021; Kassraie and Krause, 2022) have attracted much attention, where a neural network is assigned to learn the reward function along with an exploration strategy (e.g., Upper Confidence Bound (UCB) or Thompson Sampling (TS)). However, this class of works do not incorporate any collaborative effects among users, overlooking the crucial potential in improving recommendation.

In this paper, to overcome the above challenges, we introduce the problem, Neural Collaborative Filtering Bandits (NCFB), built on either linear or non-linear reward assumptions while introducing relative groups. Groups are formed by users sharing similar interests/preferences/behaviors. However, such groups are usually not static over specific contents (Li et al., 2016). For example, two users may both like "country music" but may have different opinions on "rock music". "Relative groups" are introduced in NCFB to formulate groups given a specific content, which is more practical in real problems.

To solve NCFB, we propose a meta-learning based bandit algorithm, Meta-Ban (**Meta-Ban**dits), distinct from existing related works (i.e., graph-based clustering of linear bandits (Gentile et al., 2014; Li et al., 2016; Gentile et al., 2017; Li et al., 2019; Ban and He, 2021)). Inspired by recent advances in meta-learning (Finn et al., 2017; Yao et al., 2019), in Meta-Ban, a meta-learner is assigned to represent and rapidly adapt to dynamic groups, which allows the non-linear representation of collaborative effects. And a user-learner is assigned to each user to discover the underlying relative groups. Here, we use neural networks to formulate both meta-learner and user learners, in order to learn linear or non-linear reward functions. To solve the exploitation-exploration dilemma in bandits, Meta-Ban has an informative UCB-type exploration. In the end, we provide rigorous regret analysis and empirical evaluation for Meta-Ban. To the best of our knowledge, this is the first work incorporating collaborative effects in neural bandits. The contributions of this paper can be summarized as follows:

(1) **Problem.** We introduce the problem, Neural Collaborative Filtering Bandits (NCFB), to incorporate collaborative effects among users with either linear or non-linear reward assumptions.

(2)**Algorithm.** We propose a meta-learning based bandit algorithm working in NCFB, Meta-Ban, where the meta-learner is introduced to represent and rapidly adapt to dynamic groups, along with a new informative UCB-type exploration that utilizes both meta-side and user-side information. Meta-Ban allows the non-linear representation of relative groups based on user learners.

(3) **Theoretical analysis.** Under the standard assumptions of over-parameterized neural networks, we prove that Meta-Ban can achieve the regret upper bound of complexity $\mathcal{O}(\sqrt{nT \log T})$, where $n$ is the number of users and $T$ is the number of rounds. Our bound is sharper than existing related works. Moreover, we provide a correctness guarantee of groups detected by Meta-Ban.

(4) **Empirical performance.** We evaluate Meta-Ban on 10 real-world datasets and show that Meta-Ban significantly outperforms 6 strong baselines.

Next, after introducing the problem definition in Section 2, we present the proposed Meta-Ban in Section 3 together with theoretical analysis in Section 4. In the end, we show the experiments in Section 5 and conclude the paper in Section 6. More discussion regarding **related work** is placed in Appendix Section A.1.

## 2    NEURAL COLLABORATIVE FILTERING BANDITS

In this section, we introduce the problem of Neural Collaborative Filtering Bandits, motivated by generic recommendation scenarios.

Suppose there are $n$ users, $N = \{1, \ldots, n\}$, to serve on a platform. In the $t^{\text{th}}$ round, the platform receives a user $u_t \in N$ and prepares the corresponding $k$ arms (items) $\mathbf{X}_t = \{\mathbf{x}_{t,1}, \mathbf{x}_{t,2}, \ldots, \mathbf{x}_{t,k}\}$ in which each arm is represented by its $d$-dimensional feature vector $\mathbf{x}_{t,i} \in \mathbb{R}^d, \forall i \in \{1, \ldots, k\}$. Then, like the conventional bandit problem, the platform will select an arm $\mathbf{x}_{t,i} \in \mathbf{X}_t$ and recommend it to the user $u_t$. In response to this action, $u_t$ will produce a corresponding reward (feedback) $r_{t,i}$. We use $r_{t,i}|u_t$ to represent the reward produced by $u_t$ given $\mathbf{x}_{t,i}$, because different users may generate different rewards towards the same arm.

Group behavior (collaborative effects) exists among users and has been exploited in recommender systems. In fact, the group behavior is item-varying, i.e., the users who have the same preference on a certain item may have different opinions on another item (Gentile et al., 2017; Li et al., 2016). Therefore, we define a *relative group* as a set of users with the same opinions on a certain item.

**Definition 2.1** (Relative Group). In round $t$, given an arm $\mathbf{x}_{t,i} \in \mathbf{X}_t$, a relative group $\mathcal{N}(\mathbf{x}_{t,i}) \subseteq N$ with respect to $\mathbf{x}_{t,i}$ satisfies

$$1) \; \forall u, u' \in \mathcal{N}(\mathbf{x}_{t,i}), \mathbb{E}[r_{t,i}|u] = \mathbb{E}[r_{t,i}|u']$$
$$2) \; \nexists \, \mathcal{N}' \subseteq N, \text{s.t. } \mathcal{N}' \text{ satisfies 1) and } \mathcal{N}(\mathbf{x}_{t,i}) \subset \mathcal{N}'.$$

Such flexible group definition allows users to agree on certain items while disagree on others, which is consistent with the real-world scenario.

Therefore, given an arm $\mathbf{x}_{t,i}$, the user pool $N$ can be divided into $q_{t,i}$ non-overlapping groups: $\mathcal{N}_1(\mathbf{x}_{t,i}), \mathcal{N}_2(\mathbf{x}_{t,i}), \ldots, \mathcal{N}_{q_{t,i}}(\mathbf{x}_{t,i})$, where $q_{t,i} \leq n$. Note that the group information is **unknown** to the platform. We expect that the users from different groups have distinct behavior with respect to $\mathbf{x}_{t,i}$. Thus, we provide the following constraint among groups.

**Definition 2.2** ($\gamma$-gap). Given two different groups $\mathcal{N}(\mathbf{x}_{t,i}), \mathcal{N}'(\mathbf{x}_{t,i})$, there exists a constant $\gamma > 0$, such that

$$\forall u \in \mathcal{N}(\mathbf{x}_{t,i}), u' \in \mathcal{N}'(\mathbf{x}_{t,i}), |\mathbb{E}[r_{t,i}|u] - \mathbb{E}[r_{t,i}|u']| \geq \gamma.$$

For any two groups in $N$, we assume that they satisfy the $\gamma$-gap constraint. Note that such an assumption is standard in the literature of online clustering of bandit to differentiate groups (Gentile et al., 2014; Li et al., 2016; Gentile et al., 2017; Li et al., 2019; Ban and He, 2021).

**Reward function**. The reward $r_{t,i}$ is assumed to be governed by an unknown function with respect to $\mathbf{x}_{t,i}$ given $u_t$:

$$r_{t,i}|u_t = h_{u_t}(\mathbf{x}_{t,i}) + \zeta_{t,i}, \tag{1}$$

where $h_{u_t}$ is an either linear or non-linear but unknown reward function associated with $u_t$, and $\zeta_{t,i}$ is a noise term with zero expectation $\mathbb{E}[\zeta_{t,i}] = 0$. We assume the reward $r_{t,i} \in [0, 1]$ is bounded, as in many existing works (Gentile et al., 2014; 2017; Ban and He, 2021). Note that online clustering of bandits assume $h_{u_t}$ is a linear function with respect to $\mathbf{x}_{t,i}$ (Gentile et al., 2014; Li et al., 2016; Gentile et al., 2017; Li et al., 2019; Ban and He, 2021).

**Regret analysis**. In this problem, the goal is to minimize the pseudo regret of $T$ rounds:

$$\mathbf{R}_T = \sum_{t=1}^{T} \mathbb{E}[r_t^* - r_t \mid u_t], \tag{2}$$

where $r_t$ is the reward received in round $t$ and $\mathbb{E}[r_t^*|u_t, \mathbf{X}_t] = \max_{\mathbf{x}_{t,i} \in \mathbf{X}_t} h_{u_t}(\mathbf{x}_{t,i})$.

The introduced problem definition above can naturally formulate many recommendation scenarios. For example, for a music streaming service provider, when recommending a song to a user, the platform can exploit the knowledge of other users who have the same opinions on this song, i.e., all 'like' or 'dislike' this song. Unfortunately, the potential group information is usually not available to the platform before the user's feedback. To solve this problem, we will introduce an approach that can infer and exploit such group information to improve the recommendation, in the next section.

**Notation.** Denote by $[k]$ the sequential list $\{1, \ldots, k\}$. Let $\mathbf{x}_t$ be the arm selected in round $t$ and $r_t$ be the reward received in round $t$. We use $\|\mathbf{x}_t\|_2$ and $\|\mathbf{x}_t\|_1$ to represent the Euclidean norm and Taxicab norm. For each user $u \in N$, let $\mu_t^u$ be the number of rounds that user $u$ has been served up to round $t$, i.e., $\mu_t^u = \sum_{\tau=1}^{t} \mathbb{1}\{u_\tau = u\}$, and $\mathcal{T}_t^u$ be all of $u$'s historical data up to round $t$, i.e., $\mathcal{T}_t^u = \{(\mathbf{x}_\tau, r_\tau) : u_\tau = u \wedge \tau \in [t]\}$. $m$ is the width of neural network and $L$ is depth of neural network in the proposed approach. Given a group $\mathcal{N}$, all it's data up to $t$ can be denoted by $\{\mathcal{T}_t^u\}_{u \in \mathcal{N}} = \{\mathcal{T}_t^u | u \in \mathcal{N}\}$. We use standard $\mathcal{O}, \Theta$, and $\Omega$ to hide constants.

## 3 PROPOSED ALGORITHM

In this section, we propose a meta-learning based bandit algorithm, Meta-Ban, to tackle the challenges in the NCFB problem as follows: (1) Challenge 1 (C1): Given an arm, how to infer a user's relative group, and whether the returned group is the true relative group? (2) Challenge 2 (C2): Given a relative group, how to represent the group's behavior in a parametric way? (3) Challenge 3 (C3): How to generate a model to efficiently adapt to the rapidly-changing relative groups? (4) Challenge 4 (C4): How to balance between exploitation and exploration in bandits with relative groups?

---

**Algorithm 1:** Meta-Ban

---

1 **Input:** $T$ (number of rounds), $\nu, \gamma$(group exploration parameter), $\alpha$(exploration parameter), $\lambda$(regularization parameter), $\delta$(confidence level) , $J_1$(number of iterations for user), $J_2$(number of iterations for meta), $\eta_1$(user step size), $\eta_2$(meta step size), $L$(depth of neural network).

2 Initialize $\Theta_0; \theta_0^u = \Theta_0, \mu_0^u = 0, \forall u \in N$

3 Observe one data for each $u \in N$

4 **for** $t = 1, 2, \ldots, T$ **do**

5      Receive a user $u_t \in N$ and observe $k$ arms $\mathbf{X}_t = \{\mathbf{x}_{t,1}, \ldots, \mathbf{x}_{t,k}\}$

6      **for** $i \in [k]$ **do**

7          Determine $u_t$'s relative groups:

            $\widehat{\mathcal{N}}_{u_t}(\mathbf{x}_{t,i}) = \{u \in N \mid |f(\mathbf{x}_{t,i}; \theta_{t-1}^u) - f(\mathbf{x}_{t,i}; \theta_{t-1}^{u_t})| \leq \frac{\nu-1}{\nu}\gamma\}.$

8          $\Theta_{t,i} = \text{GradientDecent\_Meta}\left(\widehat{\mathcal{N}}_{u_t}(\mathbf{x}_{t,i}), \Theta_{t-1}\right)$

9          $\mathbf{U}_{t,i} = f(\mathbf{x}_{t,i}; \Theta_{t,i}) + \alpha \cdot \left( \frac{\|g(\mathbf{x}_{t,i};\Theta_{t,i}) - g(\mathbf{x}_{t,i};\theta_0^{u_t})\|_2}{\sqrt{t}} + \frac{L+1}{\sqrt{2\mu_t^{u_t}}} + \sqrt{\frac{\log(t/\delta)}{\mu_t^{u_t}}} \right)$

10      $i' = \arg_{i \in [k]} \max \mathbf{U}_{t,i}$

11      Play $\mathbf{x}_{t,i'}$ and observe reward $r_{t,i'}$

12      $\mathbf{x}_t = \mathbf{x}_{t,i'};\ r_t = r_{t,i'};\ \Theta_t = \Theta_{t,i'}$

13      $\mu_t^{u_t} = \mu_{t-1}^{u_t} + 1$

14      $\theta_t^{u_t} = \text{GradientDecent\_User}(u_t, \Theta_t)$

15      **for** $u \in N$ *and* $u \neq u_t$ **do**

16          $\theta_t^u = \theta_{t-1}^u;\ \mu_t^u = \mu_{t-1}^u$

---

Meta-Ban has one meta-learner $\Theta$ to represent the group behavior and $n$ user-learners for each user respectively, $\{\theta^u\}_{u \in N}$, sharing the same neural network $f$. Given an arm $\mathbf{x}_{t,i}$, we use $g(\mathbf{x}_{t,i}; \theta) = \nabla_\theta f(\mathbf{x}_{t,i}; \theta)$ to denote the gradient of $f$ for the brevity. The workflow of Meta-Ban is divided into three parts as follows.

**Group inference (to C1).** As defined in Section 2, each user $u \in N$ is governed by an unknown function $h_u$. It is natural to use the universal approximator (Hornik et al., 1989), a neural network $f$ (defined in Section 4), to learn $h_u$. In round $t \in [T]$, let $u_t$ be the user to serve. Given $u_t$'s past data up to round $t - 1$, $\mathcal{T}_{t-1}^{u_t}$, we can train parameters $\theta^{u_t}$ by minimizing the following loss: $\mathcal{L}\left(\mathcal{T}_{t-1}^{u_t}; \theta^{u_t}\right) = \frac{1}{2}\sum_{(\mathbf{x},r) \in \mathcal{T}_{t-1}^{u_t}} (f(\mathbf{x}; \theta^{u_t}) - r)^2.$

Let $\theta_{t-1}^{u_t}$ represent $\theta^{u_t}$ trained with $\mathcal{T}_{t-1}^{u_t}$ in round $t - 1$. The training of $\theta^{u_t}$ can be conducted by (stochastic) gradient descent, e.g., as described in Algorithm 3.

Therefore, for each $u \in N$, we can obtain the trained parameters $\theta_{t-1}^u$. Then, given $u_t$ and an arm $\mathbf{x}_{t,i}$, we return $u_t$'s estimated group with respect to arm $\mathbf{x}_{t,i}$ by

$$\widehat{\mathcal{N}}_{u_t}(\mathbf{x}_{t,i}) = \{u \in N \mid |f(\mathbf{x}_{t,i}; \theta_{t-1}^u) - f(\mathbf{x}_{t,i}; \theta_{t-1}^{u_t})| \leq \frac{\nu - 1}{\nu}\gamma\}. \tag{3}$$

where $\gamma \in (0, 1)$ represents the assumed $\gamma$-gap and $\nu > 1$ is a tuning parameter to trade off between the exploration of group members and the cost of playing rounds.

**Meta learning (to C2 and C3).** In this paper, we propose to use one meta-learner $\Theta$ to represent and adapt to the behavior of dynamic groups. In meta-learning, the meta-learner is trained based on a number of different tasks and can quickly adapt to new tasks with a small amount of new data (Finn et al., 2017). Here, we consider each user $u \in N$ as a task and its collected data $\mathcal{T}_t^u$ as the task distribution. Therefore, Meta-Ban has two phases: User adaptation and Meta adaptation.

*User adaptation.* In the $t^{\text{th}}$ round, given $u_t$, after receiving the reward $r_t$, we have available data $\mathcal{T}_t^{u_t}$. Then, the user parameter $\theta^{u_t}$ is updated in round $t$ based on meta-learner $\Theta$, denoted by $\theta_t^{u_t}$, described in Algorithm 3.

*Meta adaptation.* In the $t^{\text{th}}$ round, given a group $\widehat{\mathcal{N}}_{u_t}(\mathbf{x}_{t,i})$, we have the available collected data $\{\mathcal{T}_{t-1}^u\}_{u \in \widehat{\mathcal{N}}_{u_t}(\mathbf{x}_{t,i})}$. The goal of meta-learner is to fast adapt to these users (tasks). Thus, given an arm $\mathbf{x}_{t,i}$, we update $\Theta$ in round $t$, denoted by $\Theta_{t,i}$, by minimizing the following meta loss:

---

**Algorithm 2:** GradientDecent_Meta $(\mathcal{N}, \Theta_{t-1})$

---

1   $\Theta_{(0)} = \Theta_{t-1}$ (or $\Theta_0$)
2   **for** $j = 1, 2, \ldots, J_2$ **do**
3     **for** $u \in \mathcal{N}$ **do**
4       Collect $\mathcal{T}_{t-1}^u$
5       Randomly choose $\widetilde{\mathcal{T}}^u \subseteq \mathcal{T}_{t-1}^u$
6       $\mathcal{L}\left(\theta_{\mu_{t-1}^u}^u\right) = \frac{1}{2}\sum_{(\mathbf{x},r)\in\widetilde{\mathcal{T}}^u}(f(\mathbf{x}; \theta_{\mu_{t-1}^u}^u) - r)^2$
7     $\mathcal{L}_{\mathcal{N}} = \sum_{u\in\mathcal{N}} \mathcal{L}\left(\theta_{\mu_{t-1}^u}^u\right) + \frac{\lambda}{\sqrt{m}}\sum_{u\in\mathcal{N}} \|\theta_{\mu_{t-1}^u}^u\|_1$.
8     $\Theta_{(j)} = \Theta_{(j-1)} - \eta_2 \nabla_{\{\theta_{\mu_{t-1}^u}^u\}_{u\in\mathcal{N}}} \mathcal{L}_{\mathcal{N}}$
9   **Return:** $\Theta_t = \Theta_{(J_2)}$

---

**Algorithm 3:** GradientDecent_User $(u, \Theta_t)$

---

1   Collect $\mathcal{T}_t^u$    # Historical data of $u$ up to round $t$
2   $\theta_{(0)}^u = \Theta_t$ ( or $\Theta_0$)
3   **for** $j = 1, 2, \ldots, J_1$ **do**
4     Randomly choose $\widetilde{\mathcal{T}}^u \subseteq \mathcal{T}_t^u$
5     $\mathcal{L}\left(\widetilde{\mathcal{T}}^u; \theta^u\right) = \frac{1}{2}\sum_{(\mathbf{x},r)\in\widetilde{\mathcal{T}}^u}(f(\mathbf{x}; \theta^u) - r)^2$
6     $\theta_{(j)}^u = \theta_{(j-1)}^u - \eta_1 \nabla_{\theta_{(j-1)}^u} \mathcal{L}\left(\widetilde{\mathcal{T}}^u; \theta^u\right)$
7   **Return:** $\theta_t^u = \theta_{(J_1)}^u$

---

$\mathcal{L}_{\widehat{\mathcal{N}}_{u_t}(\mathbf{x}_{t,i})} = \sum_{u\in\widehat{\mathcal{N}}_{u_t}(\mathbf{x}_{t,i})} \mathcal{L}\left(\theta_{\mu_{t-1}^u}^u\right) + \frac{\lambda}{\sqrt{m}}\sum_{u\in\widehat{\mathcal{N}}_{u_t}(\mathbf{x}_{t,i})} \|\theta_{\mu_{t-1}^u}^u\|_1$. where $\theta_{\mu_{t-1}^u}^u$ are the stored user parameters in Algorithm 3 at round $t-1$. Here, we add L1-regularization on meta-learner to prevent overfitting in practice and neutralize vanishing gradient in convergence analysis. Then, the meta-learner is updated by: $\Theta = \Theta - \eta_2 \nabla_{\{\theta_{\mu_{t-1}^u}^u\}_{u\in\widehat{\mathcal{N}}_{u_t}(\mathbf{x}_{t,i})}} \mathcal{L}_{\widehat{\mathcal{N}}_{u_t}(\mathbf{x}_{t,i})}$, where $\eta_2$ is the meta learning rate and $\nabla_{\{\theta_{\mu_{t-1}^u}^u\}_{u\in\widehat{\mathcal{N}}_{u_t}(\mathbf{x}_{t,i})}} \mathcal{L}_{\widehat{\mathcal{N}}_{u_t}(\mathbf{x}_{t,i})}$ is the sum of gradients of $\mathcal{L}_{\widehat{\mathcal{N}}_{u_t}(\mathbf{x}_{t,i})}$ with respect to all the user learners in the group $\widehat{\mathcal{N}}_{u_t}$. Algorithm 2 shows meta update with stochastic gradient descent (SGD) .

Note that in linear clustering of bandits (Gentile et al., 2014; Li et al., 2016; Gentile et al., 2017; Li et al., 2019; Ban and He, 2021), they represent the group behavior $\Theta$ by the linear combination of user-learners, e.g., $\Theta = \frac{1}{|\widehat{\mathcal{N}}_{u_t}(\mathbf{x}_{t,i})|}\sum_{u\in\widehat{\mathcal{N}}_{u_t}(\mathbf{x}_{t,i})} \theta_{\mu_{t-1}^u}^u$. This may not be true in real world. Instead, we use the meta adaptation to update the meta-learner $\Theta$ according to $\widehat{\mathcal{N}}_{u_t}(\mathbf{x}_{t,i})$, which can represent non-linear combinations of user learners (Finn et al., 2017; Wang et al., 2020b).

**UCB Exploration (to C4).** To balance the trade-off between the exploitation of the current group information and the exploration of new matches, we introduce the following UCB-based selection criterion. Based on Lemma C.2, with probability at least $1 - \delta$, after $T$ rounds, the cumulative error induced by meta-learner is upper bounded by $\sum_{t=1}^{T} \mathbb{E}_{r_t|\mathbf{x}_t} [|f(\mathbf{x}_t; \Theta_t) - r_t| \mid u_t] \leq$

$$\sum_{t=1}^{T} \underbrace{\frac{\mathcal{O}\left(\|g(\mathbf{x}_t; \Theta_t) - g(\mathbf{x}_t; \theta_0^{u_t})\|_2\right)}{\sqrt{t}}}_{\text{Meta-side info}} + \sum_{u\in N} \mu_t^u \left[\underbrace{\mathcal{O}\left(\frac{L+1}{\sqrt{2\mu_t^u}}\right) + \sqrt{\frac{2\log(t/\delta)}{\mu_t^u}}}_{\text{User-side info}}\right], \qquad \text{where}$$

$g(\mathbf{x}_t; \Theta_t)$ incorporates the discriminative information of meta-learner acquired from the collaborative effects within the relative group $\widehat{\mathcal{N}}_{u_t}(\mathbf{x}_t)$ and $\mathcal{O}(\frac{1}{\sqrt{\mu_t^u}})$ shows the shrinking confidence interval of user-learner to a specific user $u_t$. This bound provides necessary information we should include in the selection criterion ($\mathbf{U}_{t,i}$ in Algorithm 1), which paves the way for the regret analysis (Theorem 4.2). Therefore, we say that the bound $\mathbf{U}_{t,i}$ leverages both the collaborative effects existed

in $\widehat{\mathcal{N}}_{u_t}(\mathbf{x}_{t,i})$ and $u_t$'s personal behavior for exploitation and exploration. Then, we select an arm according to: $\mathbf{x}_t = \arg_{\mathbf{x}_{t,i} \in \mathbf{X}_t} \max(\mathbf{U}_{t,i})$.

To sum up, Algorithm 1 depicts the workflow of Meta-Ban. In each round, given a served user and a set of arms, we compute the meta-learner and its bound for each relative group (Line 5-9). Then, we choose the arm according to the UCB-type strategy (Line 10). After receiving the reward, we update the meta-learner for next round (Line 12) and update the user-learner $\theta^{u_t}$ (Line 13-14) because only $u_t$'s collected data is updated. In the end, we update all the other parameters (Lines 15-16).

**Remark 3.1** (Time Complexity). Recall that $n$ is the number of users. It takes $\mathcal{O}(n)$ to find the group. Given the detected group $\widehat{\mathcal{N}}_u$, let $b$ be the batch size of SGD and $J_2$ be the number of iterations for the updates of Meta-learner. Thus, it takes $\mathcal{O}(|\widehat{\mathcal{N}}_u|bJ)$ to update the meta-learner. Based on the fast adaptation ability of meta-learner, $J_2$ is a typically small number. $b$ is controlled by the practitioner, and $|\widehat{\mathcal{N}}_u|$ is upper bound by $n$. Therefore, the test time complexity is $\mathcal{O}(n) + \mathcal{O}(|\widehat{\mathcal{N}}_u|bJ)$. In the large recommender system, despite the large number of users, given a serving user $u$, the computational cost of Meta-Ban is mostly related to the inferred relative group $\widehat{\mathcal{N}}_u$, i.e., $\mathcal{O}(|\widehat{\mathcal{N}}_u|bJ)$. Inferring $\widehat{\mathcal{N}}_u$ is efficient because it takes $\mathcal{O}(n)$ and only needs to calculate the output of neural networks. Therefore, as long as we can control the size of $\widehat{\mathcal{N}}_u$, Meta-Ban can work properly. The first solution is to set the hyperparameter $\gamma$ to a small value, so $|\widehat{\mathcal{N}}_u|$ is usually small. Second, we confine the size of $|\widehat{\mathcal{N}}_u|$, i.e., we always choose the top-100 similar users for $u$. With a small size of $\widehat{\mathcal{N}}_u(|\widehat{\mathcal{N}}_u| << n)$, Meta-Ban can do fast meta adaptation to $\widehat{\mathcal{N}}_u$ and make prompt decisions. Therefore, it is feasible for Meta-Ban to scale to large recommender systems, with some proper approximated decisions.

## 4 REGRET ANALYSIS

In this section, we provide the regret analysis of Meta-Ban and the comparison with close related works. The analysis is built in the framework of meta-learning under the the over-parameterized neural networks regimen (Jacot et al., 2018; Allen-Zhu et al., 2019; Zhou et al., 2020).

Given an arm $\mathbf{x}_{t,i} \in \mathbb{R}^d$ with $\|\mathbf{x}_{t,i}\|_2 = 1$, $t \in [T], i \in [k]$, without loss of generality, we define $f$ as a fully-connected network with depth $L \geq 2$ and width $m$:

$$f(\mathbf{x}_{t,i}; \theta \text{ or } \Theta) = \mathbf{W}_L \sigma(\mathbf{W}_{L-1}\sigma(\mathbf{W}_{L-2}\dots\sigma(\mathbf{W}_1\mathbf{x}))) \tag{4}$$

where $\sigma$ is the ReLU activation function, $\mathbf{W}_1 \in \mathbb{R}^{m \times d}$, $\mathbf{W}_l \in \mathbb{R}^{m \times m}$, for $2 \leq l \leq L - 1$, $\mathbf{W}^L \in \mathbb{R}^{1 \times m}$, and $\theta, \Theta = [\text{vec}(\mathbf{W}_1)^\intercal, \text{vec}(\mathbf{W}_2)^\intercal, \dots, \text{vec}(\mathbf{W}_L)^\intercal]^\intercal \in \mathbb{R}^p$. To conduct the analysis, we need the following initialization and mild assumptions.

*Initialization.* For $l \in [L-1]$, each entry of $\mathbf{W}_l$ is drawn from the normal distribution $\mathcal{N}(0, 2/m)$; Each entry of $\mathbf{W}_L$ is drawn from the normal distribution $\mathcal{N}(0, 1/m)$.

**Assumption 4.1** (Arm Separability). *For any pair $\mathbf{x}_{t,i}, \mathbf{x}_{t',i'}, t, t' \in [T], i, i' \in [k], (t, i) \neq (t', i')$, these exists a constant $0 < \rho \leq \mathcal{O}(\frac{1}{L})$, such that $\|\mathbf{x}_{t,i} - \mathbf{x}_{t',i'}\|_2 \geq \rho$.*

Assumption 4.1 is satisfied as long as no two arms are identical. Assumption 4.1 is the standard input assumption in over-parameterized neural networks (Allen-Zhu et al., 2019). Moreover, most of existing neural bandit works (e.g., Assumption 4.2 in (Zhou et al., 2020), 3.4 in (Zhang et al., 2021), 4.1 in (Kassraie and Krause, 2022)) have the comparable assumptions with equivalent constraints. They require that the smallest eigenvalue $\lambda_0$ of neural tangent kernel (NTK) matrix formed by all arm contexts is positive, which implies that any two arms cannot be identical. As $L$ can be set manually, the condition $0 < \rho \leq \mathcal{O}(\frac{1}{L})$ can be easily satisfied (e.g., $L = 2$).

Then, we provide the following regret upper bound for Meta-Ban with gradient descent.

**Theorem 4.2.** *Given the number of rounds $T$, assume that each user is uniformly served and set $\widetilde{\mathcal{T}}^u = \mathcal{T}_t^u, \forall t \in [T]$. For any $\delta \in (0,1), 0 < \rho \leq \mathcal{O}(\frac{1}{L}), 0 < \epsilon_1 \leq \epsilon_2 \leq 1, \lambda > 0$, suppose*

$m, \eta_1, \eta_2, J_1, J_2$ satisfy

$$m \geq \widetilde{\Omega} \left( \max \left\{ poly(T, L, \rho^{-1}), e^{\sqrt{\log(\mathcal{O}(Tk)/\delta)}} \right\} \right), \ \eta_1 = \Theta \left( \frac{\rho}{poly(T, L) \cdot m} \right),$$

$$\eta_2 = \min \left\{ \Theta \left( \frac{\sqrt{n}\rho}{T^4 L^2 m} \right), \Theta \left( \frac{\sqrt{\rho \epsilon_2}}{T^2 L^2 \lambda n^2} \right) \right\}, \ J_1 = \Theta \left( \frac{poly(T, L)}{\rho^2} \log \frac{1}{\epsilon_1} \right) \quad (5)$$

$$J_2 = \max \left\{ \Theta \left( \frac{T^5 (\mathcal{O}(T \log^2 m) - \epsilon_2) L^2 m}{\sqrt{n \epsilon_2} \rho} \right), \Theta \left( \frac{T^3 L^2 \lambda n^2 (\mathcal{O}(T \log^2 m - \epsilon_2))}{\rho \epsilon_2} \right) \right\}.$$

*Then, with probability at least $1 - \delta$ over the random initialization, Algorithms 1-3 has the following regret upper bound:*

$$\mathbf{R}_T \leq \mathcal{O}(\sqrt{n}) \left( \sqrt{T} + L\sqrt{T} + \sqrt{2T \log(\mathcal{O}(T)/\delta)} \right) + \mathcal{O}(1).$$

**Comparison with clustering of bandits**. The existing works on clustering of bandits (Gentile et al., 2014; Li et al., 2016; Gentile et al., 2017; Li et al., 2019; Ban and He, 2021) are all based on the linear reward assumption and achieve the following regret bound complexity: $\mathbf{R}_T \leq \mathcal{O}(d\sqrt{Tn} \log T)$.

**Comparison with neural bandits**. The regret analysis in a single neural bandit (Zhou et al., 2020; Zhang et al., 2021) has been developed recently ($n = 1$ in this case), achieving

$$\mathbf{R}_T \leq \mathcal{O}(\tilde{d}\sqrt{T} \log T), \ \ \tilde{d} = \frac{\log \det(\mathbf{I} + \mathbf{H}/\lambda)}{\log(1 + Tn/\lambda)}$$

where $\mathbf{H}$ is the neural tangent kernel matrix (NTK) (Zhou et al., 2020; Jacot et al., 2018) and $\lambda$ is a regularization parameter. $\tilde{d}$ is the effective dimension first introduced by Valko et al. (2013) to measure the underlying non-linear dimensionality of the NTK kernel space.

**Remark 4.3** (Improve $\mathcal{O}(\sqrt{\log T})$). It is easy to observe that Meta-Ban achieves $\mathcal{O}(\sqrt{T \log T})$, improving by a multiplicative factor of $\mathcal{O}(\sqrt{\log T})$ over above existing works. Note that these works (Gentile et al., 2014; Li et al., 2016; Gentile et al., 2017; Li et al., 2019; Ban and He, 2021; Zhou et al., 2020; Zhang et al., 2021) all explicitly apply the Confidence Ellipsoid Bound (Theorem 2 in (Abbasi-Yadkori et al., 2011)) to their analysis, which inevitably introduces the complexity term $\mathcal{O}(\log(T))$. In contrast, Meta-Ban builds generalization bound for the user-learner (Lemma E.1), inspired by recent advances in over-parameterized network (Cao and Gu, 2019), which only brings in the complexity term $\mathcal{O}(\sqrt{\log T})$. Then, we show that the estimations of meta-learner and user-learner are close enough when $\theta$ and $\Theta$ are close enough, to bound the error incurred by the meta-learner (Lemma C.1). Thus, we have a different and novel UCB-type analysis from previous works. These different techniques lead to the non-trivial improvement of $\mathcal{O}(\sqrt{\log T})$.

**Remark 4.4** (Remove Input Dimension). The regret bound of Meta-Ban does not have $d$ or $\tilde{d}$. When input dimension is large (e.g., $d \geq T$), it may cause a considerable amount of error for $R_T$. The effective dimension $\tilde{d}$ may also incur this predicament when the determinant of $\mathbf{H}$ is very large. As (Gentile et al., 2014; Li et al., 2016; Gentile et al., 2017; Li et al., 2019; Ban and He, 2021) build the confidence ellipsoid for $\theta^*$ (optimal parameters) based on the linear function $\mathbb{E}[r_{t,i} \mid \mathbf{x}_{t,i}] = \langle \mathbf{x}_{t,i}, \theta^* \rangle$, their regret bounds contain $d$ because of $\mathbf{x}_{t,i} \in \mathbb{R}^d$. Similarly, (Zhou et al., 2020; Zhang et al., 2021) construct the confidence ellipsoid for $\theta^*$ according to the linear function $\mathbb{E}[r_{t,i} \mid \mathbf{x}_{t,i}] = \langle g(\mathbf{x}_{t,i}; \theta_0), \theta^* - \theta_0 \rangle$ and thus their regret bounds are affected by $\tilde{d}$ due to $g(\mathbf{x}_{t,i}; \theta_0) \in \mathbb{R}^p$ ($\tilde{d}$ reaches to $p$ in the worst case). On the contrary, the generalization bound derived in our analysis is only comprised of the convergence error (Lemma D.1) and the concentration bound (Lemma E.3). These two terms both are independent of $d$ and $\tilde{d}$, which paves the way for Meta-Ban to remove the curse of $d$ and $\tilde{d}$.

**Remark 4.5** (Remove i.i.d. Arm Assumption). We do not impose any assumption on the distribution of arms. However, the related clustering of bandit works (Gentile et al., 2014; Li et al., 2016; Gentile et al., 2017) assume that the arms are i.i.d. drawn from some distribution in each round, which may not be a mild assumption. In our proof, we build the martingale difference sequence only depending on the reward side (Lemma E.3), which is novel, to derive the generalization bound of user-learner and remove the i.i.d. arm assumption.

**Relative group guarantee**. Compared to detected group $\widehat{\mathcal{N}}_{u_t}(\mathbf{x}_{t,i})$ (Eq.(3)), we emphasize that $\mathcal{N}_{u_t}(\mathbf{x}_{t,i})$ ($u_t \in \mathcal{N}_{u_t}(\mathbf{x}_{t,i})$) is the ground-truth relative group satisfying Definition 2.1. Suppose $\gamma$-gap holds among $N$, we prove that when $t$ is larger than a constant, i.e., $t \geq \widetilde{T}$ (as follows), with probability at least $1 - \delta$, it is expected over all selected arms that $\widehat{\mathcal{N}}_{u_t}(\mathbf{x}_t) \subseteq \mathcal{N}_{u_t}(\mathbf{x}_t)$ and $\widehat{\mathcal{N}}_{u_t}(\mathbf{x}_t) = \mathcal{N}_{u_t}(\mathbf{x}_t)$ if $\nu \geq 2$. Then, for $\nu$, we have: (1) When $\nu \uparrow$, we have more chances to explore collaboration with other users while costing more rounds ($\widetilde{T} \uparrow$); (2) When $\nu \downarrow$, we limit the potential cooperation with other users while saving exploration rounds ($\widetilde{T} \downarrow$). More details and proof of Lemma 4.6 are in Appendix F.

**Lemma 4.6** (Relative group guarantee). *Assume the groups in $N$ satisfy $\gamma$-gap (Definition 2.2) and the conditions of Theorem 4.2 hold. For any $\nu > 1$, with probability at least $1 - \delta$ over the random initialization,, there exist constants $c_1, c_2$, such that when*

$$t \geq \frac{n64\nu^2(1+\xi_t)^2 \left( \log \frac{32\nu^2(1+\xi_t)^2}{\gamma^2} + \frac{9L^2 c_1^2 + 4\epsilon_1 + 2\zeta_t^2}{4(1+\xi_t)^2} - \log \delta \right)}{\gamma^2(1 + \sqrt{3n\log(n/\delta)})} = \widetilde{T},$$

*given a user $u \in N$, it holds uniformly for Algorithms 1-3 that*

$$\underset{\mathbf{x}_\tau \sim \mathcal{T}_t^u | \mathbf{x}}{\mathbb{E}}[\widehat{\mathcal{N}}_u(\mathbf{x}_\tau) \subseteq \mathcal{N}_u(\mathbf{x}_\tau)] \quad and \quad \underset{\mathbf{x}_\tau \sim \mathcal{T}_t^u | \mathbf{x}}{\mathbb{E}}[\widehat{\mathcal{N}}_u(\mathbf{x}_\tau) = \mathcal{N}_u(\mathbf{x}_\tau)], if \, \nu \geq 2,$$

*where $\mathbf{x}_\tau$ is uniformly drawn from $\mathcal{T}_t^u | \mathbf{x}$ and $\mathcal{T}_t^u | \mathbf{x} = \{ \mathbf{x}_\tau : u_t = u \wedge \tau \in [t] \}$ is all the historical selected arms when serving $u$ up to round $t$.*

## 5 EXPERIMENTS

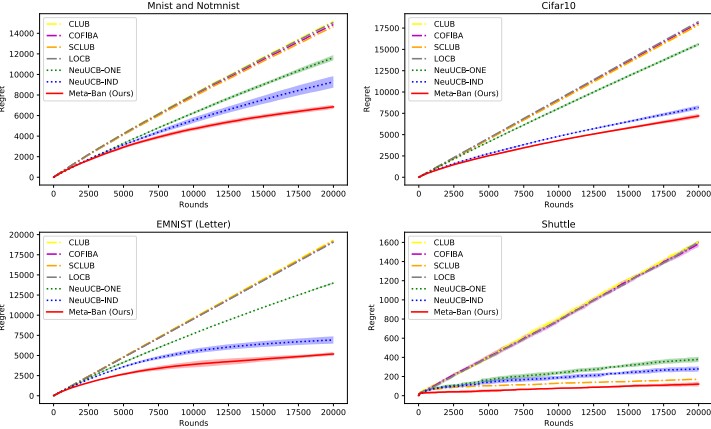

Figure 1: Regret comparison on ML datasets (10 runs). Meta-Ban outperforms all baselines. Specifically, compared to the best baseline, Meta-Ban improves **26.2%** on Mnist and Notmnist, **12.2%** on Cifar10, **25.2%** on EMNIST(Letter), and **28.8%** on Shuttle.

In this section, we evaluate Meta-Ban's empirical performance on 8 ML and 2 real-world recommendation datasets, compared to six strong state-of-the-art baselines. We first present the setup and then the results of experiments. More details are in Appendix A.

*ML datasets.* We totally use 8 public classification datasets: Mnist (LeCun et al., 1998), Notmnist (Bulatov, 2011), Cifar10 (Krizhevsky et al., 2009), Emnist (Letter) (Cohen et al., 2017), Shuttle (Dua and Graff, 2017), Fashion (Xiao et al., 2017), Mushroom (Dua and Graff, 2017), and Magictelescope (Dua and Graff, 2017). Note that ML datasets are widely used for evaluating the performance of neural bandit algorithms (e.g., (Zhou et al., 2020; Zhang et al., 2021)), which test the algorithm's ability in learning various non-linear functions between rewards and arm contexts. On ML datasets, we consider each class as a user to hold an exclusive reward function. As some classes have correlations, the goal of ML datasets also is to find the classes with strong correlations and leverage this information to improve the qualify of classification.

*Recommendation datasets.* We also use two recommendation datasets for evaluation: Movielens (Harper and Konstan, 2015) and Yelp[1]. The descriptions are in Appendix A.2.

*Baselines.* We compare Meta-Ban to six State-Of-The-Art (SOTA) baselines as follows: (1) CLUB (Gentile et al., 2014); (2) COFIBA (Li et al., 2016); (3) SCLUB (Li et al., 2019); (4) LOCB (Ban and He, 2021); (5) NeuUCB-ONE (Zhou et al., 2020); (6) NeuUCB-IND (Zhou et al., 2020). See detailed descriptions in Appendix A.2. Since LinUCB Li et al. (2010) and KernalUCB Valko et al. (2013) are outperformed by the above baselines, we do not include them in comparison.

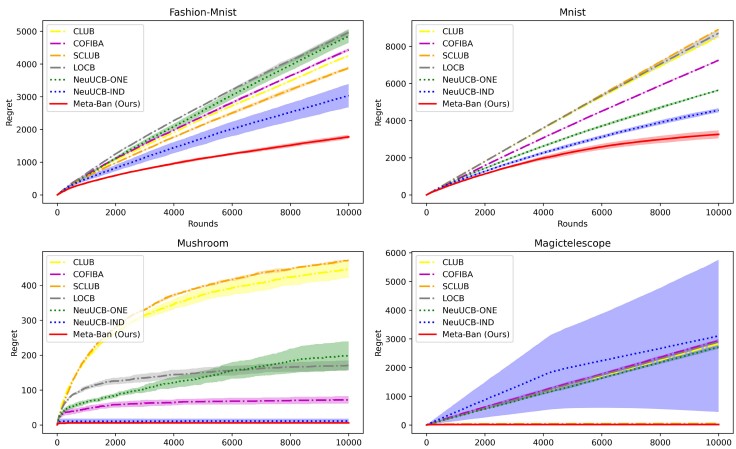

Figure 2: Regret comparison on ML datasets. Meta-Ban outperforms all baselines. Specifically, compared to the best baseline, Meta-Ban improves **41.6%** on Fashion-Mnist, **28.4%** on Mnist, **47.1%** on Mushroom, and **61.5%** on Magictelescope.

**Results.** Figure 1 - 2 shows the regret comparison on ML datasets in which Meta-Ban outperforms all baselines. Each class can be thought of as a user in these datasets. As the rewards are non-linear to the arms on these datasets, conventional linear clustering of bandits (CLUB, COFIBA, SCLUB, LOCB) perform poorly. Thanks to the representation power of neural networks, NeuUCB-ONE obtains better performance. However, it treats all the users as one group, neglecting the disparity among groups. In contrast, NeuUCB-IND deals with the user individually, not taking collaborative knowledge among users into account. Meta-Ban **significantly** outperforms all the baselines, because Meta-Ban can exploit the common knowledge of the correct group of classes where the samples from these classes have non-trivial correlations, and train the parameters on the previous group to fast adapt to new tasks, which existing works do not possess. Figure 3 reports the regret comparison on recommendation datasets where Meta-Ban still outperforms all baselines. Since these two datasets contain considerably inherent noise, all algorithms show the linear growth of regret. As rewards are almost linear to the arms on these two datasets, conventional clustering of bandits (CLUB, COFIBA, SCLUB, LOCB) achieve comparable performance. But they still are outperformed by Meta-Ban because a simple vector cannot accurately represent a user's behavior. Similarly, because Meta-Ban can discover and leverage the group information automatically, it obtains the best performance surpassing NeuUCB-ONE and NeuUCB-IND. Furthermore, hyper-parameter sensitivity study is in Appendix A.3.

## 6 CONCLUSION

In this paper, we introduce the problem, Neural Collaborative Filtering Bandits, to incorporate collaborative effects in bandits with generic reward assumptions. Then, we propose, Meta-Ban, to solve this problem, where a meta-learner is assigned to represent and rapidly adapt to dynamic groups, along with a new informative UCB-type exploration. Moreover, we provide the regret analysis of Meta-Ban and shows that Meta-Ban can achieve a sharper regret upper bound than the close related works. In the end, we conduct extensive experiments to evaluate its empirical performance compared to SOTA baselines.

---

[1]https://www.yelp.com/dataset

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

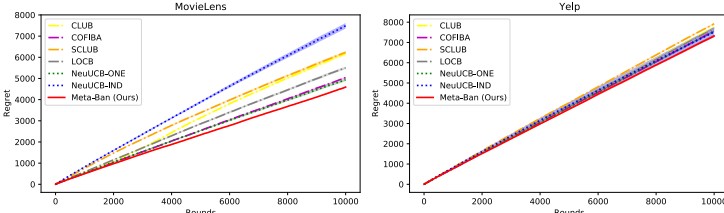

Figure 3: Regret comparison on recommendation datasets (10 runs). Meta-Ban outperforms all baselines. Specifically, compared to the best baseline, Meta-Ban improves **7.02%** on Movielens and **2.6%** on Yelp.

## A SUPPLEMENTARY

In this section, we first introduce the related works and present the experiments setup coming with extensive ablation studies.

### A.1 RELATED WORK

In this section, we briefly review the related works, including clustering of bandits and neural bandits.

**Clustering of bandits.** CLUB Gentile et al. (2014) first studies exploring collaborative effects among users in contextual bandits where each user hosts an unknown vector to represent the behavior based on the linear reward function. CLUB formulates user similarity on an evolving graph and selects an arm leveraging the clustered groups. Then, Li et al. (2016); Gentile et al. (2017) propose to cluster users based on specific contents and select arms leveraging the aggregated information of conditioned groups. Li et al. (2019) improve the clustering procedure by allowing groups to split and merge. Ban and He (2021) use seed-based local clustering to find overlapping groups, different from globally clustering on graphs. Korda et al. (2016); Yang et al. (2020); Wu et al. (2021) also study clustering of bandits with various settings in recommendation system. However, all the series of works are based on the linear reward assumption, which may fail in many real-world applications.

**Neural bandits.** Allesiardo et al. (2014) use a neural network to learn each action and then selects an arm by the committee of networks with $\epsilon$-greedy strategy. Lipton et al. (2018); Riquelme et al. (2018) adapt the Thompson Sampling to the last layer of deep neural networks to select an action. However, these approaches do not provide regret analysis. Zhou et al. (2020) and Zhang et al. (2021) first provide the regret analysis of UCB-based and TS-based neural bandits, where they apply ridge regression on the space of gradients. Ban et al. (2021a) study a combinatorial problem in multiple neural bandits with a UCB-based exploration. Jia et al. (2021) perturb the training samples for incorporating both exploitation and exploration portions. EE-Net(Ban et al., 2021b) proposes to use another neural network for exploration. Xu et al. (2020) combine the last-layer neural network embedding with linear UCB to improve the computation efficiency. Unfortunately, all these methods neglect the collaborative effects among users in contextual bandits. Dutta et al. (2019) use an off-the-shelf meta-learning approach to solve the contextual bandit problem in which the expected reward is formulated as Q-function. Santana et al. (2020) propose a Hierarchical Reinforcement Learning framework for recommendation in dynamic experiment, where a meta-bandit is used for the select independent recommender system. Kassraie and Krause (2022) revisit Neural-UCB type algorithm and show the $\widetilde{\mathcal{O}}(T)$ regret bound without the restrictive assumptions on the context. Maillard and Mannor (2014); Hong et al. (2020) study the latent bandit problem where the reward distribution of arms are conditioned on some unknown discrete latent state and prove the $\widetilde{\mathcal{O}}(T)$ regret bound for their algorithm as well.

**Key Differences from Related Work.** We emphasize that we made important improvements compared to each aspect. (1) Compared to (Gentile et al., 2017), the only similarity is that we adopt the idea of leveraging relative groups. (2) Compared to NeuUCB (Zhou et al., 2020), in addition to the fact that they do not incorporate collaborative filtering effects, we have provided important technical improvements. The UCB in NeuUCB has to maintain a gradient outer product matrix ($Z_t$ in NeuUCB) which occupies space $\mathbb{R}^{p \times p}$ ($\theta \in \mathbb{R}^p$), and only incorporates user-side information. The new UCB introduced in our paper does not need to keep the gradient matrix and contains both group-side and user-side information. (3) Compared to (Wang et al., 2020a), we achieved the

convergence of meta-learner in the online learning setting with bandit feedback, where we need to tackle the challenge that the training data of each round may come from different user distributions.

## A.2 EXPERIMENTS SETUP AND ADDITIONAL RESULTS

**ML Datasets**. In all ML datasets, following the evaluation setting of existing works (Zhou et al., 2020; Valko et al., 2013; Deshmukh et al., 2017), we transform the classification problem into a bandit problem. Take Mnist as an example. Given an image $\mathbf{x} \in \mathbb{R}^d$, it will be transformed into 10 arms, $\mathbf{x}_1 = (\mathbf{x}^\top, 0, \ldots, 0)^\top, \mathbf{x}_2 = (0, \mathbf{x}^\top, \ldots, 0)^\top, \ldots, \mathbf{x}_{10} = (0, 0, \ldots, \mathbf{x}^\top)^\top$, matching 10 class in sequence. The reward is defined as 1 if the index of selected arm equals $\mathbf{x}$' ground-truth class; Otherwise, the reward is 0. In the experiments of Cifar10, Emnist, and Shuttle, we consider each class as a user and randomly draw a class first and then randomly draw a sample from the class. Note that some classes have strong correlations and thus these datasets evaluate one approach's ability to detect and leverage these correlated classes. In the experiments of Mnist and Notmnist (in Figure 1), we add these datasets together as these two both are 10-class classification datasets, to increase the difficulty of this problem. Thus, we consider these two datasets as two groups, where each class can be thought of as a user. In each round, we randomly select a group (i.e., Mnist or Notmnist), and then we randomly choose an image from a class (user). Note that we run all approaches on the Mnist as well (in Figure 2) only this time instead of on Mnist and Notmnist together (in Figure 1).

**Movielens (Harper and Konstan, 2015) and Yelp[2] datasets.** MovieLens is a recommendation dataset consisting of 25 million reviews between $1.6 \times 10^5$ users and $6 \times 10^4$ movies. Yelp is a dataset released in the Yelp dataset challenge, composed of 4.7 million review entries made by $1.18$ million users to $1.57 \times 10^5$ restaurants. For both these two datasets, we extract ratings in the reviews and build the rating matrix by selecting the top 2000 users and top 10000 restaurants(movies). Then, we use the singular-value decomposition (SVD) to extract a normalized 10-dimensional feature vector for each user and restaurant(movie). The goal of this problem is to select the restaurants (movies) with bad ratings (due to the imbalance of these two datasets, i.e., most of the entries have good ratings). Given an entry with a specific user, we generate the reward by using the user's rating stars for the restaurant(movie). If the user's rating is less than 2 stars (5 stars totally), its reward is 1; Otherwise, its reward is 0. From these two datasets, as a single user may not have enough entries to run the experiments, we use K-means to divide users into 50 clusters, where each cluster forms a new user. Therefore, the user pool totally consists of 50 users for these two datasets. Then, in each round, a user to serve $u_t$ is randomly drawn from the user pool. For the arm pool, we randomly choose one restaurant (movie) rated from $u_t$ with reward 1 and randomly pick the other 9 restaurants (movies) rated by $u_t$ with 0 reward. Therefore, there are totally 10 arms in each round. We conduct experiments on these two datasets, respectively.

**Baselines.** We compare Meta-Ban to six State-Of-The-Art (SOTA) baselines as follows: (1) CLUB (Gentile et al., 2014) clusters users based on the connected components in the user graph and refine the groups incrementally. When selecting arm, it uses the newly formed group parameter instead of user parameter with UCB-based exploration; (2) COFIBA (Li et al., 2016) clusters on both user and arm sides based on evolving graph, and chooses arms using a UCB-based exploration strategy; (3) SCLUB (Li et al., 2019) improves the algorithm CLUB by allowing groups to merge and split to enhance the group representation; (4) LOCB (Ban and He, 2021) uses the seed-based clustering and allow groups to be overlapped, and chooses the best group candidates when selecting arms; (5) NeuUCB-ONE (Zhou et al., 2020) uses one neural network to formulate all users and select arms via a UCB-based recommendation; (6) NeuUCB-IND (Zhou et al., 2020) uses one neural network to formulate one user separately (totally $N$ networks) and apply the same strategy to choose arms. Since LinUCB (Li et al., 2010) and KernalUCB (Valko et al., 2013) are outperformed by the above baselines, we do not include them in comparison.

**Configurations.** For all the methods, they all have two parameters: $\lambda$ that is to tune regularization at initialization and $\alpha$ which is to adjust the UCB value. To find their best performance, we conduct the grid search for $\lambda$ and $\alpha$ over $(0.01, 0.1, 1)$ and $(0.0001, 0.001, 0.01, 0.1)$ respectively. For LOCB, the number of random seeds is set as 20 following their default setting. For Meta-Ban, we set $\nu$ as 5 and $\gamma$ as 0.4 to tune the group set. To compare fairly, for NeuUCB and Meta-Ban, we use the same simple neural network with 2 fully-connected layers and the width $m$ is set as 100. To save the

---

[2]https://www.yelp.com/dataset

running time, we train the neural networks every 10 rounds in first 1000 rounds and train the neural networks every 100 rounds afterwards. In our implementation, the gradient descent (Algorithm 2 and 3) stops when the training error is smaller than 0.001, but the $J_1$ and $J_2$ are restricted by 1000. In the end, we choose the best results for the comparison and report the mean and standard deviation (shadows in figures) of 10 runs for all methods.

### A.3 SENSITIVITY STUDY FOR $\nu$ AND $\gamma$

In this section, we conduct the ablation study for the group parameter $\nu$. Here, we set $\gamma$ as a fixed value $0.4$ and change the value of $\nu$ to find the effects on Meta-Ban's performance.

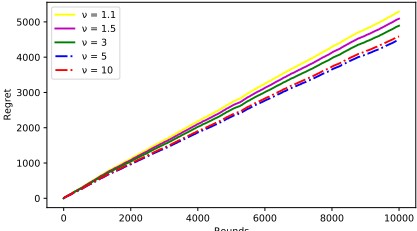

Figure 4: Sensitivity study for $\nu$ on MovieLens Dataset.

Figure 4 shows the varying of performance of Meta-Ban with respect to $\nu$. When setting $\nu = 1.1$, the exploration range of groups is very narrow. This means, in each round, the inferred group size $|\widehat{\mathcal{N}}_{u_t}(\mathbf{x}_{t,i})|$ tends to be small. Although the members in the inferred group $\widehat{\mathcal{N}}_{u_t}(\mathbf{x}_{t,i})$ is more likely to be the true member of $u_t$'s relative group, we may lose many other potential group members in the beginning phase. When setting $\nu = 5$, the exploration range of groups is wider. This indicates we have more chances to include more members in the inferred group, while this group may contain some false positives. With a larger size of group, the meta-learner $\Theta$ can exploit more information. Therefore, Meta-Ban with $\nu = 5$ outperforms $\nu = 1.1$. But, keep increasing $\nu$ does not always mean improve the performance, since the inferred group may consist of some non-collaborative users, bringing into noise. Therefore, in practice, we usually set $\nu$ as a relatively large number. Even we can set $\nu$ as the monotonically decreasing function with respect to $t$.

### A.4 SENSITIVITY STUDY FOR $\alpha$

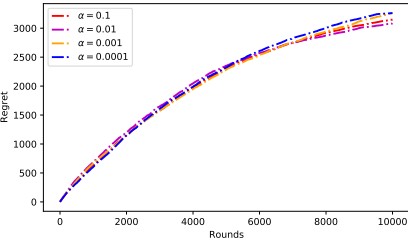

Figure 5: Sensitivity study for $\alpha$ on Mnist Dataset.

Figure 5 depicts the sensitivity of Meta-Ban with regard to $\alpha$. Meta-Ban shows the robust performance as $\alpha$ is varying, which stems from the strong discriminability of meta learner and the derived upper bound. Despite that the magnitude of $\alpha$ is changing, the order of arms ranked by Meta-Ban is slightly influenced. Thus, the Meta-Ban can obtain the robust performance, alleviating the hyperparameter tuning.

### A.5 ABLATION STUDY FOR INPUT DIMENSION

We run the experiments on MovieLens dataset with different input dimensions and report the results as follows. Table 3 summarizes the final regret of each method with different input dimensions (5 runs). Meta-Ban keeps the similar regret with the small fluctuation. This fluctuation is acceptable given that different input dimensions may contain different amount of information. NeuUCB-ONE

and NeuUCB-IND also use the neural network to learn the reward function, so they have the similar property. In contrast, the regret of linear bandits (CLUB, COFIBA, SCLUB, LOCB) is affected more drastically by the input dimensions, which complies with their regret analysis.

| | CLUB | COFIBA | SCLUB | LOCB | NeuUCB-ONE | NeuUCB-IND | Meta-Ban |
|---|---|---|---|---|---|---|---|
| 10 dim | 6174.8 | 5040.4 | 6232.8 | 5499.3 | 4939.6 | 7491.0 | 4673 |
| 20 dim | 6845.4 | 7726.0 | 6742.3 | 5828.6 | 5643.0 | 7888.4 | 5190.2 |
| 50 dim | 7384.6 | 7804.2 | 7563.2 | 6030 | 5403.4 | 7239.0 | 5178.3 |
| 100 dim | 7188 | 7877 | 7433.0 | 6354 | 5523.9 | 7991.1 | 5463.4 |
| 200 dim | 7450.5 | 8463.5 | 7805.1 | 6674.3 | 5821.1 | 8032.6 | 5245.9 |

Table 1: The cumulative regret of 10000 rounds on MovieLens with different input dimensions.

### A.6  ABLATION STUDY FOR NETWORK LAYERS

We run the experiments on MovieLens and Yelp datasets with the different number of layers of neural networks and report the results as follows. Meta-Ban achieves the best performance in the most of cases. In this paper, we try to propose a generic framework to combine meta-learning and bandits with the neural network approximation. Since the UCB in Meta-Ban only depends on the gradient, the neural network can be easily replaced by other different structures.

| | NeuUCB-ONE | NeuUCB-IND | Meta-Ban |
|---|---|---|---|
| 2 layers | 4939 | 7491 | **4673** |
| 4 layers | 5017 | 7603 | **4498** |
| 8 layers | 5033 | 7764 | **4796** |
| 10 layers | 5008 | 7797 | **4824** |

Table 2: The cumulative regret of 10000 rounds on MovieLens with the different number of layers.

| | NeuUCB-ONE | NeuUCB-IND | Meta-Ban |
|---|---|---|---|
| 2 layers | 7683 | 8351 | **7587** |
| 4 layers | **7603** | 8386 | 7767 |
| 8 layers | 7764 | 8366 | **7604** |
| 10 layers | 7797 | 8373 | **7541** |

Table 3: The cumulative regret of 10000 rounds on Yelp with the different number of layers.

## B  PROOF OF THEOREM 4.2

**Theorem B.1** (Theorem 4.2 restated). *Given the number of rounds $T$, assume that each user is uniformly served and set $\widetilde{\mathcal{T}}^u = \mathcal{T}_t^u$, $\forall t \in [T]$. For any $\delta \in (0,1), 0 < \rho \leq \mathcal{O}(\frac{1}{L}), 0 < \epsilon_1 \leq \epsilon_2 \leq 1, \lambda > 0$, suppose $m, \eta_1, \eta_2, J_1, J_2$ satisfy*

$$
\begin{aligned}
&m \geq \widetilde{\Omega}\left(\max\left\{poly(T, L, \rho^{-1}), e^{\sqrt{\log(\mathcal{O}(Tk)/\delta)}}\right\}\right), \quad \eta_1 = \Theta\left(\frac{\rho}{poly(T, L) \cdot m}\right), \\
&\eta_2 = \min\left\{\Theta\left(\frac{\sqrt{n}\rho}{T^4 L^2 m}\right), \Theta\left(\frac{\sqrt{\rho\epsilon_2}}{T^2 L^2 \lambda n^2}\right)\right\}, \quad J_1 = \Theta\left(\frac{poly(T, L)}{\rho^2}\log\frac{1}{\epsilon_1}\right) \\
&J_2 = \max\left\{\Theta\left(\frac{T^5(\mathcal{O}(T\log^2 m) - \epsilon_2)L^2 m}{\sqrt{n}\epsilon_2\rho}\right), \Theta\left(\frac{T^3 L^2 \lambda n^2(\mathcal{O}(T\log^2 m - \epsilon_2))}{\rho\epsilon_2}\right)\right\}.
\end{aligned}
\tag{6}
$$

*Then, with probability at least $1 - \delta$, Algorithms 1-3 has the following regret upper bound:*

$$
\mathbf{R}_T \leq 2\sqrt{n}\left(\sqrt{\epsilon_1 T} + \mathcal{O}\left(L\sqrt{T}\right) + (1 + \xi_1)\sqrt{2T\log(T/\delta)}\right) + \mathcal{O}\left(T\sqrt{\log m}\beta_T^{4/3}L^4\right) + Z_T
$$

*where*

$$
\begin{aligned}
\xi_T =& 2 + \mathcal{O}\left(\frac{T^4 nL\log m}{\rho\sqrt{m}}\right) + \mathcal{O}\left(\frac{T^5 nL^2\log^{11/6} m}{\rho m^{1/6}}\right), \\
\beta_T =& \frac{\mathcal{O}(n^2 T^3\sqrt{\epsilon_2\log^2 m}) + \mathcal{O}(T^2\log^2 m - t\epsilon_2)\rho^{1/2}\lambda n}{\mathcal{O}(\rho\sqrt{m}\epsilon_2)}, \\
Z_T =& \mathcal{O}\left(\frac{T^5 L^2\log^{11/6} m}{\rho m^{1/6}}\right) + T(L+1)^2\sqrt{m\log m}\beta_t^{4/3} + \mathcal{O}\left(\frac{LT^4}{\rho\sqrt{m}}\log m\right) \\
&+ \mathcal{O}\left(\frac{L^4 T^5}{\rho^{4/3}m^{2/3}}\log^{4/3} m\right).
\end{aligned}
$$

*With the proper choice of $m$, we have*

$$
\mathbf{R}_T \leq \mathcal{O}(\sqrt{n})\left(\sqrt{T} + L\sqrt{T} + \sqrt{2T\log(T/\delta)}\right) + \mathcal{O}(1).
\tag{7}
$$

**Proof Overview**. Different from existing works (Zhou et al., 2020; Zhang et al., 2021) that bound the regret of one round by kernel regression in Neural Tangent Kernel , we directly upper bound the mean of regret of overall $T$ rounds by building martingale difference sequence with respect to $h_u$. First, we decompose the regret of $T$ rounds into three key terms (Eq. (9)), where the first term is the error induced by user learner $\theta^u$, the second term is the distance between user learner and meta learner, and the third term is the error induced by the meta learner $\Theta$.

Then, Lemma E.2 provides an upper bound for the first term. Lemma E.2 is an extension of Lemma E.3, which is key to remove input dimension. Lemma E.3 has three terms with the complexity $\mathcal{O}(\sqrt{T})$, where the first term is the training error induced by a class of functions around initialization, the second term is the price of choose the function class, and the third term is confidence interval induced by concentration inequality for $f(\cdot; \theta^u)$. Lemma C.1 bounds the distance between user learner and meta learner. As this bound has the term $\mathcal{O}(1/\sqrt{m})$, this bound can be reduce to $\sqrt{T}$ with proper choice of $m$. Lemma C.2 bounds the error induced by the meta learner using triangle inequality bridged by the user learner.

Bounding the three terms in Eq. (9) completes the proof.

*Proof.* Let $\mathbf{x}_t^* = \arg\max_{\mathbf{x}_{t,i} \in \mathbf{X}_t} h_{u_t}(\mathbf{x}_{t,i})$ given $\mathbf{X}_t, u_t$, and let $\Theta_t^*$ be corresponding parameters trained by Algorithm 2 based on $\widehat{\mathcal{N}}_t^{u_t}(\mathbf{x}_t^*)$. Then, for the regret of one round $t \in [T]$, we have

$$
\begin{aligned}
& R_t | u_t \\
&= \mathop{\mathbb{E}}_{r_{t,i} | \mathbf{x}_{t,i}, i \in [k]} [r_t^* - r_t \mid u_t] \\
&= \mathop{\mathbb{E}}_{r_{t,i} | \mathbf{x}_{t,i}, i \in [k]} \left[ r_t^* - f(\mathbf{x}_t^*; \theta_{t-1}^{u_t,*}) + f(\mathbf{x}_t^*; \theta_{t-1}^{u_t,*}) - r_t \right] \\
&= \mathop{\mathbb{E}}_{r_{t,i} | \mathbf{x}_{t,i}, i \in [k]} \left[ r_t^* - f(\mathbf{x}_t^*; \theta_{t-1}^{u_t,*}) + f(\mathbf{x}_t^*; \theta_{t-1}^{u_t,*}) - f(\mathbf{x}_t^*; \Theta_t^*) \mid \widehat{\mathcal{N}}_t^{u_t}(\mathbf{x}_t^*) + f(\mathbf{x}_t^*; \Theta_t^*) \mid \widehat{\mathcal{N}}_t^{u_t}(\mathbf{x}_t^*) - r_t \right] \\
&\leq \mathop{\mathbb{E}}_{r_t^* | \mathbf{x}_t^*} \left[ r_t^* - f(\mathbf{x}_t^*; \theta_{t-1}^{u_t,*}) \right] + |f(\mathbf{x}_t^*; \theta_{t-1}^{u_t,*}) - f(\mathbf{x}_t^*; \Theta_t^*) \mid \widehat{\mathcal{N}}_t^{u_t}(\mathbf{x}_t^*) | \\
&\quad + \mathop{\mathbb{E}}_{r_t | \mathbf{x}_t} \left[ f(\mathbf{x}_t^*; \Theta_t^*) \mid \widehat{\mathcal{N}}_t^{u_t}(\mathbf{x}_t^*) - r_t \right]
\end{aligned}
$$

(8)

where the expectation is taken over $r_{t,i}$ conditioned on $\mathbf{x}_{t,i}$ for each $i \in [k]$, $\theta_{t-1}^{u_t,*}$ are intermediate user parameters introduced in Lemma E.4 trained on Bayes-optimal pairs by Algorithm 3, e.g., $(\mathbf{x}_{t-1}^*, r_{t-1}^*)$, and $\Theta_t^*$ are meta parameters trained on the group $\widehat{\mathcal{N}}_t^{u_t}(\mathbf{x}_t^*)$ using Algorithm 2. Then, the cumulative regret of $T$ rounds can be upper bounded by

$$
\begin{aligned}
\mathbf{R}_T &= \sum_{t=1}^T R_t | u_t \\
&\leq \sum_{t=1}^T \mathop{\mathbb{E}}_{r_t^* | \mathbf{x}_t^*} \left[ |r_t^* - f(\mathbf{x}_t^*; \theta_{t-1}^{u_t,*})| \right] + \sum_{t=1}^T |f(\mathbf{x}_t^*; \theta_{t-1}^{u_t,*}) - f(\mathbf{x}_t^*; \Theta_t^*)| + \sum_{t=1}^T \mathop{\mathbb{E}}_{r_t | \mathbf{x}_t} \left[ f(\mathbf{x}_t^*; \Theta_t^*) \mid \widehat{\mathcal{N}}_t^{u_t}(\mathbf{x}_t^*) - r_t \right] \\
&\overset{(a)}{\leq} \sum_{u \in N} \left[ \sqrt{\epsilon_1 \mu_T^u} + \mathcal{O}\left( L \sqrt{\mu_t^u} \right) + (1 + \xi_1) \sqrt{2 \mu_t^u \log(T/\delta)} \right] \\
&\quad + \sum_{t=1}^T \left[ \beta_t \cdot \|g(\mathbf{x}_t^*; \Theta_t^*) - g(\mathbf{x}_t^*; \theta_0^{u_t,*})\|_2 + Z_t \right] + \sum_{t=1}^T \mathop{\mathbb{E}}_{r_t | \mathbf{x}_t} \left[ f(\mathbf{x}_t^*; \Theta_t^*) \mid \widehat{\mathcal{N}}_t^{u_t}(\mathbf{x}_t^*) - r_t \right] \\
&\overset{(b)}{\leq} \sum_{u \in N} \left[ \sqrt{\epsilon_1 \mu_T^u} + \mathcal{O}\left( L \sqrt{\mu_t^u} \right) + (1 + \xi_1) \sqrt{2 \mu_t^u \log(T/\delta)} \right] \\
&\quad + \sum_{t=1}^T \left[ \beta_t \cdot \|g(\mathbf{x}_t; \Theta_t) - g(\mathbf{x}_t; \theta_0^{u_t})\|_2 + Z_t \right] + \sum_{t=1}^T \mathop{\mathbb{E}}_{r_t | \mathbf{x}_t} \left[ f(\mathbf{x}_t; \Theta_t) \mid \widehat{\mathcal{N}}_t^{u_t}(\mathbf{x}_t) - r_t \right]
\end{aligned}
$$

(9)

where $(a)$ is the applications of Lemma E.2 and Lemma C.1, and (b) is due to the selection criterion of Algorithm 1 where $\theta_0^{u_t} = \theta_0^{u_t,*}$ according to our initialization. Thus, we have

$$
\begin{aligned}
\mathbf{R}_T &= \sum_{t=1}^{T} R_t | u_t \\
&\leq \sum_{u \in N} \left[ \sqrt{\epsilon_1 \mu_T^u} + \mathcal{O}\left(L\sqrt{\mu_t^u}\right) + (1+\xi_1)\sqrt{2\mu_t^u \log(T/\delta)} \right] \\
&\quad + \sum_{t=1}^{T} \left[ \beta_t \cdot \|g(\mathbf{x}_t;\Theta_t) - g(\mathbf{x}_t;\Theta_0)\|_2 + Z_t \right] \\
&\quad + \sum_{t=1}^{T} \mathbb{E}_{r_t|\mathbf{x}_t} \left[ f(\mathbf{x}_t;\Theta_t) \mid \widehat{\mathcal{N}}_t^{u_t}(\mathbf{x}_t) - f(\mathbf{x}_t;\theta_{t-1}^{u_t}) + f(\mathbf{x}_t;\theta_{t-1}^{u_t}) - r_t \right] \\
&\leq \sum_{u \in N} \left[ \sqrt{\epsilon_1 \mu_T^u} + \mathcal{O}\left(L\sqrt{\mu_t^u}\right) + (1+\xi_1)\sqrt{2\mu_t^u \log(T/\delta)} \right] \\
&\quad + \sum_{t=1}^{T} \left[ \beta_t \cdot \|g(\mathbf{x}_t;\Theta_t) - g(\mathbf{x}_t;\Theta_0)\|_2 + Z_t \right] \\
&\quad + \sum_{t=1}^{T} |f(\mathbf{x}_t;\Theta_t) \mid \widehat{\mathcal{N}}_t^{u_t}(\mathbf{x}_t) - f(\mathbf{x}_t;\theta_{t-1}^{u_t})| + \sum_{t=1}^{T} \mathbb{E}_{r_t|\mathbf{x}_t}\left[ f(\mathbf{x}_t;\theta_{t-1}^{u_t}) - r_t \right] \\
&\stackrel{(c)}{\leq} 2 \sum_{u \in N} \left[ \sqrt{\epsilon_1 \mu_T^u} + \mathcal{O}\left(L\sqrt{\mu_t^u}\right) + (1+\xi_1)\sqrt{2\mu_t^u \log(T/\delta)} \right] \\
&\quad + 2\sum_{t=1}^{T} \left[ \beta_t \cdot \|g(\mathbf{x}_t;\Theta_t) - g(\mathbf{x}_t;\Theta_0)\|_2 + Z_t \right] \\
&\stackrel{(d)}{\leq} 2\sqrt{n}\left( \sqrt{\epsilon_1 T} + \mathcal{O}\left(L\sqrt{T}\right) + \underbrace{(1+\xi_1)}_{I_3}\sqrt{2T\log(T/\delta)} \right) \\
&\quad + \underbrace{2\sum_{t=1}^{T} \beta_t \cdot \|g(\mathbf{x}_t;\Theta_t) - g(\mathbf{x}_t;\Theta_0)\|_2}_{I_1} + \underbrace{2\sum_{t=1}^{T} Z_t}_{I_2}
\end{aligned}
$$

where $(c)$ is an application of Lemma E.1 and Lemma C.1 and (d) is an application of Lemma E.1 with Hoeffding-Azuma inequality.

For $I_1$, recall that $\beta_t = \frac{\mathcal{O}(n^2 t^3 \sqrt{\epsilon_2 \log^2 m}) + \mathcal{O}(t^2 \log^2 m - t\epsilon_2)\rho^{1/2}\lambda n}{\mathcal{O}(\rho\sqrt{m}\epsilon_2)}$. Then, using Theorem 5 in (Allen-Zhu et al., 2019), we have

$$
I_1 \leq \sum_{t=1}^{T} \beta_t \cdot \mathcal{O}\left(\sqrt{\log m}\beta_t^{1/3} L^3 \|g(\mathbf{x}_t;\Theta_0)\|_2\right) \underbrace{\leq}_{E_2} \mathcal{O}\left(T\sqrt{\log m}\beta_T^{4/3} L^4\right) \underbrace{\leq}_{E_3} \mathcal{O}(1) \tag{10}
$$

where , $E_2$ is as the Lemma E.10 and $E_3$ is because of the choice of $m$ ($\beta_t$ has the complexity of $\widetilde{\mathcal{O}}\left(\frac{1}{m^{1/2}}\right)$ and $m \geq \widetilde{\Omega}(T^{30})$).

For $I_2$, recall that $Z_t = \mathcal{O}\left(\frac{(t-1)^4 L^2 \log^{11/6} m}{\rho m^{1/6}}\right) + (L+1)^2 \sqrt{m \log m} \beta_t^{4/3} + \mathcal{O}\left(L\left(\frac{(t-1)^3}{\rho \sqrt{m}} \log m\right)\right)$
$+ \mathcal{O}(L\beta_t) + \mathcal{O}\left(L^4 \left(\frac{(t-1)^3}{\rho \sqrt{m}} \log m\right)^{4/3}\right)$. Then, we have

$$
\begin{aligned}
I_2 \leq &\mathcal{O}\left(\frac{T^5 L^2 \log^{11/6} m}{\rho m^{1/6}}\right) + T(L+1)^2 \sqrt{m \log m} \beta_t^{4/3} + \mathcal{O}\left(\frac{LT^4}{\rho \sqrt{m}} \log m\right) \\
&+ \mathcal{O}\left(\frac{L^4 T^5}{\rho^{4/3} m^{2/3}} \log^{4/3} m\right) \\
=&Z_T.
\end{aligned}
\tag{11}
$$

$I_2$ has the complexity of $\widetilde{\mathcal{O}}\left(\frac{1}{m^{1/6}}\right)$. Therefore, $I_2 \leq \mathcal{O}(1)$ when $m \geq \widetilde{\Omega}(T^{30})$.

For $I_3$, as the choice of $m$, we have $(1 + \xi_1) \leq \mathcal{O}(1)$. The proof is complete. $\qquad\square$

## C    BRIDGE META-LEARNER AND USER-LEARNER

**Lemma C.1.** *For any $\delta \in (0,1), \rho \in (0, \mathcal{O}(\frac{1}{L})], 0 < \epsilon_1 \leq \epsilon_2 \leq 1, \lambda > 0$, suppose $m, \eta_1, \eta_2, J_1, J_2$ satisfy the conditions in Eq.(6). Then, with probability at least $1 - \delta$, for any $t \in [T]$ and $\mathbf{x}_t$ satisfying $\|\mathbf{x}_t\|_2 = 1$, given the serving user $u \in N$ and $\Theta_t$ returned by Algorithm 2 based on $\widehat{\mathcal{N}}_t^u(\mathbf{x}_t)$, it holds uniformly for Algorithms 1-3 that*

$$
|f(\mathbf{x}_t; \theta_{t-1}^u) - f(\mathbf{x}_t; \Theta_t)| \leq \beta_t \cdot \|g(\mathbf{x}_t; \Theta_t) - g(\mathbf{x}_t; \theta_0^u)\|_2 + Z_t,
\tag{12}
$$

*where*

$$
\begin{aligned}
\beta_t =& \frac{\mathcal{O}(n^2 t^3 \sqrt{\epsilon_2 \log^2 m}) + \mathcal{O}(t^2 \log^2 m - t\epsilon_2)\rho^{1/2}\lambda n}{\mathcal{O}(\rho \sqrt{m} \epsilon_2)}, \\
Z_t =& \mathcal{O}\left(\frac{(t-1)^4 L^2 \log^{11/6} m}{\rho m^{1/6}}\right) + (L+1)^2 \sqrt{m \log m} \beta_t^{4/3} \\
&+ \mathcal{O}\left(L\left(\frac{(t-1)^3}{\rho \sqrt{m}} \log m\right)\right).
\end{aligned}
$$

*Proof.* First, we have

$$
\begin{aligned}
|f(\mathbf{x}_t; \theta_{t-1}^u) - f(\mathbf{x}_t; \Theta_t)| \leq &\underbrace{|f_{u_t}(\mathbf{x}_t; \theta_{t-1}^u) - \langle g(\mathbf{x}_t; \theta_{t-1}^u), \theta_{t-1}^u - \theta_0^u\rangle - f(\mathbf{x}_t; \theta_0^u)|}_{I_1} \\
&+ \underbrace{|\langle g(\mathbf{x}_t; \theta_{t-1}^u), \theta_{t-1}^u - \theta_0^u\rangle + f(\mathbf{x}_t; \theta_0^u) - f(\mathbf{x}_t; \Theta_t)|}_{I_2}
\end{aligned}
\tag{13}
$$

where the inequality is using Triangle inequality. For $I_1$, based on Lemma E.9, we have

$$
I_1 \leq \mathcal{O}(w^{1/3} L^2 \sqrt{m \log(m)})\|\theta_{t-1}^u - \theta_0^u\|_2 \leq \mathcal{O}\left(\frac{t^4 L^2 \log^{11/6} m}{\rho m^{1/6}}\right),
$$

where the second equality is based on the Lemma E.8 (4): $\|\theta_{t-1}^u - \theta_0^u\|_2 \leq \mathcal{O}\left(\frac{(\mu_{t-1}^u)^3}{\rho \sqrt{m}} \log m\right) \leq \mathcal{O}\left(\frac{(t-1)^3}{\rho \sqrt{m}} \log m\right) = w$.

For $I_2$, we have

$$
\begin{aligned}
&|\langle g(\mathbf{x}_t; \theta_{t-1}^u), \theta_{t-1}^u - \theta_0^u\rangle + f(\mathbf{x}_t; \theta_0^u) - f(\mathbf{x}_t; \Theta_t)| \\
&\underbrace{\leq}_{E_1} |\langle g(\mathbf{x}_t; \theta_{t-1}^u), \theta_{t-1}^u - \theta_0^u\rangle - \langle g(\mathbf{x}_t; \Theta_t), \Theta_t - \Theta_0\rangle| \\
&\qquad + |\langle g(\mathbf{x}_t; \Theta_t), \Theta_t - \Theta_0\rangle + f(\mathbf{x}_t; \theta_0^u) - f(\mathbf{x}_t; \Theta_t)| \\
&\underbrace{\leq}_{E_2} \underbrace{|\langle g(\mathbf{x}_t; \theta_{t-1}^u), \theta_{t-1}^u - \theta_0^u\rangle - \langle g(\mathbf{x}_t; \theta_0^u), \Theta_t - \Theta_0\rangle|}_{I_3} \\
&\qquad + \underbrace{|\langle g(\mathbf{x}_t; \theta_0^u), \Theta_t - \Theta_0\rangle - \langle g(\mathbf{x}_t; \Theta_t), \Theta_t - \Theta_0\rangle|}_{I_4} \\
&\qquad + \underbrace{|\langle g(\mathbf{x}_t; \Theta_t), \Theta_t - \Theta_0\rangle + f(\mathbf{x}_t; \theta_0^u) - f(\mathbf{x}_t; \Theta_t)|}_{I_5}
\end{aligned}
\tag{14}
$$

where $E_1, E_2$ use Triangle inequality. For $I_3$, we have

$$
\begin{aligned}
&|\langle g(\mathbf{x}_t; \theta_{t-1}^u), \theta_{t-1}^u - \theta_0^u\rangle - \langle g(\mathbf{x}_t; \theta_0^u), \Theta_t - \Theta_0\rangle| \\
&\leq |\langle g(\mathbf{x}_t; \theta_{t-1}^u), \theta_{t-1}^u - \theta_0^u\rangle - \langle g(\mathbf{x}_t; \theta_0^u), \theta_{t-1}^u - \theta_0^u\rangle| + |\langle g(\mathbf{x}_t; \theta_0^u), \theta_{t-1}^u - \theta_0^u\rangle - \langle g(\mathbf{x}_t; \theta_0^u), \Theta_t - \Theta_0\rangle| \\
&\leq \underbrace{\|g(\mathbf{x}_t; \theta_{t-1}^u) - g(\mathbf{x}_t; \theta_0^u)\|_2 \cdot \|\theta_{t-1}^u - \theta_0^u\|_2}_{M_1} + \underbrace{\|g(\mathbf{x}_t; \theta_0^u)\|_2 \cdot \|\theta_{t-1}^u - \theta_0^u - (\Theta_t - \Theta_0)\|_2}_{M_2}
\end{aligned}
\tag{15}
$$

For $M_1$, we have

$$
\begin{aligned}
M_1 &\underbrace{\leq}_{E_3} \mathcal{O}\left(\frac{(t-1)^3}{\rho\sqrt{m}}\log m\right) \cdot \|g(\mathbf{x}_t; \theta_{t-1}^u) - g(\mathbf{x}_t; \theta_0^u)\|_2 \\
&\underbrace{\leq}_{E_4} \mathcal{O}\left(L^4\left(\frac{(t-1)^3}{\rho\sqrt{m}}\log m\right)^{4/3}\right)
\end{aligned}
\tag{16}
$$

where $E_3$ is the application of Lemma E.8 and $E_4$ utilizes Theorem 5 in Allen-Zhu et al. (2019) with Lemma E.8. For $M_2$, we have

$$
\begin{aligned}
&\|g(\mathbf{x}_t; \Theta_0)\|_2 \left(\|\theta_{t-1}^u - \theta_0^u - (\Theta_t - \Theta_0)\|_2\right) \\
&\leq \|g(\mathbf{x}_t; \Theta_0)\|_2 \left(\|\theta_{t-1}^u - \theta_0^u\|_2 + \|\Theta_t - \Theta_0\|_2\right) \\
&\underbrace{\leq}_{E_5} \mathcal{O}(L) \cdot \left[\mathcal{O}\left(\frac{(t-1)^3}{\rho\sqrt{m}}\log m\right) + \beta_t\right]
\end{aligned}
\tag{17}
$$

where $E_5$ use Lemma E.10, E.8, and D.1. Combining Eq.(16) and Eq.(C), we have

$$
I_3 \leq \mathcal{O}\left(L^4\left(\frac{(t-1)^3}{\rho\sqrt{m}}\log m\right)^{4/3}\right) + \mathcal{O}\left(L\left(\frac{(t-1)^3}{\rho\sqrt{m}}\log m\right)\right) + \mathcal{O}(L\beta_t).
\tag{18}
$$

. For $I_4$, we have

$$
\begin{aligned}
I_4 &= |\langle g(\mathbf{x}_t; \Theta_0), \Theta_t - \Theta_0\rangle - \langle g(\mathbf{x}_t; \Theta_t), \Theta_t - \Theta_0\rangle| \\
&\leq \|g(\mathbf{x}_t; \Theta_t) - g(\mathbf{x}_t; \Theta_0)\|_2 \|\Theta_t - \Theta_0\|_2 \\
&\leq \beta_t \cdot \|g(\mathbf{x}_t; \Theta_t) - g(\mathbf{x}_t; \Theta_0)\|_2
\end{aligned}
\tag{19}
$$

where the first inequality is because of Cauchy–Schwarz inequality and the last inequality is by Lemma D.1. For $I_5$, we have

$$
I_5 = |\langle g(\mathbf{x}_t; \Theta_t), \Theta_t - \Theta_0\rangle + f(\mathbf{x}_t; \Theta_0) - f(\mathbf{x}_t; \Theta_t)| \leq (L+1)^2 \sqrt{m\log m}\beta_t^{4/3}
\tag{20}
$$

where this inequality uses Lemma D.2 with Lemma D.1.

Combing Eq.(13), (18), (19), and (20) completes the proof. $\qquad\square$

**Lemma C.2.** *For any $\delta \in (0,1), \rho \in (0, \mathcal{O}(\frac{1}{L})], 0 < \epsilon_1 \leq \epsilon_2 \leq 1, \lambda > 0$, suppose $m, \eta_1, \eta_2, J_1, J_2$ satisfy the conditions in Eq.(6). Then, with probability at least $1 - \delta$ over the random initialization,*

*after $t$ rounds, the error induced by meta-learner is upper bounded by:*

$$\sum_{t=1}^{T} \mathbb{E}_{r_t|\mathbf{x}_t} [|f(\mathbf{x}_t; \Theta_t) - r_t| \mid u_t]$$
$$\leq \sum_{t=1}^{T} \frac{\mathcal{O}\left(\|g(\mathbf{x}_t; \Theta_t) - g(\mathbf{x}_t; \theta_0^{u_t})\|_2\right)}{\sqrt{t}} + \sum_{u \in N} \mu_t^u \left[\mathcal{O}\left(\frac{L+1}{\sqrt{2\mu_t^u}}\right) + \sqrt{\frac{2\log(t/\delta)}{\mu_t^u}}\right]. \tag{21}$$

*where the expectation is taken over $r_t$ conditioned on $x_t$.*

*Proof.*

$$\sum_{t=1}^{T} \mathbb{E}_{r_t|\mathbf{x}_t} [|f(\mathbf{x}_t; \Theta_t) - r_t||u_t]$$
$$= \sum_{t=1}^{T} \mathbb{E}_{r_t|\mathbf{x}_t} [|f(\mathbf{x}_t; \Theta_t) - f(\mathbf{x}_t; \theta_{t-1}^{u_t}) + f(\mathbf{x}_t; \theta_{t-1}^{u_t}) - r_t| \mid u_t] \tag{22}$$
$$\leq \underbrace{\sum_{t=1}^{T} |f(\mathbf{x}_t; \Theta_t) - f(\mathbf{x}_t; \theta_{t-1}^{u_t})|}_{I_1} + \underbrace{\sum_{t=1}^{T} \mathbb{E}_{r_t|\mathbf{x}_t} [|f(\mathbf{x}_t; \theta_{t-1}^{u_t}) - r_t| \mid u_t]}_{I_2}$$

.

For $I_1$, applying Lemma C.1, with probability at least $1 - \delta$, for any $\|\mathbf{x}_{t,j}\|_2 = 1$, we have

$$I_1 \leq \sum_{t=1}^{T} (\beta_t \cdot \|g(\mathbf{x}_t; \Theta_t) - g(\mathbf{x}_t; \theta_0^u)\|_2 + Z_t) \overset{E_1}{\leq} \sum_{t=1}^{T} \frac{\mathcal{O}\left(\|g(\mathbf{x}_t; \Theta_t) - g(\mathbf{x}_t; \theta_0^{u_t})\|_2\right)}{\sqrt{t}} \tag{23}$$

where $E_1$ is the result of choice of $m$ ($m \geq \widetilde{\Omega}(T^{27})$) for $\beta_t$ and $Z_t$.

For $I_2$, based on the Lemma E.1, with probability at least $1 - \delta$, for any $\epsilon_1 \in (0, 1]$, we have

$$I_2 \leq \sum_{u \in N} \left[\sqrt{\epsilon_1 \mu_t^u} + \mathcal{O}\left(L\sqrt{\mu_t^u}\right) + (1 + \xi_t)\sqrt{2\mu_t^u \log(t/\delta)}\right]$$
$$\leq \sum_{u \in N} \mu_t^u \left[\mathcal{O}\left(\frac{L+1}{\sqrt{2\mu_t^u}}\right) + \sqrt{\frac{2\log(t/\delta)}{\mu_t^u}}\right]. \tag{24}$$

The proof is complete. $\qquad\qquad\qquad\qquad\qquad\qquad\qquad\qquad\qquad\qquad\qquad\qquad\qquad\qquad\square$

## D   ANALYSIS FOR META-LEARNER

**Lemma D.1.** *Given any $\delta \in (0, 1)$, $0 < \epsilon_1 \leq \epsilon_2 \leq 1, \lambda > 0$, $\rho \in (0, \mathcal{O}(\frac{1}{L})]$, suppose $m, \eta_1, \eta_2, J_1, J_2$ satisfy the conditions in Eq.(6) and $\Theta_0, \theta_0^u$ are randomly initialized ,$\forall u \in N$. Then, with probability at least $1 - \delta$, these hold for Algorithms 1-3:*

1. *Given any $\mathcal{N} \subseteq N$, define $\mathcal{L}_{\mathcal{N}}(\Theta_{t,i}) = \frac{1}{2} \sum_{\substack{u \in \mathcal{N} \\ (\mathbf{x},r) \in \mathcal{T}_{t-1}^u}} (f(\mathbf{x}; \Theta_{t,i}) - r)^2$, where $\Theta_{t,i}$ is re-turned by Algorithm 2 given $\mathcal{N}$. Then, we have $\mathcal{L}_{\mathcal{N}}(\Theta_{t,i}) \leq \epsilon_2$ in $J_2$ rounds.*

2. *For any $j \in [J_2]$, $\|\Theta_{(j)} - \Theta_{(0)}\|_2 \leq \frac{\mathcal{O}(n^2 t^3 \sqrt{\epsilon_2 \log^2 m}) + \mathcal{O}(t^2 \log^2 m - t\epsilon_2)\rho^{1/2}\lambda n}{\mathcal{O}(\rho \sqrt{m}\epsilon_2)} = \beta_1$.*

*Proof.* Define the *sign matrix*

$$\text{sign}(\theta_{[i]}) = \begin{cases} 1 & \text{if } \theta_{[i]} \geq 0; \\ -1 & \text{if } \theta_{[i]} < 0 \end{cases} \tag{25}$$

where $\theta_{[i]}$ is the $i$-th element in $\theta$.

For the brevity, we use $\widehat{\theta}_t^u$ to denote $\widehat{\theta}_{\mu_t^u}^u$, For each $u \in \mathcal{N}$, we have $\mathcal{T}_{t-1}^u$. Given a group $\mathcal{N}$, then recall that

$$\mathcal{L}_{\mathcal{N}} = \sum_{u \in \mathcal{N}} \mathcal{L}\left(\widehat{\theta}_t^u\right) + \frac{\lambda}{\sqrt{m}} \sum_{u \in \mathcal{N}} \|\widehat{\theta}_t^u\|_1.$$

Then, in round $t + 1$, for any $j \in [J_2]$ we have

$$\Theta_{(j)} - \Theta_{(j-1)} = \eta_2 \cdot \nabla_{\{\widehat{\theta}_t^u\}_{u \in \mathcal{N}}} \mathcal{L}_{\mathcal{N}}$$

$$= \eta_2 \cdot \left( \sum_{n \in \mathcal{N}} \nabla_{\widehat{\theta}_t^u} \mathcal{L} + \frac{\lambda}{\sqrt{m}} \sum_{u \in \mathcal{N}} \text{sign}(\widehat{\theta}_t^u) \right) \tag{26}$$

According to Theorem 4 in (Allen-Zhu et al., 2019), given $\Theta_{(j)}, \Theta_{(j-1)}$, we have

$$\mathcal{L}_{\mathcal{N}}(\Theta_{(j)}) \leq \mathcal{L}_{\mathcal{N}}(\Theta_{(j-1)}) - \langle \nabla_{\Theta_{(j-1)}} \mathcal{L}_{\mathcal{N}}, \Theta_{(j)} - \Theta_{(j-1)} \rangle$$
$$+ \sqrt{t \mathcal{L}_{\mathcal{N}}(\Theta_{(j-1)})} \cdot w^{1/3} L^2 \sqrt{m \log m} \cdot \mathcal{O}(\|\Theta_{(j)} - \Theta_{(j-1)}\|_2) + \mathcal{O}(t L^2 m) \|\Theta_{(j)} - \Theta_{(j-1)}\|_2^2$$

$$\underbrace{\leq}_{E_1} \mathcal{L}_{\mathcal{N}}(\Theta_{(j-1)}) - \eta_2 \| \sum_{n \in \mathcal{N}} \nabla_{\widehat{\theta}_t^u} \mathcal{L} + \frac{\lambda}{\sqrt{m}} \sum_{u \in \mathcal{N}} \text{sign}(\widehat{\theta}_t^u) \|_2 \|\nabla_{\Theta_{(j-1)}} \mathcal{L}_{\mathcal{N}}\|_2 +$$
$$+ \eta_2 w^{1/3} L^2 \sqrt{t m \log m} \| \sum_{n \in \mathcal{N}} \nabla_{\widehat{\theta}_t^u} \mathcal{L} + \frac{\lambda}{\sqrt{m}} \sum_{u \in \mathcal{N}} \text{sign}(\widehat{\theta}_t^u) \|_2 \sqrt{\mathcal{L}_{\mathcal{N}}(\Theta_{(j-1)})}$$
$$+ \eta_2^2 \mathcal{O}(t L^2 m) \| \sum_{n \in \mathcal{N}} \nabla_{\widehat{\theta}_t^u} \mathcal{L} + \frac{\lambda}{\sqrt{m}} \sum_{u \in \mathcal{N}} \text{sign}(\widehat{\theta}_t^u) \|_2^2 \tag{27}$$

$$\Rightarrow \mathcal{L}_{\mathcal{N}}(\Theta_{(j)}) \leq \mathcal{L}_{\mathcal{N}}(\Theta_{(j-1)}) - \eta_2 \sqrt{n} \sum_{u \in \mathcal{N}} \|\nabla_{\widehat{\theta}_t^u} \mathcal{L}\|_2 \|\nabla_{\Theta_{(j-1)}} \mathcal{L}_{\mathcal{N}}\|_2 +$$
$$+ \eta_2 w^{1/3} L^2 \sqrt{t n m \log m} \sum_{n \in \mathcal{N}} \|\nabla_{\widehat{\theta}_t^u} \mathcal{L}\|_2 \sqrt{\mathcal{L}_{\mathcal{N}}(\Theta_{(j-1)})} + \eta_2^2 \mathcal{O}(t L^2 m) n \sum_{n \in \mathcal{N}} \|\nabla_{\widehat{\theta}_t^u} \mathcal{L}\|_2^2$$
$$- \frac{\eta_2 \lambda}{\sqrt{m}} \|\nabla_{\Theta_{(j-1)}} \mathcal{L}_{\mathcal{N}}\|_2 + \eta_2 w^{1/3} n L^2 \sqrt{t \log m} \lambda \sqrt{\mathcal{L}_{\mathcal{N}}(\Theta_{(j-1)})} + \mathcal{O}(2 \eta_2^2 t L^2) \lambda^2 n^2 \tag{28}$$

$$\Rightarrow \mathcal{L}_{\mathcal{N}}(\Theta_{(j)}) \underbrace{\leq}_{E_2} \mathcal{L}_{\mathcal{N}}(\Theta_{(j-1)}) \underbrace{-\eta_2 \sqrt{n} \sum_{u \in \mathcal{N}} \frac{\rho m}{t \mu_t^u} \sqrt{\mathcal{L}(\widehat{\theta}_t^u) \mathcal{L}_{\mathcal{N}}(\Theta_{(j-1)})} +}_{I_1}$$
$$\underbrace{+ \eta_2 w^{1/3} L^2 m \sqrt{t \rho n \log m} \sum_{n \in \mathcal{N}} \sqrt{\mathcal{L}(\widehat{\theta}_t^u) \mathcal{L}_{\mathcal{N}}(\Theta_{(j-1)})} + \eta_2^2 t^2 L^2 m^2 n \sum_{n \in \mathcal{N}} \mathcal{L}(\widehat{\theta}_t^u)}_{I_1}$$
$$\underbrace{- \frac{\eta_2 \lambda \sqrt{\rho}}{t} \sqrt{\mathcal{L}_{\mathcal{N}}(\Theta_{(j-1)})} + \eta_2 w^{1/3} n L^2 \sqrt{t \log m} \lambda \sqrt{\mathcal{L}_{\mathcal{N}}(\Theta_{(j-1)})} + \mathcal{O}(2 \eta_2^2 t L^2) \lambda^2 n^2}_{I_2} \tag{29}$$

where $E_1$ is because of Cauchy–Schwarz inequality inequality, $E_2$ is due to Theorem 3 in (Allen-Zhu et al., 2019), i.e., the gradient lower bound. Recall that

$$\eta_2 = \min \left\{ \Theta\left( \frac{\sqrt{n}\rho}{t^4 L^2 m} \right), \Theta\left( \frac{\sqrt{\rho \epsilon_2}}{t^2 L^2 \lambda n^2} \right) \right\}, \quad L_{\mathcal{N}}(\Theta_0) \leq \mathcal{O}(t \log^2 m)$$

$$J_2 = \max \left\{ \Theta\left( \frac{t^5 (\mathcal{O}(t \log^2 m) - \epsilon_2) L^2 m}{\sqrt{n \epsilon_2 \rho}} \right), \Theta\left( \frac{t^3 L^2 \lambda n^2 (\mathcal{O}(t \log^2 m - \epsilon_2))}{\rho \epsilon_2} \right) \right\}. \tag{30}$$

Before achieving $\mathcal{L}_{\mathcal{N}}(\Theta_{(j)}) \leq \epsilon_2$, we have, for each $u \in \mathcal{N}$, $\mathcal{L}(\widehat{\theta}_t^u) \leq \mathcal{L}_{\mathcal{N}}(\Theta_{(j-1)})$, for $I_1$, we have

$$
\begin{aligned}
I_1 \quad \leq & - \eta_2 \sqrt{n} \sum_{u \in \mathcal{N}} \frac{\rho m}{t \mu_t^u} \sqrt{\mathcal{L}(\widehat{\theta}_t^u) \mathcal{L}_{\mathcal{N}}(\Theta_{(j-1)})} + \\
& + \eta_2 w^{1/3} L^2 m \sqrt{t \rho n \log m} \sum_{n \in \mathcal{N}} \sqrt{\mathcal{L}(\widehat{\theta}_t^u) \mathcal{L}_{\mathcal{N}}(\Theta_{(j-1)})} + \eta_2^2 t^2 L^2 m^2 n \sum_{n \in \mathcal{N}} \sqrt{\mathcal{L}(\widehat{\theta}_t^u) \mathcal{L}_{\mathcal{N}}(\Theta_{(j-1)})} \\
\leq & - \frac{\eta_2 n \sqrt{n} \rho m}{t^2} \sum_{n \in \mathcal{N}} \sqrt{\mathcal{L}(\widehat{\theta}_t^u) \mathcal{L}_{\mathcal{N}}(\Theta_{(j-1)})} \\
& + \left( \eta_2 w^{1/3} L^2 m \sqrt{t \rho n \log m} + \eta_2^2 t^2 L^2 m^2 n \right) \sum_{n \in \mathcal{N}} \sqrt{\mathcal{L}(\widehat{\theta}_t^u) \mathcal{L}_{\mathcal{N}}(\Theta_{(j-1)})} \\
\underbrace{\leq}_{E_3} & - \boldsymbol{\Theta}\left( \frac{\eta_2 n \sqrt{n} \rho m}{t^2} \right) \sum_{n \in \mathcal{N}} \sqrt{\mathcal{L}(\widehat{\theta}_t^u) \mathcal{L}_{\mathcal{N}}(\Theta_{(j-1)})} \\
\underbrace{\leq}_{E_4} & - \boldsymbol{\Theta}\left( \frac{\eta_2 n \sqrt{n} \rho m}{t^2} \right) \sum_{n \in \mathcal{N}} \mathcal{L}(\widehat{\theta}_t^u)
\end{aligned}
\tag{31}
$$

where $E_3$ is because of the choice of $\eta_2$. As $\mathcal{L}_{\mathcal{N}}(\Theta_0) \leq \mathcal{O}(t \log^2 m)$, we have $\mathcal{L}_{\mathcal{N}}(\Theta_{(j)}) \leq \epsilon_2$ in $J_\Theta$ rounds. For $I_2$, we have

$$
\begin{aligned}
I_2 \underbrace{\leq}_{E_5} & - \frac{\eta_2 \lambda \sqrt{\rho}}{t} \sqrt{\epsilon_2} + \eta_2 w^{1/3} n L^2 \sqrt{t \log m} \lambda \sqrt{\mathcal{L}_{\mathcal{N}}(\Theta_{(0)})} + \mathcal{O}(2 \eta_2^2 t L^2) \lambda^2 n^2 \\
\underbrace{\leq}_{E_6} & - \frac{\eta_2 \lambda \sqrt{\rho}}{t} \sqrt{\epsilon_2} + \eta_2 w^{1/3} n L^2 \sqrt{t \log m} \lambda \sqrt{\mathcal{O}(t \log^2 m)} + \mathcal{O}(2 \eta_2^2 t L^2) \lambda^2 n^2 \\
\leq & \left( - \frac{\eta_2 \sqrt{\rho}}{t} \sqrt{\epsilon_2} + \eta_2 w^{1/3} n L^2 \sqrt{t \log m} \sqrt{\mathcal{O}(t \log^2 m)} + \mathcal{O}(2 \eta_2^2 t L^2) \lambda n^2 \right) \lambda \\
\underbrace{\leq}_{E_7} & - \boldsymbol{\Theta}(\frac{\eta_2 \sqrt{\rho \epsilon_2}}{t}) \lambda
\end{aligned}
\tag{32}
$$

where $E_5$ is by $\mathcal{L}_{\mathcal{N}}(\Theta_{(j-1)}) \geq \epsilon_2$ and $\mathcal{L}_{\mathcal{N}}(\Theta_{(j-1)}) \leq \mathcal{L}_{\mathcal{N}}(\Theta_{(0)})$, $E_6$ is according to Eq.(30), and $E_7$ is because of the choice of $\eta_2$.

Combining above inequalities together, we have

$$
\begin{aligned}
\mathcal{L}_{\mathcal{N}}(\Theta_{(j)}) \leq & \mathcal{L}_{\mathcal{N}}(\Theta_{(j-1)}) - \boldsymbol{\Theta}\left( \frac{\eta_2 n \sqrt{n} \rho m}{t^2} \right) \sum_{n \in \mathcal{N}} \mathcal{L}(\widehat{\theta}_t^u) - \boldsymbol{\Theta}(\frac{\eta_2 \sqrt{\rho \epsilon_2}}{t}) \lambda \\
\leq & \mathcal{L}_{\mathcal{N}}(\Theta_{(j-1)}) - \boldsymbol{\Theta}(\frac{\eta_2 \sqrt{\rho \epsilon_2}}{t}) \lambda
\end{aligned}
\tag{33}
$$

Thus, because of the choice of $J_2, \eta_2$, we have

$$
\begin{aligned}
\mathcal{L}_{\mathcal{N}}(\Theta_{(J_2)}) \leq & \mathcal{L}_{\mathcal{N}}(\Theta_{(0)}) - J_2 \cdot \boldsymbol{\Theta}(\frac{\eta_2 \sqrt{\rho \epsilon_2}}{t}) \lambda \\
\leq & \mathcal{O}(t \log^2 m) - J_2 \cdot \boldsymbol{\Theta}(\frac{\eta_2 \sqrt{\rho \epsilon_2}}{t}) \leq \epsilon_2.
\end{aligned}
\tag{34}
$$

The proof of (1) is completed.

According to Lemma E.8, For any $j \in [J_1]$, $\mathcal{L}(\theta_{(j)}^u) \leq (1 - \Omega(\frac{\eta \rho m}{d \mu_t^{u2}})) \mathcal{L}(\theta_{(j-1)}^u)$. Therefore, for any $u \in [n]$, we have

$$
\begin{aligned}
\sqrt{\mathcal{L}(\widehat{\theta}_t^u)} \leq & \sum_{j=0}^{J_1} \sqrt{\mathcal{L}(\theta_{(j)}^u)} \leq \mathcal{O}\left( \frac{(\mu_t^u)^2}{\eta_1 \rho m} \right) \cdot \sqrt{\mathcal{L}(\theta_{(0)}^u)} \\
\leq & \mathcal{O}\left( \frac{(\mu_t^u)^2}{\eta_1 \rho m} \right) \cdot \mathcal{O}(\sqrt{\mu_t^u \log^2 m}),
\end{aligned}
\tag{35}
$$

where the last inequality is because of Lemma E.8 (3).

Second, we have

$$
\begin{aligned}
\|\Theta_{(J_2)} - \Theta_0\|_2 &\leq \sum_{j=1}^{J_2} \|\Theta_{(j)} - \Theta_{(j-1)}\|_2 \\
&\leq \sum_{j=1}^{J_2} \eta_2 \| \sum_{n \in \mathcal{N}} \nabla_{\widehat{\theta}_t^u} \mathcal{L} + \frac{\lambda}{\sqrt{m}} \sum_{u \in \mathcal{N}} \operatorname{sign}(\widehat{\theta}_t^u)\|_2 \\
&\leq \underbrace{\sum_{j=1}^{J_2} \eta_2 \| \sum_{u \in \mathcal{N}} \nabla_{\widehat{\theta}_t^u} \mathcal{L}\|_F}_{I_3} + \frac{J_2 \eta_2 \lambda n}{\sqrt{m}}
\end{aligned}
\tag{36}
$$

For $I_3$, we have

$$
\begin{aligned}
\sum_{j=1}^{J_2} \eta_2 \| \sum_{u \in \mathcal{N}} \nabla_{\widehat{\theta}_t^u} \mathcal{L}\|_2 &\leq \sum_{j=1}^{J_2} \eta_2 \sqrt{|\mathcal{N}|} \sum_{u \in \mathcal{N}} \|\nabla_{\widehat{\theta}_t^u} \mathcal{L}\|_2 \\
&\underset{E_8}{\leq} \sum_{j=1}^{J_2} \eta_2 \sqrt{n} \sum_{u \in N} \|\nabla_{\widehat{\theta}_t^u} \mathcal{L}\|_2 \\
&\underset{E_9}{\leq} \mathcal{O} \sum_{j=1}^{J_2} (\eta_2) \sqrt{ntm} \sum_{u \in N} \sqrt{\mathcal{L}(\widehat{\theta}_t^u)}
\end{aligned}
\tag{37}
$$

$$
\begin{aligned}
\Rightarrow \sum_{j=1}^{J_2} \eta_2 \| \sum_{u \in \mathcal{N}} \nabla_{\widehat{\theta}_t^u} \mathcal{L}\|_2 &\underset{E_{10}}{\leq} \mathcal{O}(\eta_2) \sqrt{ntm} \sum_{u \in N} \sum_{j=1}^{J_2} \sqrt{\mathcal{L}(\widehat{\theta}_t^u)} \\
&\underset{E_{11}}{\leq} \mathcal{O}(\eta_2) \sqrt{ntm} \cdot n \cdot \mathcal{O}\left(\frac{(\mu_t^u)^2}{\eta_1 \rho m}\right) \cdot \mathcal{O}(\sqrt{\mu_t^u \log^2 m}) \\
&\leq \mathcal{O}\left(\frac{\eta_2 n^{3/2} t^{5/2} \sqrt{t \log^2 m}}{\eta_1 \rho \sqrt{m}}\right)
\end{aligned}
\tag{38}
$$

where $E_1$ is because of $|\mathcal{N}| \leq n$, $E_2$ is due to Theorem 3 in (Allen-Zhu et al., 2019), and $E_3$ is as the result of Eq.(35).

Combining Eq.(36) and Eq.(38), we have

$$
\begin{aligned}
\|\Theta_{(J_2)} - \Theta_0\|_2 &\leq \mathcal{O}\left(\frac{\eta_2 n^{3/2} t^3 \sqrt{\log^2 m} + J_2 \eta_2 \eta_1 \rho \lambda n}{\eta_1 \rho \sqrt{m}}\right) \\
&\leq \mathcal{O}\left(\frac{\eta_2 n^{3/2} t^3 \sqrt{\log^2 m} + \mathcal{O}(t^2 \log^2 m - t\epsilon_2))\eta_1 \sqrt{\rho} \lambda n}{\eta_1 \rho \sqrt{m} \epsilon_2}\right) \\
&\leq \frac{\mathcal{O}(n^2 t^3 \sqrt{\epsilon_2 \log^2 m}) + \mathcal{O}(t^2 \log^2 m - t\epsilon_2)\rho^{1/2} \lambda n}{\mathcal{O}(\rho \sqrt{m} \epsilon_2)} \\
&= \beta_t.
\end{aligned}
\tag{39}
$$

The proof is completed. $\square$

## D.1 ANCILLARY LEMMAS

**Lemma D.2** ((Wang et al., 2020a)). *Suppose $m$ satisfies the condition2 in Eq.(6), if*

$$
\Omega(m^{-3/2} L^{-3/2} [\log(TkL^2/\delta)]^{3/2}) \leq \nu \leq \mathcal{O}((L+1)^{-6} \sqrt{m}).
$$

*then with probability at least $1 - \delta$, for all $\Theta, \Theta'$ satisfying $\|\Theta - \Theta_0\|_2 \leq \nu$ and $\|\Theta' - \Theta_0\|_2 \leq \nu$, $\mathbf{x} \in \mathbb{R}^d, \|\mathbf{x}\|_2 = 1$, we have*

$$|f(\mathbf{x}; \Theta) - f(\mathbf{x}; \Theta') - \langle \nabla_\Theta f(\mathbf{x}; \Theta), \Theta' - \Theta \rangle| \leq \mathcal{O}(\nu^{4/3}(L+1)^2 \sqrt{m \log m}).$$

**Lemma D.3.** *With probability at least $1 - \delta$, set $\eta_2 = \Theta(\frac{\nu}{\sqrt{2tm}})$, for any $\Theta' \in \mathbb{R}^p$ satisfying $\|\Theta' - \Theta_0\|_2 \leq \beta_1$, such that*

$$\sum_{\tau=1}^{t} |f(\mathbf{x}_\tau; \Theta_{(j)}) - r_\tau| \leq \sum_{\tau=1}^{t} |f(\mathbf{x}_\tau; \Theta') - r_\tau| + \mathcal{O}\left(\frac{3L\sqrt{t}}{\sqrt{2}}\right)$$

*Proof.* Then, the proof is a direct application of Lemma 4.3 in (Cao and Gu, 2019) by setting the loss as $L_\tau(\widehat{\Theta}_\tau) = |f(\mathbf{x}_\tau; \widehat{\Theta}_\tau) - r_\tau|$, $R = \beta_1 \sqrt{m}$, $\epsilon = \frac{LR}{\sqrt{2\nu t}}$, and $\nu = R^2$.

$\square$

# E ANALYSIS FOR USER-LEARNER

**Lemma E.1.** *For any $\delta \in (0, 1), 0 < \rho \leq \mathcal{O}(\frac{1}{L})$, suppose $0 < \epsilon_1 \leq 1$ and $m, \eta_1, J_1$ satisfy the conditions in Eq.(6). After $T$ rounds, with probability $1 - \delta$ over the random initialization, the cumulative error induced by the user-learners is upper bounded by*

$$\frac{1}{T} \sum_{t=1}^{T} \mathbb{E}_{r_t|\mathbf{x}_t} [|f(\mathbf{x}_t; \theta_{t-1}^{u_t}) - r_t| \mid \mathcal{T}_{t-1}^{u_t}, u_t]$$

$$\leq \sqrt{n} \left[ \sqrt{\frac{\epsilon_1}{T}} + \mathcal{O}\left(\frac{LR}{\sqrt{T}}\right) + \mathcal{O}(1 + \xi_t)\sqrt{\frac{2\log(T/\delta)}{T}} \right],$$

*where the expectation is taken over $r_t^u$ conditioned on $\mathbf{x}_t^u$ and $\mathcal{T}_t^u$ is the historical data of $u$ up to round $t$.*

*Proof.* Applying Lemma E.3 over all users, we have

$$\frac{1}{T} \sum_{t=1}^{T} \mathbb{E}_{r_t|\mathbf{x}_t} [|f(\mathbf{x}_t; \theta_{t-1}^{u_t}) - r_t| \mid \mathcal{T}_{t-1}^{u_t}, u_t]$$

$$= \frac{1}{T} \sum_{u \in N} \sum_{(\mathbf{x}_\tau, r_\tau) \in \mathcal{T}_t^u} \mathbb{E}_{r_t|\mathbf{x}_t} [|f(\mathbf{x}_\tau; \theta_{t-1}^u) - r_\tau| \mid \mathcal{T}_{t-1}^u, u] \tag{40}$$

$$\leq \frac{1}{T} \sum_{u \in N} \left[ \sqrt{\epsilon_1 \mu_T^u} + \mathcal{O}\left(L\sqrt{\mu_T^u}\right) + (1 + \xi_t)\sqrt{2\mu_T^u \log(T/\delta)} \right]$$

where we applied the union bound to $\delta$ over all $n$ users and so we get $\log(T/\delta)$ because of $\sum_{u \in N} \mu_T^u = T$. Then, given a user $u$, then, $\mu_T^u = \sum_{t=1}^{T} \mathbb{1}\{u_t = u\}$ where $\mathbb{1}\{u_t = u\}$ is the indicator function. Then, applying Hoeffding-Azuma inequality on the sequence $\sqrt{\mu_T^u}, \forall u \in N$, we have

$$\sum_{u \in N} \sqrt{\mu_T^u} \leq \sum_{u \in N} \mathbb{E}[\sqrt{\mu_T^u}] + \sqrt{2n \log(1/\delta)}$$

$$= \sqrt{nT} + \sqrt{2n \log(1/\delta)}.$$

Then, by simplification, we have

$$\frac{1}{T} \sum_{t=1}^{T} \mathbb{E}_{r_t|\mathbf{x}_t} [|f(\mathbf{x}_t; \theta_{t-1}^{u_t}) - r_t| \mid \mathcal{T}_{t-1}^{u_t}, u_t]$$

$$\leq \sqrt{n} \left[ \sqrt{\frac{\epsilon_1}{T}} + \mathcal{O}\left(\frac{L}{\sqrt{T}}\right) + \mathcal{O}(1 + \xi_t)\sqrt{\frac{2\log(T/\delta)}{T}} \right]. \tag{41}$$

The proof is complete.

$\square$

**Lemma E.2.** *For any $\delta \in (0,1), 0 < \rho \leq \mathcal{O}(\frac{1}{L})$, suppose $0 < \epsilon_1 \leq 1$ and $m, \eta_1, J_1$ satisfy the conditions in Eq.(6). In round $t \in [T]$, given $u \in N$, let*

$$x_t^* = \arg \max_{\mathbf{x}_{t,i}, i \in [k]} h_u(\mathbf{x}_{t,i})$$

*the Bayes-optimal arm for $u$ and $r_t^*$ is the corresponding reward. Then, with probability at least $1 - \delta$ over the random initialization, after $T$ rounds, with probability $1 - \delta$ over the random initialization, the cumulative error induced by the user-learners is upper bounded by:*

$$\frac{1}{T} \sum_{t=1}^{T} \mathbb{E}_{r_t^*|\mathbf{x}_t^*} [|f(\mathbf{x}_t^*; \theta_{t-1}^{u_t,*}) - r_t^*| \mid \mathcal{T}_{t-1}^{u_t,*}, u_t]$$

$$\leq \sqrt{n} \left[ \sqrt{\frac{\epsilon_1}{T}} + \mathcal{O}\left(\frac{L}{\sqrt{T}}\right) + \mathcal{O}(1 + \xi_t) \sqrt{\frac{2 \log(T/\delta)}{T}} \right].$$

*where the expectation is taken over $r_\tau^*$ conditioned on $\mathbf{x}_\tau^*$, $\mathcal{T}_t^{u,*} = \{(\mathbf{x}_\tau^*, r_\tau^*) : u_\tau = u, \tau \in [t]\}$ are stored Bayes-optimal pairs up to round $t$ for $u$, and $\theta_{t-1}^{u_t,*}$ are the parameters trained on $\mathcal{T}_{t-1}^{u_t,*}$ according to Algorithm 3 in round $t - 1$.*

*Proof.* Based on Lemma E.4, we have

$$\frac{1}{T} \sum_{t=1}^{T} \mathbb{E}_{r_t^*|\mathbf{x}_t^*} [|f(\mathbf{x}_t^*; \theta_{t-1}^{u_t,*}) - r_t^*| \mid \mathcal{T}_{t-1}^{u_t,*}, u_t]$$

$$= \frac{1}{T} \sum_{u \in N} \sum_{(\mathbf{x}_\tau^*, r_\tau^*) \in \mathcal{T}_t^{u,*}} \mathbb{E}_{r_t^*|\mathbf{x}_t^*} [|f(\mathbf{x}_\tau^*; \theta_{t-1}^{u,*}) - r_\tau^*| \mid \mathcal{T}_{t-1}^{u,*}, u] \quad (42)$$

$$\leq \frac{1}{T} \sum_{u \in N} \left[ \sqrt{\epsilon_1 \mu_T^u} + \mathcal{O}\left(L\sqrt{\mu_t^u}\right) + (1 + \xi_t)\sqrt{2\mu_t^u \log(T/\delta)} \right]$$

where we applied the union bound to $\delta$ over all $n$ users and so we get $\log(T/\delta)$ because of $\sum_{u \in N} \mu_T^u = T$. Then, given a user $u$, then, $\mu_T^u = \sum_{t=1}^{T} \mathbb{1}\{u_t = u\}$ where $\mathbb{1}\{u_t = u\}$ is the indicator function. Then, applying Hoeffding-Azuma inequality on the sequence $\sqrt{\mu_T^u}, \forall u \in N$, we have

$$\sum_{u \in N} \sqrt{\mu_T^u} \leq \sum_{u \in N} \mathbb{E}[\sqrt{\mu_T^u}] + \sqrt{2n \log(1/\delta)}$$

$$= \sqrt{nT} + \sqrt{2n \log(1/\delta)}.$$

Then, we have

$$\frac{1}{T} \sum_{t=1}^{T} \mathbb{E}_{r_t^*|\mathbf{x}_t^*} [|f(\mathbf{x}_t; \theta_{t-1}^{u_t,*}) - r_t^*| \mid \mathcal{T}_{t-1}^{u_t,*}, u_t]$$

$$\leq \sqrt{n} \left[ \sqrt{\frac{\epsilon_1}{T}} + \mathcal{O}\left(\frac{L}{\sqrt{T}}\right) + \mathcal{O}(1 + \xi_t) \sqrt{\frac{2 \log(T/\delta)}{T}} \right]. \quad (43)$$

The proof is complete. $\qquad \square$

**Lemma E.3.** *For any $\delta \in (0,1), 0 < \rho \leq \mathcal{O}(\frac{1}{L})$, suppose $0 < \epsilon_1 \leq 1$ and $m, \eta_1, J_1$ satisfy the conditions in Eq.(6). In a round $\tau$ where $u \in N$ is serving user, let $x_\tau$ be the arm selected by some fixed policy $\pi_\tau$ and $r_\tau$ is the corresponding received reward. Then, with probability at least $1 - \delta$ over the randomness of initialization, after $t \in [T]$ rounds, the cumulative regret induced by $u$ is upper bounded by:*

$$\frac{1}{\mu_t^u} \sum_{(\mathbf{x}_\tau, r_\tau) \in \mathcal{T}_t^u} \mathbb{E}_{r_\tau|\mathbf{x}_\tau} [|f(\mathbf{x}_\tau; \theta_{\tau-1}^u) - r_\tau| \mid \pi_\tau, u]$$

$$\leq \sqrt{\frac{\epsilon_1}{\mu_t^u}} + \mathcal{O}\left(\frac{3L}{\sqrt{2\mu_t^u}}\right) + (1 + \xi_t)\sqrt{\frac{2 \log(\mu_t^u/\delta)}{\mu_t^u}}.$$

*where the expectation is taken over $r_\tau$ conditioned on $\mathbf{x}_\tau$ and $\mathcal{T}_t^u = \{(\mathbf{x}_\tau, r_\tau) : u_\tau = u, \tau \in [t]\}$ is the historical data of $u$ up to round $t$.*

*Proof.* According to Lemma E.5, with probability at least $1 - \delta$, given any $\|\mathbf{x}\|_2 = 1, r \leq 1$, for any round $\tau$ in which $u$ is the serving user, we have

$$|f(\mathbf{x}; \theta^u_{\tau-1}) - r| \leq \xi_t + 1.$$

Here, we will apply the union bound of $\delta$ over all $\mu^u_T$ rounds, to make this bound hold for every round of $u$. Then, in a round $\tau$ where $u$ is the serving user, let $\mathbf{x}_\tau$ be the arm selected by some fixed policy $\pi_\tau$ and $r_\tau$ is the corresponding reward. Then, we define

$$V_\tau = \mathop{\mathbb{E}}_{r_\tau | \mathbf{x}_\tau} [|f(\mathbf{x}_\tau; \theta^u_{\tau-1}) - r_\tau|] - |f(\mathbf{x}_\tau; \theta^u_{\tau-1}) - r_\tau|, \tag{44}$$

where the expectation is taken over $r_\tau$ conditioned on $\mathbf{x}_\tau$. Then, we have

$$\mathbb{E}[V_\tau | \mathbf{F}^u_\tau] = \mathop{\mathbb{E}}_{r_\tau | \mathbf{x}_\tau} [|f(\mathbf{x}_\tau; \theta^u_{\tau-1}) - r_\tau|] - \mathbb{E}[|f(\mathbf{x}_\tau; \theta^u_{\tau-1}) - r_\tau| \mid \mathbf{F}^u_\tau] = 0$$

where $\mathbf{F}^u_\tau$ denotes the $\sigma$-algebra generated by $\mathcal{T}^u_{\tau-1}$. Thus, we have the following form:

$$\frac{1}{\mu^u_t} \sum_{(\mathbf{x}_\tau, r_\tau) \in \mathcal{T}^u_t} V_\tau = \frac{1}{\mu^u_t} \sum_{(\mathbf{x}_\tau, r_\tau) \in \mathcal{T}^u_t} \mathop{\mathbb{E}}_{r_\tau | \mathbf{x}_\tau} [|f(\mathbf{x}_\tau; \theta^u_{\tau-1}) - r_\tau|] - \frac{1}{\mu^u_t} \sum_{(\mathbf{x}_\tau, r_\tau) \in \mathcal{T}^u_t} |f(\mathbf{x}_\tau; \theta^u_{\tau-1}) - r_\tau|. \tag{45}$$

Because $V_1, \ldots, V_{\mu^u_t}$ is the martingale difference sequence, applying Hoeffding-Azuma inequality over $V_1, \ldots, V_{\mu^u_t}$, we have

$$\frac{1}{\mu^u_t} \sum_{(\mathbf{x}_\tau, r_\tau) \in \mathcal{T}^u_t} \mathop{\mathbb{E}}_{r_\tau | \mathbf{x}_\tau} [|f(\mathbf{x}_\tau; \theta^u_{\tau-1}) - r_\tau| \mid \pi_\tau, u]$$
$$\leq \underbrace{\frac{1}{\mu^u_t} \sum_{(\mathbf{x}_\tau, r_\tau) \in \mathcal{T}^u_t} |f(\mathbf{x}_\tau; \theta^u_{\tau-1}) - r_\tau|}_{I_1} + (1 + \xi_t) \sqrt{\frac{2 \log(1/\delta)}{\mu^u_t}}. \tag{46}$$

For $I_1$, for any $\widetilde{\theta}$ satisfying $\|\widetilde{\theta} - \theta^u_0\|_2 \leq \mathcal{O}\left(\frac{(\mu^u_t)^3}{\rho\sqrt{m}} \log m\right)$, we have

$$\frac{1}{\mu^u_t} \sum_{(\mathbf{x}_\tau, r_\tau) \in \mathcal{T}^u_t} |f(\mathbf{x}_\tau; \theta^u_{\tau-1}) - r_\tau| \stackrel{(a)}{\leq} \frac{1}{\mu^u_t} \sum_{(\mathbf{x}_\tau, r_\tau) \in \mathcal{T}^u_t} |f(\mathbf{x}_\tau; \widetilde{\theta}) - r_\tau| + \mathcal{O}\left(\frac{3L}{\sqrt{2\mu^u_t}}\right)$$
$$\stackrel{(b)}{\leq} \frac{1}{\mu^u_t} \sqrt{\mu^u_t} \sqrt{\sum_{(\mathbf{x}_\tau, r_\tau) \in \mathcal{T}^u_t} (f(\mathbf{x}_\tau; \widetilde{\theta}) - r_\tau)^2} + \mathcal{O}\left(\frac{3L}{2\sqrt{\mu^u_t}}\right) \tag{47}$$
$$\underbrace{\leq}_{I_3} \sqrt{\frac{2\epsilon_1}{\mu^u_t}} + \mathcal{O}\left(\frac{3L}{\sqrt{\mu^u_t}}\right).$$

where $I_2$ is because of Lemma E.6 and $I_3$ is the direct application of Lemma E.8 (2): there exists $\widetilde{\theta}$ satisfying $\|\widetilde{\theta} - \theta^u_0\|_2 \leq \mathcal{O}\left(\frac{(\mu^u_t)^3}{\rho\sqrt{m}} \log m\right)$ such that $\frac{1}{2} \sum_{\tau=1}^{\mu^u_t} (f(\mathbf{x}_\tau; \widetilde{\theta}) - r_\tau)^2 \leq \epsilon_1$.

Combing Eq.(46) and Eq.(47), we have

$$\frac{1}{\mu^u_t} \sum_{(\mathbf{x}_\tau, r_\tau) \in \mathcal{T}^u_t} \mathop{\mathbb{E}}_{r_\tau | \mathbf{x}_\tau} [|f(\mathbf{x}_t; \theta^u_{t-1}) - r_t| \mid \pi_\tau, u]] \leq \sqrt{\frac{2\epsilon_1}{\mu^u_t}} + \mathcal{O}\left(\frac{3L}{\sqrt{2\mu^u_t}}\right) + (1 + \xi_t) \sqrt{\frac{2 \log(1/\delta)}{\mu^u_t}}. \tag{48}$$

Then, applying the union bound over $\delta$, for any $i \in [k], \tau \in [\mu^u_t]$.

Based on Lemma E.8 (4), for any $\widehat{\theta}^u_\tau, \tau \in [t]$, we have $\|\widehat{\theta}^u_\tau - \theta^u_0\|_2 \leq \mathcal{O}\left(\frac{(\mu^u_t)^3}{\rho\sqrt{m}} \log m\right)$. Thus, it holds that $\|\theta^u_\tau - \theta^u_0\|_2 \leq \mathcal{O}\left(\frac{(\mu^u_t)^3}{\rho\sqrt{m}} \log m\right)$.

Then, apply the union bound of $\delta$ over all $\mu^u_T$ rounds. The proof is completed. $\qquad\square$

**Lemma E.4.** *For any $\delta \in (0,1), 0 < \rho \leq \mathcal{O}(\frac{1}{L})$, suppose $0 < \epsilon_1 \leq 1$ and $m, \eta_1, J_1$ satisfy the conditions in Eq.(6). In a round $\tau$ where $u \in N$ is the serving user, let $x_\tau^*$ be the arm selected according to Bayes-optimal policy $\pi^*$:*

$$x_\tau^* = \arg \max_{\mathbf{x}_{\tau,i}, i \in [k]} h_u(\mathbf{x}_{\tau,i}),$$

*and $r_\tau^*$ is the corresponding reward. Then, with probability at least $1 - \delta$ over the randomness of initialization, after $t \in [T]$ rounds, the cumulative regret induced by $u$ with policy $\pi^*$ is upper bounded by:*

$$\frac{1}{\mu_t^u} \sum_{(\mathbf{x}_\tau^*, r_\tau^*) \in \mathcal{T}_t^{u,*}} \mathbb{E}_{r_\tau^* | \mathbf{x}_\tau^*} [|f(\mathbf{x}_\tau^*; \theta_{\tau-1}^{u,*}) - r_\tau^*| \mid \pi^*, u]$$

$$\leq \sqrt{\frac{2\epsilon_1}{\mu_t^u}} + \mathcal{O}\left(\frac{3L}{\sqrt{2\mu_t^u}}\right) + (1 + \xi_t)\sqrt{\frac{2\log(\mu_t^u/\delta)}{\mu_t^u}}.$$

*where the expectation is taken over $r_\tau^*$ conditioned on $\mathbf{x}_\tau^*$, $\mathcal{T}_t^{u,*} = \{(\mathbf{x}_\tau^*, r_\tau^*) : u_\tau = u, \tau \in [t]\}$ are stored Bayes-optimal pairs up to round $t$ for $u$, and $\theta_{\tau-1}^{u,*}$ are the parameters trained on $\mathcal{T}_{\tau-1}^{u,*}$ according to Algorithm 3 in round $\tau - 1$.*

*Proof.* This proof is analogous to Lemma E.3. In a round $\tau$ where $u$ is the serving user, we define

$$V_\tau = \mathbb{E}_{r_\tau^* | \mathbf{x}_\tau^*} [|f(\mathbf{x}_\tau^*; \widehat{\theta}_{\tau-1}^{u,*}) - r_\tau^*|] - |f(\mathbf{x}_\tau^*; \widehat{\theta}_{\tau-1}^{u,*}) - r_\tau^*|. \tag{49}$$

where the expectation is taken over $r_\tau^*$ conditioned on $\mathbf{x}_\tau^*$ . Then, we have

$$\mathbb{E}[V_\tau | \mathbf{F}_\tau] = \mathbb{E}_{r_\tau^* | \mathbf{x}_\tau^*} [|f(\mathbf{x}_\tau^*; \widehat{\theta}_{\tau-1}^{u,*}) - r_{\tau,*}|] - \mathbb{E}[|f(\mathbf{x}_\tau^*; \widehat{\theta}_{\tau-1}^{u,*}) - r_\tau^*| \mid \mathbf{F}_\tau] = 0$$

Therefore, $V_1, \ldots, V_{\mu_t^u}$ is the martingale difference sequence. Then, following the same procedure of Lemma E.3, we can derive

$$\frac{1}{\mu_t^u} \sum_{(\mathbf{x}_\tau^*, r_\tau^*) \in \mathcal{T}_t^{u,*}} \mathbb{E}_{r_\tau^* | \mathbf{x}_\tau^*} [|f(\mathbf{x}_\tau^*; \theta_{\tau-1}^{u,*}) - r_\tau^*| \mid u]$$

$$\leq \sqrt{\frac{2\epsilon_1}{\mu_t^u}} + \mathcal{O}\left(\frac{3L}{\sqrt{2\mu_t^u}}\right) + (1 + \xi_t)\sqrt{\frac{2\log(1/\delta)}{\mu_t^u}}.$$

Based on Lemma E.8 (4), for any $\widehat{\theta}_\tau^{u,*}, \tau \in [t]$, we have $\|\widehat{\theta}_\tau^{u,*} - \theta_0^u\|_2 \leq \mathcal{O}\left(\frac{(\mu_t^u)^3}{\rho\sqrt{m}} \log m\right)$. Thus, it holds that $\|\theta_\tau^{u,*} - \theta_0^u\|_2 \leq \mathcal{O}\left(\frac{(\mu_t^u)^3}{\rho\sqrt{m}} \log m\right)$.

$\square$

## E.1 ANCILLARY LEMMAS

**Lemma E.5.** *Suppose $m, \eta_1, \eta_1$ satisfy the conditions in Eq. (6). With probability at least $1 - \delta$, for any $\mathbf{x}$ with $\|\mathbf{x}\|_2 = 1$ and $t \in [T], u \in N$, it holds that*

$$|f(\mathbf{x}; \theta_t^u)| \leq 2 + \mathcal{O}\left(\frac{t^4 nL \log m}{\rho\sqrt{m}}\right) + \mathcal{O}\left(\frac{t^5 nL^2 \log^{11/6} m}{\rho m^{1/6}}\right) = \xi_t.$$

*Proof.* This is an application of Lemma C.3 in (Ban et al., 2021b). Let $\theta_0$ be randomly initialized. Then applying Lemma E.9, for any $\|\mathbf{x}\|_2 = 1$ and $\|\widehat{\theta}_t^u - \theta_0\| \leq w$, we have

$$
\begin{aligned}
|f(\mathbf{x}; \widehat{\theta}_t^u)| &\leq \underbrace{|f(\mathbf{x}; \theta_0)|}_{I_1} + |\langle \nabla_{\theta_0} f(\mathbf{x}_i; \theta_0), \widehat{\theta}_t^u - \theta_0 \rangle| + \mathcal{O}(L^2 \sqrt{m \log(m)}) \|\widehat{\theta}_t^u - \theta_0\|_2 w^{1/3} \\
&\leq \underbrace{2\|\mathbf{x}\|_2}_{I_1} + \underbrace{\|\nabla_{\theta_0} f(\mathbf{x}_i; \theta_0)\|_2 \|\widehat{\theta}_t^u - \theta_0\|_2}_{I_2} + \mathcal{O}(L^2 \sqrt{m \log(m)}) \underbrace{\|\widehat{\theta}_t^u - \theta_0\|_2 w^{1/3}}_{I_3} \\
&\leq 2 + \underbrace{\mathcal{O}(L) \cdot \mathcal{O}\left(\frac{t^3}{\rho\sqrt{m}} \log m\right)}_{I_2} + \underbrace{\mathcal{O}\left(L^2 \sqrt{m \log(m)}\right) \cdot \mathcal{O}\left(\frac{t^3}{\rho\sqrt{m}} \log m\right)^{4/3}}_{I_3} \quad (50) \\
&= 2 + \mathcal{O}\left(\frac{t^3 L \log m}{\rho\sqrt{m}}\right) + \mathcal{O}\left(\frac{t^4 L^2 \log^{11/6} m}{\rho m^{1/6}}\right)
\end{aligned}
$$

where $I_1$ is an application of Lemma 7.3 in (Allen-Zhu et al., 2019), $I_2$ is by Lemma E.10 (1) and Lemma E.8 (4), and $I_3$ is due to Lemma E.8 (4). $\qquad \square$

**Lemma E.6.** *For any $\delta \in (0,1)$, suppose $m$ satisfy the conditions in Eq.(6) and $\nu = \Theta((\mu_t^u)^6/\rho^2)$. Then, with probability at least $1 - \delta$, set $\eta_1 = \Theta(\frac{\nu}{\sqrt{2}\mu_t^u m})$ for algorithm 1-3, for any $\widetilde{\theta}$ satisfying $\|\widetilde{\theta} - \theta_0^u\|_2 \leq \mathcal{O}\left(\frac{(\mu_t^u)^3}{\rho\sqrt{m}} \log m\right)$ such that*

$$
\sum_{\tau=1}^{\mu_t^u} |f(\mathbf{x}_\tau; \theta_{\tau-1}^u) - r_\tau| \leq \sum_{\tau=1}^{\mu_t^u} |f(\mathbf{x}_\tau; \widetilde{\theta}) - r_\tau| + \mathcal{O}\left(\frac{3L\sqrt{\mu_t^u}}{\sqrt{2}}\right) \quad (51)
$$

*Proof.* This is a direct application of Lemma 4.3 in (Cao and Gu, 2019) by setting the loss as $L_\tau(\theta_{\tau-1}^u) = |f(\mathbf{x}_\tau; \theta_{\tau-1}^u) - r_\tau|$, and $R = \frac{(\mu_t^u)^3}{\rho} \log m, \epsilon = \frac{LR}{\sqrt{2\nu\mu_t^u}}$, and $\nu = \nu' R^2$, where $\nu'$ is some small enough absolute constant. Then, for any $\widetilde{\theta}$ satisfying $\|\widetilde{\theta} - \theta_0^u\|_2 \leq \mathcal{O}\left(\frac{(\mu_t^u)^3}{\rho\sqrt{m}} \log m\right)$, there exist a small enough absolute constant $\nu'$, such that

$$
\sum_{\tau=1}^{t} L_\tau(\widehat{\theta}_{\tau-1}^u) \leq \sum_{\tau=1}^{t} L_\tau(\widetilde{\theta}) + 3\mu_t^u \epsilon. \quad (52)
$$

Then, replacing $\epsilon$ completes the proof. $\qquad \square$

**Lemma E.7** (Lemma C.2 (Ban et al., 2021b)). *For any $\delta \in (0,1), \rho \in (0, \mathcal{O}(\frac{1}{L}))$, suppose the conditions in Theorem 4.2 are satisfied. Then, with probability at least $1 - \delta$, in each round $t \in [T]$, for any $\|\mathbf{x}\|_2 = 1$, $\theta_{t-1}^{u,*}, \theta_{t-1}^u$ satisfying $\|\theta_{t-1}^{u,*} - \theta_0^u\|_2 \leq \mathcal{O}\left(\frac{(\mu_t^u)^3}{\rho\sqrt{m}} \log m\right)$ and $\|\theta_{t-1}^u - \theta_0^u\|_2 \leq \mathcal{O}\left(\frac{(\mu_t^u)^3}{\rho\sqrt{m}} \log m\right)$, we have*

$$
\begin{aligned}
(1) \quad &|f(\mathbf{x}; \theta_{t-1}^{u,*}) - f(\mathbf{x}; \theta_{t-1}^u)| \\
&\leq \left(1 + \mathcal{O}\left(\frac{tL^3 \log^{5/6} m}{\rho^{1/3} m^{1/6}}\right)\right) \mathcal{O}\left(\frac{Lt^3}{\rho\sqrt{m}} \log m\right) + \mathcal{O}\left(\frac{t^4 L^2 \log^{11/6} m}{\rho^{4/3} m^{1/6}}\right) \quad (53) \\
&= \zeta_t
\end{aligned}
$$

$$
(2) \|\nabla_{\theta_{t-1}^u} f_1(\mathbf{x}; \theta_{t-1}^u)\|_2 \leq \left(1 + \mathcal{O}\left(\frac{tL^3 \log^{5/6} m}{\rho^{1/3} m^{1/6}}\right)\right) \mathcal{O}(L). \quad (54)
$$

**Lemma E.8** (Theorem 1 in (Allen-Zhu et al., 2019)). *For any $0 < \epsilon_1 \leq 1, 0 < \rho \leq \mathcal{O}(1/L)$. Given a user $u$, the collected data $\{\mathbf{x}_\tau, r_\tau^u\}_{\tau=1}^{\mu_t^u}$, suppose $m, \eta_1, J_1$ satisfy the conditions in Eq.(6). Define $\mathcal{L}(\theta^u) = \frac{1}{2}\sum_{(\mathbf{x},r)\in\mathcal{T}_t^u}(f(\mathbf{x}; \theta^u) - r)^2$. Then with probability at least $1 - \delta$, these hold that:*

1. For any $j \in [J]$, $\mathcal{L}(\theta_{(j)}^u) \leq (1 - \Omega(\frac{\eta_1 \rho m}{\mu_t^{u2}}))\mathcal{L}(\theta_{(j-1)}^u)$

2. $\mathcal{L}(\widehat{\theta}_{\mu_t^u}^u) \leq \epsilon_1$ in $J_1 = \frac{poly(\mu_t^u, L)}{\rho^2} \log(1/\epsilon_1)$ rounds.

3. $\mathcal{L}(\theta_0^u) \leq \mathcal{O}(\mu_t^u \log^2 m)$.

4. For any $j \in [J]$, $\|\theta_{(j)}^u - \theta_{(0)}^u\|_2 \leq \mathcal{O}\left(\frac{(\mu_t^u)^3}{\rho\sqrt{m}} \log m\right)$.

**Lemma E.9** (Lemma 4.1, (Cao and Gu, 2019)). *Suppose* $\mathcal{O}(m^{-3/2}L^{-3/2}[\log(TnL^2/\delta)]^{3/2}) \leq w \leq \mathcal{O}(L^{-6}[\log m]^{-3/2})$. *Then, with probability at least* $1 - \delta$ *over randomness of* $\theta_0$, *for any* $t \in [T]$, $\|\mathbf{x}\|_2 = 1$, *and* $\theta, \theta'$ *satisfying* $\|\theta - \theta_0\| \leq w$ *and* $\|\theta' - \theta_0\| \leq w$, *it holds uniformly that*

$$|f(\mathbf{x}; \theta) - f(\mathbf{x}; \theta') - \langle \nabla_{\theta'} f(\mathbf{x}; \theta'), \theta - \theta' \rangle| \leq \mathcal{O}(w^{1/3}L^2\sqrt{m\log(m)})\|\theta - \theta'\|_2.$$

**Lemma E.10.** *For any* $\delta \in (0, 1)$, *suppose* $m, \eta_1, J_1$ *satisfy the conditions in Eq.(6) and* $\theta_0$ *are randomly initialized. Then, with probability at least* $1 - \delta$, *for any* $\|\mathbf{x}\|_2 = 1$, *these hold that*

1. $\|\nabla_{\theta_0} f(\mathbf{x}; \theta_0)\|_2 \leq \mathcal{O}(L)$,

2. $|f(\mathbf{x}; \theta_0)| \leq 2$.

*Proof.* For (2), based on Lemma 7.1 in (Allen-Zhu et al., 2019), we have $|f(\mathbf{x}; \theta_0)| \leq 2$. Denote by $D$ the ReLU function. For any $l \in [L]$,

$$\|\nabla_{\mathbf{W}_l} f(\mathbf{x}; \theta_0)\|_F \leq \|\mathbf{W}_L D \mathbf{W}_{L-1} \cdots D \mathbf{W}_{l+1}\|_F \cdot \|D \mathbf{W}_{l+1} \cdots \mathbf{x}\|_F \leq \mathcal{O}(\sqrt{L})$$

where the inequality is according to Lemma 7.2 in (Allen-Zhu et al., 2019). Therefore, we have $\|\nabla_{\theta_0} f(\mathbf{x}; \theta_0)\|_2 \leq \mathcal{O}(L)$. $\qquad\square$

# F RELATIVE GROUP GUARANTEE

In this section, we provide a relative group guarantee with the expectation taken over all past selected arms. For $u, u' \in N \wedge u \neq u'$, we define

$$\underset{\mathbf{x}_\tau \sim \mathcal{T}_t^u | \mathbf{x}}{\mathbb{E}}[\widehat{\mathcal{N}}_u(\mathbf{x}_\tau)] = \{u, u' : \underset{\mathbf{x}_\tau \sim \mathcal{T}_t^u | \mathbf{x}}{\mathbb{E}}[|f(\mathbf{x}_\tau; \theta_{t-1}^u)] - \underset{\mathbf{x}_\tau \sim \mathcal{T}_t^u | \mathbf{x}}{\mathbb{E}}[f(\mathbf{x}_\tau; \theta_{t-1}^{u'})]| \leq \frac{\nu - 1}{\nu}\gamma\} \quad (55)$$

and

$$\underset{\mathbf{x}_\tau \sim \mathcal{T}_t^u | \mathbf{x}}{\mathbb{E}}[\mathcal{N}_u(\mathbf{x}_\tau)] = \{u, u' : \underset{\mathbf{x}_\tau \sim \mathcal{T}_t^u | \mathbf{x}}{\mathbb{E}}[\mathbb{E}[r_\tau | u]] = \underset{\mathbf{x}_\tau \sim \mathcal{T}_t^u | \mathbf{x}}{\mathbb{E}}[\mathbb{E}[r_\tau | u']]\} \quad (56)$$

where $\widehat{\mathcal{N}}_u(\mathbf{x}_\tau)$ is the detected group and $\mathcal{N}_u(\mathbf{x}_\tau)$ is the ground-truth group. Then, we provide the following lemma.

**Lemma F.1** (Lemma 4.6 Restated). *Assume the groups in* $N$ *satisfy* $\gamma$-*gap (Definition 2.2) and the conditions of Theorem 4.2 are satisfied. For any* $\delta \in (0, 1), \nu > 1$, *with probability at least* $1 - \delta$ *over the random initialization, there exist constants* $c_1, c_2$, *such that when*

$$t \geq \frac{n64\nu^2(1 + \xi_t)^2 \left(\log \frac{32\nu^2(1+\xi_t)^2}{\gamma^2} + \frac{9L^2c_1^2 + 4\epsilon_1 + 2\zeta_t^2}{4(1+\xi_t)^2} - \log\delta\right)}{\gamma^2(1 + \sqrt{3n\log(n/\delta)})} = \widetilde{T},$$

*given a user* $u \in N$, *it holds uniformly for Algorithms 1-3 that*

$$\underset{\mathbf{x}_\tau \sim \mathcal{T}_t^u | \mathbf{x}}{\mathbb{E}}[\widehat{\mathcal{N}}_u(\mathbf{x}_\tau) \subseteq \mathcal{N}_u(\mathbf{x}_\tau)]$$

*and* $\underset{\mathbf{x}_\tau \sim \mathcal{T}_t^u | \mathbf{x}}{\mathbb{E}}[\widehat{\mathcal{N}}_u(\mathbf{x}_\tau) = \mathcal{N}_u(\mathbf{x}_\tau)], if \nu \geq 2,$

*where* $\mathbf{x}_\tau$ *is uniformly drawn from* $\mathcal{T}_t^u | \mathbf{x}$ *and* $\mathcal{T}_t^u | \mathbf{x} = \{\mathbf{x}_\tau : u_t = u \wedge \tau \in [t]\}$ *is all the historical selected arms when serving* $u$ *up to round* $t$. *Recall that*

$$\zeta_t = \left(1 + \mathcal{O}\left(\frac{tL^3 \log^{5/6} m}{\rho^{1/3}m^{1/6}}\right)\right)\mathcal{O}\left(\frac{Lt^3}{\rho\sqrt{m}}\log m\right) + \mathcal{O}\left(\frac{t^4L^2 \log^{11/6} m}{\rho^{4/3}m^{1/6}}\right);$$

$$\xi_t = 2 + \mathcal{O}\left(\frac{t^4nL \log m}{\rho\sqrt{m}}\right) + \mathcal{O}\left(\frac{t^5nL^2 \log^{11/6} m}{\rho m^{1/6}}\right).$$

*Proof.* Given two user $u, u' \in N$ and an arm $\mathbf{x}_\tau$, let $r_\tau$ be the reward $u$ generated on $\mathbf{x}_\tau$ and $r'_\tau$ be the reward $u'$ generated on $\mathbf{x}_\tau$. Then, in round $t \in [T]$, we have

$$
\underset{\mathbf{x}_\tau \sim \mathcal{T}_t^u | \mathbf{x}}{\mathbb{E}} \left[ \underset{r_\tau, r'_\tau}{\mathbb{E}} [|r_\tau - r'_\tau| \mid \mathbf{x}_\tau] \right]
$$

$$
= \frac{1}{\mu_t^u} \sum_{\mathbf{x}_\tau \in \mathcal{T}_t^u | \mathbf{x}} \underset{r_\tau, r'_\tau}{\mathbb{E}} [|r_\tau - r'_\tau| \mid \mathbf{x}_\tau]
$$

$$
= \frac{1}{\mu_t^u} \sum_{\mathbf{x}_\tau \in \mathcal{T}_t^u | \mathbf{x}} \underset{r_\tau, r'_\tau}{\mathbb{E}} [|r_\tau - f(\mathbf{x}_\tau; \theta_{t-1}^u) + f(\mathbf{x}_\tau; \theta_{t-1}^u) - f(\mathbf{x}_\tau; \theta_{t-1}^{u'}) + f(\mathbf{x}_\tau; \theta_{t-1}^{u'}) - r'_\tau| \mid \mathbf{x}_\tau]
$$

$$
\leq \frac{1}{\mu_t^u} \sum_{\mathbf{x}_\tau \in \mathcal{T}_t^u | \mathbf{x}} \underset{r_\tau | \mathbf{x}_\tau}{\mathbb{E}} [|r_\tau - f(\mathbf{x}_\tau; \theta_{t-1}^u)| \mid u] + \frac{1}{\mu_t^u} \sum_{\mathbf{x}_\tau \in \mathcal{T}_t^u | \mathbf{x}} [|f(\mathbf{x}_\tau; \theta_{t-1}^u) - f(\mathbf{x}_\tau; \theta_{t-1}^{u'})|]
$$

$$
+ \frac{1}{\mu_t^u} \sum_{\mathbf{x}_\tau \in \mathcal{T}_t^u | \mathbf{x}} \underset{r'_\tau | \mathbf{x}_\tau}{\mathbb{E}} [|f(\mathbf{x}_\tau; \theta_{t-1}^{u'}) - r'_\tau| \mid u'],
$$

$$(57)$$

where the expectation is taken over $r_\tau, r'_\tau$ conditioned on $\mathbf{x}_\tau$. According to Lemma E.3 and Corollary F.2 respectively, for each $u \in N$, we have

$$
\frac{1}{\mu_t^u} \sum_{\mathbf{x}_\tau \in \mathcal{T}_t^u | \mathbf{x}} \underset{r_\tau | \mathbf{x}_\tau}{\mathbb{E}} [|r_\tau - f(\mathbf{x}_\tau; \theta_{t-1}^u)| \mid u],
$$

$$
\leq \sqrt{\frac{2\epsilon_1}{\mu_t^u}} + \mathcal{O}\left( \frac{3L}{\sqrt{2\mu_t^u}} \right) + (1 + \xi_t) \sqrt{\frac{2 \log(\mu_t^u/\delta)}{\mu_t^u}};
$$

$$
\frac{1}{\mu_t^u} \sum_{\mathbf{x}_\tau \in \mathcal{T}_t^u | \mathbf{x}} \underset{r'_\tau | \mathbf{x}_\tau}{\mathbb{E}} [|f(\mathbf{x}_\tau; \theta_{t-1}^{u'}) - r'_\tau| \mid u']
$$

$$(58)$$

$$
\leq \sqrt{\frac{2\epsilon_1}{\mu_t^u}} + \mathcal{O}\left( \frac{3L}{\sqrt{2\mu_t^u}} \right) + (1 + \xi_t) \sqrt{\frac{2 \log(\mu_t^u/\delta)}{\mu_t^u}} + \zeta_t.
$$

Due to the setting of Algorithm 1, $|f(\mathbf{x}_\tau; \theta_{t-1}^u) - f(\mathbf{x}_\tau; \theta_{t-1}^{u'})| \leq \frac{\nu-1}{\nu}\gamma$ for any $u, u' \in \widehat{\mathcal{N}}_{u_t}(\mathbf{x}_\tau)$, given $\mathbf{x}_\tau \in \mathcal{T}_t^u | \mathbf{x}$. Therefore, we have

$$
\underset{\mathbf{x}_\tau \sim \mathcal{T}_t^u | \mathbf{x}}{\mathbb{E}} \left[ \underset{r_\tau, r'_\tau}{\mathbb{E}} [|r_\tau - r'_\tau| \mid \mathbf{x}_\tau] \right]
$$

$$(59)$$

$$
\leq \frac{\nu-1}{\nu}\gamma + 2 \left( \sqrt{\frac{2\epsilon_1}{\mu_t^u}} + \mathcal{O}\left( \frac{3L}{\sqrt{2\mu_t^u}} \right) + (1 + \xi_t) \sqrt{\frac{2 \log(\mu_t^u/\delta)}{\mu_t^u}} + \zeta_t \right)
$$

Next, we need to lower bound $t$ as the following:

$$
\sqrt{\frac{2\epsilon_1}{t}} + \frac{3Lc_1}{\sqrt{2\mu_t^u}} + (1 + \xi_t) \sqrt{\frac{2 \log(t/\delta)}{\mu_t^u}} + \zeta_t \leq \frac{\gamma}{2\nu}
$$

$$
\left( \sqrt{\frac{2\epsilon_1}{\mu_t^u}} + \frac{3Lc_1}{\sqrt{2\mu_t^u}} + (1 + \xi_t) \sqrt{\frac{2 \log(\mu_t^u/\delta)}{\mu_t^u}} + \zeta_t \right)^2 \leq \frac{\gamma^2}{4\nu^2} \quad (60)
$$

$$
\Rightarrow 4 \left( \left( \sqrt{\frac{2\epsilon_1}{\mu_t^u}} \right)^2 + \left( \frac{3Lc_1}{\sqrt{2\mu_t^u}} \right)^2 + \left( (1 + \xi_t) \sqrt{\frac{2 \log(\mu_t^u/\delta)}{\mu_t^u}} \right)^2 + (\zeta_t)^2 \right) \leq \frac{\gamma^2}{4\nu^2}
$$

By simple calculations, we have

$$
\log \mu_t^u \leq \frac{\gamma^2 \mu_t^u}{32\nu^2(1 + \xi_t)^2} - \frac{9L^2 c_1^2 + 4\epsilon_1 + 2\zeta_t^2}{4(1 + \xi_t)^2} + \log \delta \quad (61)
$$

Then, based on Lemme 8.1 in (Ban and He, 2021), we have

$$
\mu_t^u \geq \frac{64\nu^2(1 + \xi_t)^2}{\gamma^2} \left( \log \frac{32\nu^2(1 + \xi_t)^2}{\gamma^2} + \frac{9L^2 c_1^2 + 4\epsilon_1 + 2\zeta_t^2}{4(1 + \xi_t)^2} - \log \delta \right) \quad (62)
$$

Given the binomially distributed random variables, $x_1, x_2, \ldots, x_t$, where for $\tau \in [t]$, $x_\tau = 1$ with probability $1/n$ and $x_\tau = 0$ with probability $1 - 1/n$. Then, we have

$$\mu_t^u = \sum_{\tau=1}^{t} x_\tau \text{ and } \mathbb{E}[\mu_t^u] = \frac{t}{n}. \tag{63}$$

Then, apply Chernoff Bounds on the $\mu_t^u$ with probability at least $1 - \delta$, for each $u \in N$, we have

$$\mu_t^u \le \left(1 + \sqrt{\frac{3n \log(n/\delta)}{t}}\right) \frac{t}{n} \Rightarrow t \ge \frac{n\mu_t^u}{1 + \sqrt{3n \log(n/\delta)}} \tag{64}$$

Combining Eq.(62) and Eq.(64), we have: When

$$t \ge \frac{n64\nu^2(1+\xi_t)^2 \left(\log \frac{32\nu^2(1+\xi_t)^2}{\gamma^2} + \frac{9L^2 c_1^2 + 4\epsilon_1 + 2\zeta_t^2}{4(1+\xi_t)^2} - \log \delta\right)}{\gamma^2(1 + \sqrt{3n \log(n/\delta)})} = \widetilde{T}$$

it holds uniformly that:

$$2\left(\sqrt{\frac{2\epsilon_1}{\mu_t^u}} + \frac{3L}{\sqrt{2\mu_t^u}} + (1+\xi_t)\sqrt{\frac{2\log(t/\delta)}{t}} + \zeta_t^2\right) \le \frac{\gamma}{\nu}.$$

This indicates for any $u, u' \in N$ and satisfying $|f(\mathbf{x}_\tau; \theta_{t-1}^u) - f(\mathbf{x}_\tau; \theta_{t-1}^{u'})| \le \frac{\nu-1}{\nu}\gamma$, i.e., $u, u' \in \widehat{\mathcal{N}}_u(\mathbf{x}_\tau)$, we have

$$\mathbb{E}_{\mathbf{x}_\tau \sim \mathcal{T}_t^u|\mathbf{x}} \left[\mathbb{E}_{r_\tau, r_\tau'}[|r_\tau - r_\tau'| \mid \mathbf{x}_\tau]\right] \le \gamma. \tag{65}$$

This implies $\mathbb{E}_{\mathbf{x}_\tau \sim \mathcal{T}_t^u|\mathbf{x}}[\widehat{\mathcal{N}}_u(\mathbf{x}_\tau) \subseteq \mathcal{N}_u(\mathbf{x}_\tau)]$.

For any $u, u' \in \mathcal{N}_u(\mathbf{x}_\tau)$, we have

$$\mathbb{E}_{\mathbf{x}_\tau \sim \mathcal{T}_t^u|\mathbf{x}} \left[\mathbb{E}_{r_\tau, r_\tau'}[r_\tau - r_\tau' \mid \mathbf{x}_\tau]\right]$$
$$= \frac{1}{\mu_t^u} \sum_{\mathbf{x}_\tau \in \mathcal{T}_t^u|\mathbf{x}} \mathbb{E}_{r_\tau|\mathbf{x}_\tau}[r_\tau - f(\mathbf{x}_\tau; \theta_{t-1}^u) \mid u] + \frac{1}{\mu_t^u} \sum_{\mathbf{x}_\tau \in \mathcal{T}_t^u|\mathbf{x}} [f(\mathbf{x}_\tau; \theta_{t-1}^u) - f(\mathbf{x}_\tau; \theta_{t-1}^{u'})]$$
$$+ \frac{1}{\mu_t^u} \sum_{\mathbf{x}_\tau \in \mathcal{T}_t^u|\mathbf{x}} \mathbb{E}_{r_\tau'|\mathbf{x}_\tau}[f(\mathbf{x}_\tau; \theta_{t-1}^{u'}) - r_\tau' \mid u']$$
$$= 0$$

Thus, when $t \ge \widetilde{T}$, we have

$$\mathbb{E}_{\mathbf{x}_\tau \sim \mathcal{T}_t^u|\mathbf{x}}[|f(\mathbf{x}_\tau; \theta_{t-1}^u) - f(\mathbf{x}_\tau; \theta_{t-1}^{u'})|]$$
$$\le \frac{1}{\mu_t^u} \sum_{\mathbf{x}_\tau \in \mathcal{T}_t^u|\mathbf{x}} \mathbb{E}_{r_\tau|\mathbf{x}_\tau}[|r_\tau - f(\mathbf{x}_\tau; \theta_{t-1}^u)| \mid u] + \frac{1}{\mu_t^u} \sum_{\mathbf{x}_\tau \in \mathcal{T}_t^u|\mathbf{x}} \mathbb{E}_{r_\tau'|\mathbf{x}_\tau}[|f(\mathbf{x}_\tau; \theta_{t-1}^{u'}) - r_\tau'| \mid u']$$
$$\le \frac{\gamma}{\nu}$$

Because $\frac{\gamma}{\nu} \le \frac{\nu-1}{\nu}\gamma$ when $\nu \ge 2$. Thus, we have $\mathbb{E}_{\mathbf{x}_\tau \sim \mathcal{T}_t^u|\mathbf{x}}[|f(\mathbf{x}_\tau; \theta_{t-1}^u) - f(\mathbf{x}_\tau; \theta_{t-1}^{u'})|] \le \frac{\nu-1}{\nu}\gamma$.

Therefore, by induction, this is enough to show $\mathbb{E}_{\mathbf{x}_\tau \sim \mathcal{T}_t^u|\mathbf{x}}[\widehat{\mathcal{N}}_u(\mathbf{x}_\tau) = \mathcal{N}_u(\mathbf{x}_\tau)]$ when $\nu \ge 2$ and $t \ge \widetilde{T}$.

The proof is completed. $\square$

**Corollary F.2.** *For any $\delta \in (0, 1), \rho \in (0, \mathcal{O}(\frac{1}{L})]$, suppose $0 < \epsilon_1 \le 1$ and $m, \eta_1, J_1$ satisfy the conditions in Eq.(6). In each round $t \in [T]$, given $u \in N$, let $(\mathbf{x}_{t,j}, r_{t,j})$ be pair produced by some*

*policy $\pi_j$. Then, with probability at least $1 - \delta$ over the random initialization, for the user-learner $\theta_{t-1}^u$, we have*

$$\frac{1}{\mu_t^u} \sum_{(r_{\tau,j}, \mathbf{x}_{\tau,j}) \in \mathcal{T}_t^u | \pi_j} \mathbb{E}_{r_{\tau,j} | \mathbf{x}_{\tau,j}} \left[ |r_{\tau,j} - f(\mathbf{x}_{\tau,j}; \theta_{\tau-1}^u)| \mid \pi_j, u \right]$$

$$\leq \sqrt{\frac{2\epsilon_1}{\mu_t^u}} + \mathcal{O}\left(\frac{3L}{\sqrt{2\mu_t^u}}\right) + (1 + \xi_t)\sqrt{\frac{2\log(\mu_t^u/\delta)}{\mu_t^u}} + \zeta_t,$$

*where $\mathcal{T}_t^u | \pi_j = \{(r_{\tau,j}, \mathbf{x}_{\tau,j}) : u_t = u, \tau \in [t]\}$ is the historical data according to $\pi_j$ in the rounds where $u$ is the serving user.*

*Proof.* By the application of Lemma E.4, there exists $\theta_{t-1}^{u,j}$ satisfying $\|\theta_{t-1}^{u,j} - \theta_0^u\|_2 \leq \mathcal{O}\left(\frac{(\mu_t^u)^3}{\rho\sqrt{m}} \log m\right)$ trained on $\mathcal{T}_t^u | \pi_j$ following Algorithm 3. Then, similar to Lemma E.2, we have

$$\frac{1}{\mu_t^u} \sum_{(r_{\tau,j}, \mathbf{x}_{\tau,j}) \in \mathcal{T}_t^u | \pi_j} \mathbb{E}_{r_{\tau,j} | \mathbf{x}_{\tau,j}} \left[ |r_{t,j} - f(\mathbf{x}_{t,j}; \theta_{t-1}^u)| \right]$$

$$\leq \underbrace{\frac{1}{\mu_t^u} \sum_{(r_{\tau,j}, \mathbf{x}_{\tau,j}) \in \mathcal{T}_t^u | \pi_j} \mathbb{E}_{r_{\tau,j} | \mathbf{x}_{\tau,j}} \left[ |r_{t,j} - f(\mathbf{x}_{t,j}; \theta_{t-1}^{u,j})| \right]}_{I_1}$$

$$+ \underbrace{\frac{1}{\mu_t^u} \sum_{(r_{\tau,j}, \mathbf{x}_{\tau,j}) \in \mathcal{T}_t^u | \pi_j} \left[ |f(\mathbf{x}_{t,j}; \theta_{t-1}^{u,j}) - f(\mathbf{x}_{t,j}; \theta_{t-1}^u)| \right]}_{I_2}$$

$$\leq \sqrt{\frac{2\epsilon_1}{\mu_t^u}} + \mathcal{O}\left(\frac{3L}{\sqrt{2\mu_t^u}}\right) + (1 + \xi_t)\sqrt{\frac{2\log(\mathcal{O}(\mu^u k)/\delta)}{\mu^u}} + \zeta_t.$$

$$(66)$$

where $I_1$ is an application of Lemma E.4 and $I_2$ is because of E.7. The proof is complete. $\qquad\square$

