# OpenReview forum: "Neural Collaborative Filtering Bandits via Meta Learning"
_ICLR.cc/2023/Conference — Submitted to ICLR 2023_

### Official Review · Reviewer_2vmf · 2022-10-24

**Confidence:** 4
**Correctness:** 3
**Technical Novelty And Significance:** 3
**Empirical Novelty And Significance:** 3
**Recommendation:** 8

**Clarity, Quality, Novelty And Reproducibility:**

The paper is clear, it is sound. Best of my knowledge the idea of relative groups seems new. In the experimental section more details on the values of the parameters must be provided.

**Strength And Weaknesses:**

The paper is well written. Theoretical and experimental evidences of the efficiency of the proposed algorithm (Meta-Ban) are provided.

However, the reviewer has some comments and questions.

1/ The authors assume that the reward is bounded, but the noise seems to be unbounded. So it seems that equation (1) is not accurate. Why do the author not assume a sub-gaussian noise as in (Zhou et al 2020)?

2/ The authors use the over-parametrized neural network framework to analyze their algorithm, and they show a \sqrt {log T} regret bound improvement in comparison to the state-of-the-art. But in comparison to the state of the art, the regime where the neural network is analyzed is even more unrealistic than that of the state of the art. Indeed at each time step, the needed number of steps of the gradient descent in the order of T^56, while it was in \tilde O (T) in (Zhou et al 2020).

3/ In the experimental section, the value of J_2 is not given. I suggest to the authors to reassure the reader by giving the used value of J_2.


**Summary Of The Paper:**

In this paper the authors study the collaborative effects among users in order to improve recommendations. This line of works is known as collaborative filtering bandits.
The authors propose an interesting approach for this problem. Rather than assuming a linear model between the arm features and the expected reward, they use a neural bandit approach that allows dropping the linear assumption. Rather than using a clustering algorithm for capturing the collaborative effect, they define a relative group as a set of users that have similar expected rewards on the same item. Then, for each group (i.e. for each item) they learn a meta-neural bandit that shares historical data between users of the same group. The choice of the item for user u_t is done at the group level. This allows capturing collaborative effect among users. A neural bandit per user is also learnt in order to approximate the expected reward of items for the considered user. This allows building the estimated relative groups.
The proposed algorithm is analyzed using over-parameterized neural network regime. They improve by a factor \sqrt{\log T} the previous results of (Zhou et al 2020). Moreover the obtained pseudo-regret upper bound does not depend on the dimension of arm features. They also provide a high probability guarantee on the time spent to find the true relative groups.
Finally, the proposed algorithm is favorably tested with respect to baselines on different datasets.


**Summary Of The Review:**

Despite some drawbacks that the authors can fix, overall it is a good paper.
_______________________________________________
I read the rebuttal and I thank the authors for their answers.

---

> ### Author Response · Authors · 2022-11-13
> **Response to Reviewer 2vmf**
>
> Thank you very much for the constructive comments and suggestions.
>
>
> ### A1.
> Yes, our noise assumption is different from (Zhou et al 2020). We didn't impose any assumption on the distribution of noise, as long as its expectation is zero. We believe our assumption is more practical, because in real-world applications such as recommender systems or robot learning, the reward is usually bounded (e.g., in the range of [0,1]), but we don't have any knowledge about the noise. In our analysis, as long as the expectation of noise is zero, our regret bound will hold, given that our proof workflow is different from (Zhou et al 2020) as discussed in Remarks 4.3 and 4.4.  (Zhou et al 2020) has to keep the sub-Gaussian assumption of noise, in order to use Theorem 2 in (Abbasi-Yadkori et al., 2011).
>
>
> ### A2.
> Yes, we need wider neural networks compared to (Zhou et al 2020). This is the price to pay for achieving the improvement over (Zhou et al 2020). We remove the dependence on the effective dimension in (Zhou et al 2020) and improve by a multiplicative factor of $\sqrt{\log T}$.
>
> ### A3.
> For our implementation, the gradient descent (Algorithm s2 and 3) stops when the training error is smaller than $0.001$, but the $J_1$ and $J_2$ are restricted by $1000$. Thanks for the remainder. We've added this detail to the manuscript.

---

### Official Review · Reviewer_GgGh · 2022-10-24

**Confidence:** 4
**Correctness:** 4
**Technical Novelty And Significance:** 3
**Empirical Novelty And Significance:** 2
**Recommendation:** 5

**Clarity, Quality, Novelty And Reproducibility:**

The paper has good quality and clarity in general. However, in order to improve the current version, the authors should also emphasise more on recommendation in the experimentation.

**Strength And Weaknesses:**

Strength:

1. The presentation of the paper is intuitive and easy to understand. The authors introduce four challenges (i.e., C1-4) and subsequently tackle each challenge, which make it easy for the readers to follow
2. This is good that the authors also provide pseudo-code for Meta-Ban, with GradientDecent_Meta and GradientDecent_User.
3. The authors also have good regret analysis (Section 4), in which it shows that the proposed method is comparable with existing clustering of bandits and neural bandits
4. The experiments show that Meta-Ban outperforms all baselines across datasets over 10 runs, according to Figure 1-2.

Weaknesses:

1. Although the experiments yield good results, I believe the authors should focus more on recommendation datasets/tasks, rather than ML datasets. Specifically, among all the datasets used in the paper, only two of them (i.e., MovieLens and Yelp) are common recommendation datasets. In addition, the preprocessing steps in Appendix A.2 seem to be not common in my opinion. For example, “If the user’s rating is less than 2 stars (5 stars totally), its reward is 1; Otherwise, its reward is 0” or “we use K-means to divide users into 50 clusters”. Could you please give more explanations of the preprocessing steps? How did we decide the ‘threshold’ of 2 stars to set the reward of 1? How did we come up with 50 clusters? Can we have more/less clusters?
2. Also from Configurations paragraph of Appendix A.2, the authors use only a simple neural network of 2 fully-connected layers, which is a bit surprising to me as this paper introduces neural collaborative filtering bandits but with only a simple neural network of 2 layers. Can we have a larger neural network with more layers? If not, why?
3. In Section 5 Experiments, NeuUCB-ONE and NeuUCB-IND seem to perform quite well. Would it be possible to combine NeuUCB based models with ‘relative groups' and compare them with Meta-Ban? Moreover, can we also compare Meta-Ban with NeuMF [1] beside NeuUCB-* so that we can compare our method with two groups of baselines: i) neural network and ii) neural network + bandits for personalized recommendation tasks?
4. In the Appendix A.5, the authors mentioned that “This fluctuation is acceptable given that different input dimensions may contain different amount of information”. How can we determine if the fluctuation is ‘acceptable’?

[1] Neural Collaborative Filtering. WWW 2017.

**Summary Of The Paper:**

This paper explores the neural collaborative filtering bandits problem. Specifically, the authors first introduce relative groups to formulate groups given a specific content. Then, they use a meta learner and user learners to learn non-linear and linear reward functions. The authors also claim that this is the first work incorporating ‘collaborative effects’ in neural bandits. Besides experimental results, the authors also provide theoretical analysis of the regret upper bound of complexity, in which it is sharper than existing related works.

**Summary Of The Review:**

All in all, although the analysis is good, I personally found the experiments are a bit shallow towards recommendation. The authors should focus/provide more experiments on recommendation datasets such as Netflix, and also with more baselines towards personalized recommendations. In addition, I am also curious to see deeper analyses of run time, complexity, and scalability issues with dealing with large-scale systems.

---

> ### Author Response · Authors · 2022-11-18
> **Response to Reviewer GgGh**
>
> Thank you very much for the valuable comments and questions. We've provided the following explanations and experiments to address your concerns.
>
> ### A1.
>
> In each round, we tried to construct a pool of 10 arms, where only one arm has the reward of 1 and the 9 remaining arms have the rewards of 0. However, most restaurants in these two datasets are good restaurants (Star $\geq$ 3). If we set the good restaurant with reward 1, then the bad restaurants (start $\leq$ 2) will be outnumbered, since we need 9 bad restaurants in one round (9000 in 1000 rounds). Thus, we construct the arm pool in the inverse way by setting bad restaurants with reward 1. Here, we define a bad restaurant if it has less than 2 stars. If we set the threshold to 1,
> there will be very few bad restaurants. If we set the threshold to 3, these restaurants with 3 stars may not be ``bad".
> Since we are doing the experiments with 10000 rounds, the number of distinct users should not be too large. For example, if we use 2000 users, then each user will only have 5 data entries at the end. So, we set up a small user pool, e.g., 50 users.
>
>
> ### A2.
>
> In this paper, we try to propose a generic framework to combine meta-learning and bandits with the neural network approximation. Since the UCB in Meta-Ban only depends on the gradient, the neural network can be easily replaced by other different structures.
>
> We have added the ablation study to investigate the effect of the number of layers of neural networks in Appendix A.6 and as follows.
>
> We also would like to clarify that the evaluation on ML datasets is very important.
> On ML datasets, we consider each class as a user to hold an exclusive reward function. As some classes have correlations, the goal of ML datasets also is to find the classes with strong correlations and leverage this information to improve the qualify of classification.
> Meta-Ban (Proposed algorithm) can cluster similar classes (users) and leverage "collaborative effects" among classes to improve the performance. In contrast, NeuUCB-ONE will always consider all classes as one group which may contain some classes with negative effects; NeuUCB-IND will consider each class as a single group.  The fact that the proposed algorithm outperforms NeuUCB-ONE and NeuUCB-IND, demonstrates the importance of utilizing correlations among classes. This is one of the crucial applications of Meta-Ban.
>
> **The cumulative regret of 10000 rounds on MovieLens with the different number of layers.**
>
>    |            | NeuUCB-ONE         | NeuUCB-IND     | Meta-Ban      |
> |         -----    |        -----         |      -----          |   -----|
> |2 layers   | 4939         | 7491          |  **4673**      |
> |4 layers | 5017 | 7603    | **4498**   |
> |8 layers |  5033 | 7764  | **4796** |
> |10 layers | 5008 | 7797 |  **4824** |
>
> **The cumulative regret of 10000 rounds on Yelp with the different number of layers.**
>
> |             |NeuUCB-ONE  |NeuUCB-IND |Meta-Ban    |
> |   -----    |        -----         |      -----          |   -----  |
> |2 layers |  7683    | 8351  |  **7587** |
> |4 layers |  **7603** | 8386 | 7767 |
> | 8 layers|  7764  | 8366 | **7604** |
> |10 layers| 7797 | 8373 | **7541**|
>
> ### A3.
>
> NeuMF works on the conventional recommendation setting, where a training dataset is given to train the model. However, our algorithms and baselines all work on the cold-start and online recommendation setting to balance between the exploitation and exploration. So, it is difficult to construct a setting to compare these two lines of approaches fairly. Meanwhile, to the best of our knowledge, all existing bandit works like [1,2] do not include the conventional approach like NeuML as the baseline.
>
>
>
> ### A4.
>
> We are sorry for the use of subjective sentences here. The fluctuation rate of Meta-Ban is 12.2 \% ((highest regret - lowest regret)/lowest regret) in Table 1,  while NeuUCB-ONE is 17.8\%,  LOCB is 21.3 \% , SCLUB is 25.2 \% , COFIBA  67.9\%, CLUB is 20.6 \%. Note that the performance of NeuUCB-IND is not good, which is close to random guessing. Thus, we will say that Meta-Ban achieves the lowest fluctuation rate among all methods in the updated version.
>
>
>
>
> [1] Li, Shuai, Alexandros Karatzoglou, and Claudio Gentile. "Collaborative filtering bandits." In Proceedings of the 39th International ACM SIGIR conference on Research and Development in Information Retrieval, pp. 539-548. 2016.
>
> [2] Gentile, Claudio, et al. "On context-dependent clustering of bandits." International Conference on Machine Learning. PMLR, 2017.

---

> > ### Comment · Reviewer_GgGh · 2022-11-23
> > **Thanks for the response, but I decided to keep my score**
> >
> > Hi. Thanks for the response. I have also read the reviews and responses from other reviewers.
> >
> > A1, A2, and A4 somewhat answered my concerns, although Appendix A.6 (A2) needs to have further detailed explanations instead of providing only the cumulative regret; but I do not see A3 convincing. NeuMF [1] was published in 2017; thus, it is understandable that [2, 3] did not compare. However, since the proposed method is trying to do neural network approximation in the context of recommendation, it would be beneficial to the recsys community to compare with neural network based recommendation methods. At the current version, I still do not see it's convincing to raise the score. Thank you.
> >
> > [1] Neural Collaborative Filtering. WWW 2017.
> >
> > [2] Li, Shuai, Alexandros Karatzoglou, and Claudio Gentile. "Collaborative filtering bandits." In Proceedings of the 39th International ACM SIGIR conference on Research and Development in Information Retrieval, pp. 539-548. 2016.
> >
> > [3] Gentile, Claudio, et al. "On context-dependent clustering of bandits." International Conference on Machine Learning. PMLR, 2017.

---

> > > ### Author Response · Authors · 2022-11-24
> > > **Add Experiments Compared to NCF**
> > >
> > > We really appreciate reviewer's response, and we are glad that we addressed your concerns to some extent.
> > >
> > > We agree with reviewer's opinion that adding NCF[1] as a baseline will be beneficial to recsys community. Thus, we adapted NCF to online recommendation setting and conducted the following experiments.
> > >
> > > [1] Neural Collaborative Filtering. WWW 2017.
> > >
> > > **Experiment Setup**. There are totally $T=10000$ rounds of recommendation. In each round $t \in [T]$, a user will be served, and 10 items will be presented. The algorithm should select an item and recommend it to the user. After this recommendation, a reward will be returned. The evaluation metric is the cumulative regret. The regret of one round is 1 if the reward is 0; Otherwise, regret is 0. Here, we choose the two recommendation datasets: Movielens and Yelp. The methods of processing datasets and extracting features are as same as the descriptions in the Manuscript.
> > >
> > > **Implementation of NCF**.  We chose NeuMF in [1] as the baseline, which is the fusion of GMF and MLP and has achieved the best performance in [1]. We implemented NeuMF following the Figure 3 and source codes in [1]. For the MLP part, there are two layers with 100 neurons as same as Meta-Ban for the fair comparison. Then, one NeuMF Layer is added to calculate the output. As the user and item features are provided in these two datasets, the user and item feature vectors will directly be the input of NeuMF. The label of NeuMF will be the reward, because a reward of 1 indicates that it is a successful recommendation, and the reward of 0 shows this is a bad recommendation. The key adaptation is the training process. Because this is the online recommendation setting, we have to train NeuMF incrementally. For the fair comparison with Meta-Ban, in first 1000 rounds, we train NeuMF every 10 rounds; after 1000 rounds, we train NeuMF every 100 rounds. In each training, we will regard all past data as the training data to train the neural network incrementally. For example, in the $t=100$ round, we will treat all selected items and rewards of past 100 rounds as the training data to train NeuMF using mini-batch with the latest parameters. We use Adam as the optimizer consistent with other methods.
> > >
> > > **Results**. The results on Movielens and Yelp are reported as follows (5 runs).
> > >
> > > Movielens (the cumulative regret):
> > >
> > > | |1000| 2000| 4000| 7000|10000 (rounds)|
> > > |---  |--- |--- |--- |--- |--- |
> > > |NeuMF| 528.6 $\pm$ 16.8|  1017.6 $\pm$ 15.7|  1950.8 $\pm$ 23.8|  3361.2 $\pm$ 34.5| 4754.2 $\pm$ 35.7 |
> > > |Meta-Ban| **499.25 $\pm$ 13.9**| **973.5 $\pm$ 16.3**| **1882.0 $\pm$ 22.1**| **3237.2 $\pm$ 6.2**| **4592.5 $\pm$ 29.5**|
> > >
> > >
> > > Yelp (the cumulative regret):
> > >
> > > |  | 1000 |2000| 4000| 7000| 10000 (rounds)|
> > > |    --- |--- |--- |--- |--- |--- |
> > > |NeuMF| 791.8 $\pm$ 34.0| 1562.8 $\pm$ 25.9| 3067.0 $\pm$ 19.42| 5320.6 $\pm$ 27.33| 7565.2 $\pm$ 62.1|
> > > |Meta-Ban| **776.5 $\pm$ 8.3**| **1526.5 $\pm$ 11.0**| **2998.2 $\pm$ 32.2**| **5194.7 $\pm$ 64.9**| **7369.5 $\pm$ 63.1**|
> > >
> > >
> > >
> > >
> > >
> > > **Analysis**. Meta-Ban outperforms NeuMF on these two datasets. There are two possible reasons. (1) Meta-Ban has explicit exploration while NeuMF is a greedy approach. Exploration is very important in online recommendation setting to obtain new information for the long-term benefits. Meta-Ban has the UCB-based exploration to incorporate user-level and group-level exploration. In contrast, NeuMF strongly relies on the past data, which can be considered as a greedy approach. (2) Meta-Ban explicitly clusters users to leverage collaborative effects while NeuMF implicitly utilizes the collaborative effects. Meta-Ban dynamically clusters users via meta learning and leverage the user group to make recommendation. In contrast, NeuMF does not have the clustering procedure and implicitly incorporates the collaborative effects via matrix factorization and MLP training.
> > >
> > >
> > > We will definitely add these experimental results to the manuscript and believe this will be a good addition to our work, as the reviewer suggested.
> > >
> > > Thanks,
> > >
> > > sincerely,
> > >
> > > Authors

---

### Official Review · Reviewer_ygez · 2022-10-27

**Confidence:** 3
**Correctness:** 3
**Technical Novelty And Significance:** 3
**Empirical Novelty And Significance:** 3
**Recommendation:** 5

**Clarity, Quality, Novelty And Reproducibility:**

questions:
- In the first step of the proposed algos, it would like to infer a user's relative group. The question is what assumptions made on these relative group? are they independent? can a user belong to multiple groups? If they are dependent, how does it impact the algorithm in terms of regret analysis and performance? Also, what are constraints or structure properties do we want to place when designing relative group?
- In Challenge 1, the paper also asked whether the returned group is the true relative group, but was not clear on its importance and what's the consequence if it is not true relative group (sorry if overlook).
- in Page 4 when defining Group Inference, the paper mentioned that it is natural to use the universal approximator. Could they explain intuition on why and this is chosen?
-  In challenge 3, the paper mentioned the rapidly-changing relative groups, and wonder if the authors could provide more detail information on what mathematical assumptions and properties on them, as they would likely impact how we think/propose solutions in terms of efficacy and efficiency.

cosmetics:
1. In Algorithm 1 on page 4, missing a space in row 1 between greek letters. Also, no space before J_1 and comma was misplaced.

**Strength And Weaknesses:**

Strengths:
- The paper studies Neural Collaborative filtering bandits that incorporate collaborative effects among users with both linear or non-linear reward assumptions, and proposes a meta-learning based bandits algorithm (Meta-Ban) that could adapt to dynamic groups with a UCB-type exploration.
- the paper provides theoretical analysis on its regret upper bound for the proposed algo, Meta-Ban, and evaluate them on 10 datasets and show that the proposed algorithms outperforms against baselines.

Weakness or questions:
- In the first step of the proposed algos, it would like to infer a user's relative group. The question is what assumptions made on these relative group? are they independent? can a user belong to multiple groups? If they are dependent, how does it impact the algorithm in terms of regret analysis and performance? Also, what are constraints or structure properties do we want to place when designing relative group?
- In Challenge 1, the paper also asked whether the returned group is the true relative group, but was not clear on its importance and what's the consequence if it is not true relative group (sorry if overlook).
- in Page 4 when defining Group Inference, the paper mentioned that it is natural to use the universal approximator. Could they explain intuition on why and this is chosen?
-  In challenge 3, the paper mentioned the rapidly-changing relative groups, and wonder if the authors could provide more detail information on what mathematical assumptions and properties on them, as they would likely impact how we think/propose solutions in terms of efficacy and efficiency.

**Summary Of The Paper:**


The paper studies Neural Collaborative filtering bandits that incorporate collaborative effects among users with both linear or non-linear reward assumptions, and proposes a meta-learning based bandits algorithm (Meta-Ban) that could adapt to dynamic groups with a UCB-type exploration. It then provides theoretical analysis on its regret upper bound for the proposed algo, Meta-Ban, and evaluate them on 10 datasets and show that the proposed algorithms outperforms against baselines.

**Summary Of The Review:**


The paper studies Neural Collaborative filtering bandits that incorporate collaborative effects among users with both linear or non-linear reward assumptions, and proposes a meta-learning based bandits algorithm (Meta-Ban) that could adapt to dynamic groups with a UCB-type exploration. It then provides theoretical analysis on its regret upper bound for the proposed algo, Meta-Ban, and evaluate them on 10 datasets and show that the proposed algorithms outperforms against baselines.

Overall it was an interesting and important problems. However, it also raised many of questions that were not clear in the paper in terms of what assumptions for a couple of 4 challenges they aim to tackle in the paper, and how does assumptions and structure properties on these challenges could impact the efficacy and performance of the proposed algorithms. Please see some detail questions below:
- In the first step of the proposed algos, it would like to infer a user's relative group. The question is what assumptions made on these relative group? are they independent? can a user belong to multiple groups? If they are dependent, how does it impact the algorithm in terms of regret analysis and performance? Also, what are constraints or structure properties do we want to place when designing relative group?
- In Challenge 1, the paper also asked whether the returned group is the true relative group, but was not clear on its importance and what's the consequence if it is not true relative group (sorry if overlook).
- in Page 4 when defining Group Inference, the paper mentioned that it is natural to use the universal approximator. Could they explain intuition on why and this is chosen?
-  In challenge 3, the paper mentioned the rapidly-changing relative groups, and wonder if the authors could provide more detail information on what mathematical assumptions and properties on them, as they would likely impact how we think/propose solutions in terms of efficacy and efficiency.

---

> ### Author Response · Authors · 2022-11-13
> **Response to Reviewer ygez**
>
> Thanks very much for the reviewer's comments and questions. We provide the following clarifications and explanations to address your concerns.
>
> ### A1.
> Definition 2.1 indicates that the relative groups are non-overlapping and independent to each other. Based on Definition 2.1, for all the users within the same group, it is required that they have the same expected reward. So, the case does not exist that a user belongs to multiple groups. Definition 2.2 is the constraint we placed on relative groups. This is to guarantee that, for any two groups, they have different behaviors, so that the proposed algorithm can detect and distinguish them.
>
>
> ### A2.
> The proposed algorithm, Meta-Ban, has a good property: the performance of Meta-Ban will not be worse than the algorithm that only considers the current user (no collaborative effects), even if the relative groups are wrongly estimated. This is because we use the meta-learner to represent a group. In existing works like [1] [2], the group parameters are the linear combination of all users, so their performance may be deteriorated by clustering wrong users under the real-world application scenarios. In contrast, we use the meta-learner to represent a group to minimize the objective in Algorithm 2. Based on our analysis (Lemma D.1), the objective can be minimized. Thus, the meta-learner has the ability of generalizing to every user in this group.
>
> If the detected relative group is true, Meta-Ban can significantly improve the performance. For instance, given a true group of 10 users, Meta-Ban will consider these 10 users as one ``big" user. In this case, the objective in Algorithm 2 can be thought of as one user with 10 times data points, because these 10 users have the same expected reward. With more data, Meta-Ban can improve the quality of decision making.
>
>
>
> ### A3.
> Based on Eq.(1), the reward is governed by some unknown function $h_u$, which can be either linear or non-linear. So, we should use some universal approximator (e.g., over-parameterized neural networks) to estimate $h_u$, which will be used to infer groups.
>
>
> ### A4.
> Definition 2.1 defines the relative groups and Definition 2.2 is the only constraint placed on the groups.
> Since the user and items may be changing in each round, it is required that the algorithm could accurately and rapidly sense the groups.
> Our proposed algorithm, Meta-Ban, uses the meta-learner to represent groups that can adapt to newly formed groups with a few steps of gradient descent to improve the efficiency. Moreover, we provide the performance guarantee of Meta-Ban (Theorem 4.2) and the convergence of the meta-learner (Lemma D.1) to prove its effectiveness.
>
>
>
>
> [1] Li, Shuai, Alexandros Karatzoglou, and Claudio Gentile. "Collaborative filtering bandits." In Proceedings of the 39th International ACM SIGIR conference on Research and Development in Information Retrieval, pp. 539-548. 2016.
>
> [2] Gentile, Claudio, et al. "On context-dependent clustering of bandits." International Conference on Machine Learning. PMLR, 2017.

---

### Official Review · Reviewer_Uhmg · 2022-10-28

**Confidence:** 4
**Clarity, Quality, Novelty And Reproducibility:** see my comments above
**Correctness:** 2
**Technical Novelty And Significance:** 3
**Empirical Novelty And Significance:** 2
**Recommendation:** 3

**Strength And Weaknesses:**

Strengths:
The problem considered in the paper is a nice generalization of the past work to non-linear rewards, and is relevant in practice. Some of the theoretical results in the paper (e.g., regret bounds not depending on dimension, no distributional assumptions on the arms) are interesting, but unfortunately lack a thorough discussion on how these were obtained.

Weaknesses:
I believe the paper is a little bit under-cooked. The clarity and presentation in the paper can be significantly improved. The algorithm is not well motivated and described. The theoretical results are not well discussed and the proofs in the appendix could be presented better. Here are some key concerns I have:

* Clarity: I had to take multiple passes through the paper to clearly understand the notation, the main algorithm, and the theoretical results.
  - first of all, the algorithm needs to be clearly explained. what is the motivation behind introducing a meta learner in Algorithm 2? It is never mentioned.  What happens if we solve the objective in Algorithm 2 exactly (i.e., increase $J_2$ and don't warm-start $\Theta_{(0)}$ with $\Theta_{t-1}$)?  Will the resulting algorithm have poor performance?
  - why is $\ell_1$ norm used in Algorithm 2? Why not $\ell_2$ norm?
  -  it is not clear how the proposed algorithm is different from the algorithms proposed in prior works such as Li et al. 2016 [1]. Is the extension to nonlinear reward functions straightforward? What are the key challenges and insights needed for this extension? These things were never discussed in the paper.
  - while Theorem 4.2 is interesting, a proof sketch in the main paper would have helped the reader understand the proof better. Currently, it is very hard to understand the key ideas in the proof. I tried looking at the appendix, it is poorly organized and without a high level overview of the proof, I found it hard to read through the appendix. For example, I tried looking at Theorem B.1. It refers to Lemma E.4, which in turn refers to Lemma E.3, which in turn refers to E.5 and so on. By the time I traced this graph, I lost track of many things.
  - some key details in the experiment section are missing. For example, details on how the classification datasets are converted into collaborative bandit problem  should be added to the main paper. This will help the reader better understand the task being solved.
  - there are several issues with the notation. Here are few examples: L in section 3 appeared before it was defined. Operator grad_{theta^u for u in N} in line 8 of algorithm 2 was never defined.

* Theoretical Results:
  - the regret looks too good to be true. why is there no dependence on $d$? Even in the linear case (without and collaborative filtering stuff), the minimax regret has dependence on dimension. Are there any assumptions that are causing this to happen? In particular, is assumption 4.1 the main reason for these rates? A discussion on this would be appreciated.
  - On a related note, why is there only a $T^{1/2}$ term in the regret? The NTK kernel is a very complex kernel with slowly decaying eigen spectrum. It's worst case regret bound in standard contextual bandit setting scales as $O(T^{1-d^{-1}})$. Can the authors explain what is causing this discrepancy?
  - the cluster sizes and the number of clusters never came up in the analysis at all ($q_{t,i}$). Shouldn't the regret depend on these quantities? I'd appreciate if the authors make this dependence more explicit.

* Analysis:
  -  The analysis in the paper seems to be in the NTK regime. In fact, the paper relies on a number of results proved in the past works on NTK. There are some caveats with these results that the authors never brought up in the paper.
      - First, most of the past works on NTK assume certain kind of initialization for the NN. But the initialization scheme used in Algorithm 3 doesn't match with the past works. Given this, do the past results carry over directly to the setting in this paper? In particular, do Lemmas E.6 to E.10 follow directly from past works?
      - Second, the result in Theorem 4.2 doesn't say anything about how the context vectors are generated suggesting that the result holds even if the context vectors are generated adversarially. But I doubt if this is the case. The NTK results such as the ones in [2] require the context vectors for all the rounds to be generated by an oblivious adversary before the start of the algorithm. If the context vectors are generated adversarially, the NTK analysis in these works doesn't go through. Can the authors comment on this?

* Experiments Issues:
  - I have a concern regarding how the ML classification datasets are converted to the collaborative bandits problem. Based on the description in Appendix A.2, it looks like the reward function for any given context vector is the same for all the users (i.e., h_u(x) is independent of u). This suggests that there is a single cluster for any given arm. If this is the case, the problem simply boils down to the neural contextual bandit problem studied in [2]. Then, why is there a huge difference in performance between the proposed technique and NeuUCB-ONE?


[2] Zhou, Dongruo, Lihong Li, and Quanquan Gu. "Neural contextual bandits with ucb-based exploration." In International Conference on Machine Learning, pp. 11492-11502. PMLR, 2020.

**Summary Of The Paper:**

The paper considers the problem of online collaborative filtering. In each round of this problem, a user is randomly sampled and a set of items/arms are provided to the learner. The goal of the learner is to recommend one of these items to the user so that the cumulative reward it collects over time is maximized. One key structure that is usually imposed on this problem is the cluster structure: for each item, it is assumed that the users can be grouped into a small number of clusters, where users within the same cluster have the same reward for the item. This problem has received significant attention in recent years. Previous works assumed that the reward of each user-item pair is a linear function of the item vector [1]. The current paper extends this to consider more general non-linear reward functions. In particular, the paper uses neural networks to model the unknown reward function. The authors provide a meta learning based algorithm (meta-ban) that achieves O(sqrt{nT}) regret for this problem, where n is the number of users and T is the number of online rounds. Experimentally, the authors show that the proposed technique achieves better regret than baselines on benchmark tasks.

[1] Li, Shuai, Alexandros Karatzoglou, and Claudio Gentile. "Collaborative filtering bandits." In Proceedings of the 39th International ACM SIGIR conference on Research and Development in Information Retrieval, pp. 539-548. 2016.

**Summary Of The Review:**

While the problem being studied is interesting, the clarity in the paper needs to be significantly improved. And there are several concerns with the theoretical and experimental results presented in the paper. I'd appreciate if the authors address my comments above.

---

> ### Author Response · Authors · 2022-11-13
> **Resonpse to Reviewer Uhmg (3)**
>
> ## Analysis
>
> ### A9.
> We would like to clarify that, the initialization scheme can be found on Page 4, right above Assumption 4.1, which is consistent with past work [6] and can be covered by [5]. We also want to point out that  E.6 to E.10 indeed are direct applications of past works and we've cited their papers in the Lemmas.
>
>
> ### A10.
> We would like to clarify that, our regret analysis does not depend on the NTK regime, so we didn't introduce NTK in this manuscript.
> Remarks 4.3 and 4.4 provide some insights regarding how our proof workflow is different from [2]. The proof sketch is on page 17.
> In the proof of our analysis, no matter how the context is generated, as long as it satisfies Eq. (1) in Problem Definition, the regret bound will hold. The intuition behind this is that we are learning the fixed reward function $h_u$. Given any $x_{t,i}$, the expected reward is fixed $E[r_{t,i}] = h_u(x_{t,i})$. So, with more data pairs $(x_{t,i}, r_{t,i})$, the neural network will be able to better approximate $h_u$, no matter how $x_{t,i}$ is generated.
> In Lemma E.3,
> we follow this insight to construct the martingale difference sequence with respect to $h_u$.
>
>
> [6] Cao, Yuan, and Quanquan Gu. "Generalization bounds of stochastic gradient descent for wide and deep neural networks." Advances in neural information processing systems 32 (2019).
>
>
>
> ## Experiments
>
> ### A11.
>
>
> Sorry for the insufficient details provided and we would like to make a clarification here. In the ML dataset, each class is considered as a user, with a specific reward function $h_u$. So, in the 10-class dataset, there are 10 users with ten different reward functions.
> As some classes have correlations, the goal of ML datasets also is to find the classes with strong correlations and leverage this information to improve the qualify of classification. This is one of the crucial applications of the proposed algorithm.
> Meta-Ban (Proposed algorithm) can cluster similar classes (users) and leverage the correlations ("collaborative effects") among classes to improve the performance. In contrast, NeuUCB-ONE will always consider all classes as one group which may contain some classes with negative effects.  So, the proposed algorithm outperforms NeuUCB-ONE.
>
>
>
>
> **We really appreciate the effort the reviewer spent on reading this paper. Since we are not 100\% sure that we understand the reviewer's questions correctly, we would be glad to provide more details and further modifications on the manuscript.**

---

> ### Author Response · Authors · 2022-11-13
> **Response to Reviewer Uhmg (2)**
>
> ## Theoretical Results
>
> ### A6 (Independent of $d$).
> It is true that linear bandits have to depend on $d$, because the approximation quality of linear (ridge) regression is affected by $d$. However, in neural bandit where neural networks are used to approximate the reward functions, the dependence on $d$ can be removed. Because as long as the dataset is ``distinguishable", no matter how large the input dimension $d$ of each data point is, we can always find a neural network to achieve small training error, i.e., training error $<\epsilon$ for any $ 0< \epsilon <1$. The price of finding such a neural network is taking longer training time and applying wider networks. [5] has proven this property by providing the convergence guarantee of networks.
> This again indicates that we can have the same approximation quality no matter how large $d$ is, given that we can manually setup and train the neural network.
> To better understand this, given the dataset with size $T$, we can always consider $d=1$ for each input data point for neural networks, because $d=1$ and $d=100$ will have the same approximation quality (training error).
> Then, we can use the technique in non-contextual bandits (concentration inequality) to obtain a confidence interval for this approximation, achieving a regret bound without the dependence on $d$.
> This is the intuition behind this regret bound. Lemma E.3 is following the above idea to provide a confidence bound for the user parameter $\theta^u$.
>
> As the reviewer observed, Assumption 4.1 is indispensable to guarantee that the dataset formed by arms and rewards is distinguishable. Because if two arms are identical but have different rewards, then the training error cannot be zero for neural networks. Another price for this regret bound is that we have to adopt wider neural network, i.e., $m$ also depends on arm separateness $\rho$ in Theorem 4.2.
>
> We would like to emphasize that our proof workflow is different from [2]. The essence of [2] is still linear (ridge) regression but in the RKHS induced by NTK. Our proof leverages the approximation property of neural networks, which is distinct from the NTK regime (see Remarks 4.3 and 4.4).
>
> ### A7.
> As discussed in A6, our proof is not based on linear (ridge) regression or in the RKHS induced by kernels, which is the main reason causing this discrepancy.
>
> ### A8.
> We thank the reviewer for this interesting question.
> Theorem 4.2 provides the worst-case regret bound. In the worst case, there are $q_{t,i} = n$ groups, i.e., each user forms a group satisfying $\gamma$-gap constraint. Theorem 4.2 considers this case and thus Theorem 4.2 only depends on $n$. If we impose an additional assumption on the smallest size of user group $\geq \bar{n}$, then the regret bound will depend on $\bar{n}$. Because this additional assumption will guarantee that each group as least has $\bar{n}$ users. Then, we can use these $\bar{n}$ users' data to get tighter regret bound that depends on $O(\sqrt{nT/\bar{n}})$. This can be considered as one of the future works.
>
>
> [5] Z. Allen-Zhu, Y. Li, and Z. Song. A convergence theory for deep learning via over-parameterization. In International Conference on Machine Learning, pages 242–252. PMLR, 2019.

---

> ### Author Response · Authors · 2022-11-13
> **Response to Reviewer Uhmg (1)**
>
> Thanks very much for the detailed comments and valuable suggestions. We will try our best to address your concerns.
>
> ## Clarity.
>
> ### A1 (Motivation of Meta-Learning).
> We introduce the meta-learner to represent the behavior of a user group. The intuition behind this design is the following. We consider the behavior of a single user as a task, represented by a set of user parameters $\theta^u$. To incorporate the collaborative effects, we apply another set of parameters $\Theta$ to learn the behavior of a group of users. Because in meta-learning, a meta-learner can learn from a large number of different tasks quickly with a small amount of data [1], we use the meta-learner $\Theta$ to represent the behavior of a group of users. Thus, Algorithm 2 is for the meta-adaptation process in meta learning [1] starting with the meta-learner of the last round (warm-up) and a new group of users (tasks).  This intuition was briefly discussed on Page 4.
>
> If the objective in Algorithm 2 is solved exactly, then the meta-learner $\Theta_t$ will ``well-fit" the historical data of each user in this group, i.e., the training error is zero if we consider the historical data of all users in this group as the training dataset. If we start training $\Theta_t$ from $\Theta_{(0)}$, it will take more steps of gradient descent until convergence, and thus cost more time.  The resulting algorithm will have the same performance as the one with warm-up, given that they end up with the same training error.
>
> ### A2 (Choice of L1 Norm).
> We choose L1 regularization to neutralize the vanishing gradient in convergence analysis, because the use of the L1 regularization makes it easier to derive the upper bound and lower bound, which is used in the proof of Lemma D.1. Meanwhile, we would like to clarify that we are not suggesting that L2 regularization is a worse choice or makes the training process unable to converge.
>
>
>
> ### A3 (Key difference from previous works).
> There are two crucial differences from existing works. First, as the reviewer mentioned, we use the neural network to learn user-level non-linear reward functions, represented by $\theta^u$. This part is the adaption of neural bandits like in [2], but we use a different approach to finish the analysis in order to achieve the better regret bound. Remarks 4.3 and 4.4 provide some insights regarding this. The key challenge (novelty) is bringing in meta-learning to learn the non-linear representations of user groups with respect to individual user parameters. In contrast, existing works such as [3,4] only allow linear (fixed form) representations of user groups with respect to individual user parameters. Take [4] as an example. Given a user group $N$, we have the user parameters $\theta_u, u \in N$. In [4], they simply use the average of user parameters to represent the behavior of the group $N$, i.e., $\Theta = \frac{1}{|N|}\sum_{u \in N} \theta_u$. However, in real-world cases, each user can have different impacts on the behavior of his/her user group. All existing related works use a fixed form (linear combination of user parameters) to represent group behavior. So, to overcome this limitation, we propose to use the meta-learner to flexibly learn the behavior from a group of users by minimizing the objective in Algorithm 2, which has the generalization ability to each user.
>
> ### A4.
> We've added a proof sketch in Appendix (Page 17).
>
> ### A5.
> We've added these details to the manuscript (Page 8), and please refer to A11.
>
> ### A6.
> We've added the notations on Page 3.
>
>
>
>
> **Thanks for your valuable suggestions again and we've updated the manuscript.**
>
>
> [1] Finn, Chelsea, Pieter Abbeel, and Sergey Levine. "Model-agnostic meta-learning for fast adaptation of deep networks." International conference on machine learning. PMLR, 2017.
>
> [2] Zhou, Dongruo, Lihong Li, and Quanquan Gu. "Neural contextual bandits with ucb-based exploration." In International Conference on Machine Learning, pp. 11492-11502. PMLR, 2020.
>
> [3] Li, Shuai, Alexandros Karatzoglou, and Claudio Gentile. "Collaborative filtering bandits." In Proceedings of the 39th International ACM SIGIR conference on Research and Development in Information Retrieval, pp. 539-548. 2016.
>
> [4] Gentile, Claudio, et al. "On context-dependent clustering of bandits." International Conference on Machine Learning. PMLR, 2017.’’

---

> > ### Author Response · Authors · 2022-11-19
> > **Proof Overview of Theorem 4.2**
> >
> > ### Proof Sketch of Theorem 4.2.
> > Different from existing works (Zhou et al., 2020; Zhang et al., 2021)  that bound the regret of one round by kernel regression in Neural Tangent Kernel, we directly upper bound the mean of regret of overall $T$ rounds by building martingale difference sequence with respect to $h_u$.  First, we decompose the regret of $T$ rounds into three key terms (Eq. (9)), where the first term is the error induced by user learner $\theta^u $, the second term is the distance between user learner and meta learner, and the third term is the error induced by the meta learner $\Theta$.
> >
> > Then, Lemma E.2 provides an upper bound for the first term. Lemma E.2 is an extension of Lemma E.3, which is the key to remove the input dimension. Lemma E.3 has three terms with the complexity $O(\sqrt{T})$, where the first term is the training error induced by a class of functions around initialization, the second term is the price of choosing the function class, and the third term is confidence interval induced by concentration inequality for $f(\cdot; \theta^u)$. Lemma C.1 bounds the distance between user learner and meta learner. As this bound has the term $O(1/\sqrt{m})$, this bound can be reduced to $\sqrt{T}$ with the proper choice of $m$.
> > Lemma C.2 bounds the error induced by the meta learner using triangle inequality bridged by the user learner.
> >
> > Bounding the three terms in Eq. (9) completes the proof.

---

### Author Response · Authors · 2022-11-18
**Updates of Manuscript**

Dear all reviewers,

We've updated the manuscript. The main modifications are as follows:

(1) We fixed some notation issues and typos.

(2) We added descriptions for ML datasets (Page 8), to clarify that the evaluation on ML datasets is important and it can measure the ability of an algorithm to leverage the collaborative effects.

(3) We added one ablation study to investigate the effects of number of layers of neural network. Meta-Ban achieves the best performance across most cases.

---

### Decision · Program_Chairs · 2023-01-20

**Decision:**

Reject

**Justification For Why Not Higher Score:**

The theory is not strong enough and the experiments alone do not carry the paper.

This paper should be treated as a **weak reject**, reject and bump up if there are not better papers.

**Justification For Why Not Lower Score:**

This paper solves a novel problem in an active research area.

**Metareview: Summary, Strengths And Weaknesses:**

This paper studies meta-learning in neural bandits. The authors bound the regret of their algorithm and also evaluate it empirically.

The strength of the paper is that this is the first time that this problem is studied. Arguably though, general algorithms have been proposed in the bandit community before, such as

* [Meta-Thompson Sampling](https://proceedings.mlr.press/v139/kveton21a.html)

* [Bayesian Decision-Making Under Misspecified Priors with Applications to Meta-Learning](https://proceedings.neurips.cc/paper/2021/hash/ddcbe25988981920c872c1787382f04d-Abstract.html)

* [Hierarchical Bayesian Bandits](https://proceedings.mlr.press/v151/hong22c.html)

It is just that their analysis is limited to linear models, which this paper overcomes. The paper has three weaknesses:

* Strong assumptions in the analysis, such as the arm separability in Assumption 4.1 and the uniformity conditions in Theorem 4.2. These assumptions lead to a regret bound where the model complexity does not appear, which is unusual. While this may be common in some neural bandit analyses, it needs to be discussed in detail and explained.

* The algorithm cannot be run as analyzed and requires hyper-parameter tuning. Therefore, it is hard to run online and its impact on practice may be minimal.

* Some experiments require heavy and unnatural preprocessing (Appendix A.2). For instance, in "ML Datasets", Mnist and Notmnist datasets are merged together and each is treated as a user cluster.

This paper was discussed in an online meeting with all reviewers. It is a borderline with a bias to reject.

**Summary Of Ac-Reviewer Meeting:**

All points in the meta-review were discussed. At the end, the most positive reviewer agreed that they are fine with a weak reject, reject and bump up if there are not better papers.